# Computational design of non-porous pH-responsive antibody nanoparticles

**Erin C. Yang** [1,2,3], **Robby Divine**[1,2,4,5], **Marcos C. Miranda** [1,6],
**Andrew J. Borst** [1,2], **Will Sheffler**[1], **Jason Z. Zhang** [1,2], **Justin Decarreau**[1,2],
**Amijai Saragovi**[1,2], **Mohamad Abedi**[1,2], **Nicolas Goldbach** [1,7],
**Maggie Ahlrichs** [1,2], **Craig Dobbins**[1,2], **Alexis Hand**[1,2], **Suna Cheng**[1,2],
**Mila Lamb**[1,2], **Paul M. Levine**[1,2], **Sidney Chan** [1,2], **Rebecca Skotheim**[1,2],
**Jorge Fallas**[1,2], **George Ueda** [1,2], **Joshua Lubner** [1,2], **Masaharu Somiya** [1,8],
**Alena Khmelinskaia**[1,9,10], **Neil P. King** [1,2] ✉ & **David Baker** [1,2,11] ✉

Programming protein nanomaterials to respond to changes in environmental conditions is a current challenge for protein design and is important for targeted delivery of biologics. Here we describe the design of octahedral non-porous nanoparticles with a targeting antibody on the two-fold symmetry axis, a designed trimer programmed to disassemble below a tunable pH transition point on the three-fold axis, and a designed tetramer on the four-fold symmetry axis. Designed non-covalent interfaces guide cooperative nanoparticle assembly from independently purified components, and a cryo-EM density map closely matches the computational design model. The designed nanoparticles can package protein and nucleic acid payloads, are endocytosed following antibody-mediated targeting of cell surface receptors, and undergo tunable pH-dependent disassembly at pH values ranging between 5.9 and 6.7. The ability to incorporate almost any antibody into a non-porous pH-dependent nanoparticle opens up new routes to antibody-directed targeted delivery.

There is considerable interest in tailoring nanoparticle platforms for targeted delivery of therapeutic molecules. Effective nanoparticle platforms for targeted delivery require in vitro cargo encapsulation, followed by target recognition, triggered nanoparticle disassembly and controlled cargo release once inside the cell[1–13]. Although several self-assembling protein nanoparticles with customized structures have been designed, they are composed of just one or two static building blocks, and efforts to adapt them for cargo-packaging and delivery applications are still in their infancy[1,14–20]. Antibodies are particularly attractive targeting moieties for delivery applications, and several previous studies have described various ways in which antibodies can be incorporated into nanoparticle delivery platforms[21–25]. We recently reported the computational design of antibody-incorporating nanoparticles in which a designed homo-oligomer drives the assembly of the antibody of interest into bounded, multivalent, polyhedral architectures[26] (Fig. 1a). Although such antibody nanoparticles can activate signaling through a variety of cell surface receptors, they have large pores, complicating the packaging and retention of molecular cargoes.

[1]Institute for Protein Design, University of Washington, Seattle, WA, USA. [2]Department of Biochemistry, University of Washington, Seattle, WA, USA. [3]Graduate Program in Biological Physics, Structure & Design, University of Washington, Seattle, WA, USA. [4]Graduate Program in Biochemistry, University of Washington, Seattle, WA, USA. [5]Department of Chemistry, University of California, Davis, Davis, CA, USA. [6]Department of Medicine Solna, Division of Immunology and Allergy, Karolinska Institutet and Karolinska University Hospital, Stockholm, Sweden. [7]Technical University of Munich, Munich, Germany. [8]SANKEN, Osaka University, Osaka, Japan. [9]Transdisciplinary Research Area 'Building Blocks of Matter and Fundamental Interactions (TRA Matter)', University of Bonn, Bonn, Germany. [10]Life and Medical Sciences Institute, University of Bonn, Bonn, Germany. [11]Howard Hughes Medical Institute, University of Washington, Seattle, WA, USA. ✉e-mail: neilking@uw.edu; dabaker@uw.edu

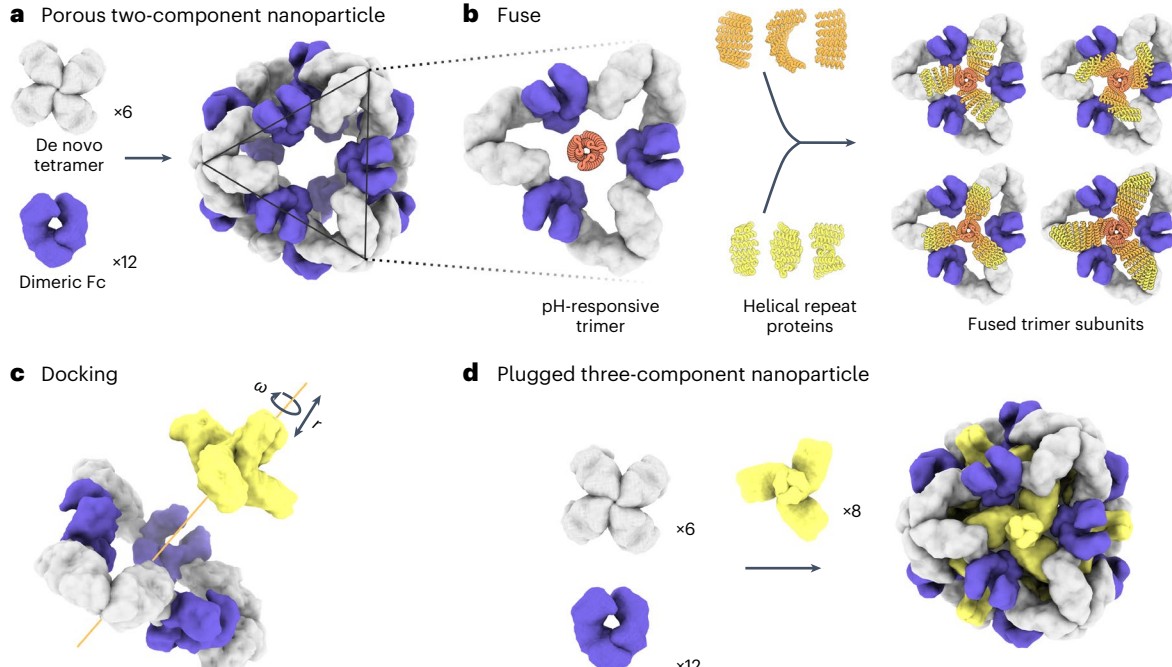

**Fig. 1 | Design of symmetry-matched plugs to fill empty symmetry axes in protein nanoparticles. a**, Six de novo tetramers (gray) and 12 dimeric Fc domains (purple) assemble into a porous octahedral O42 nanoparticle. The tetramers are aligned along the four-fold symmetry axis, and the Fc domains along the two-fold symmetry axis. **b**, Combinations of helical repeat proteins are fused to each other and to the pH trimer subunit at regions of high backbone overlap between pairs of helices to generate fused trimer subunits large enough to fully occupy the void along the three-fold axis in the original nanoparticle. **c**, The three-fold symmetry axes of the resulting pH-dependent trimeric fusions and the nanoparticle are aligned. Favorable docked arrangements are then generated by sampling rotations and translations along this axis. **d**, The resulting docked three-component nanoparticles have eight new trimeric plug subunits (yellow) that occupy the three-fold symmetry axes of the octahedral architecture. UCSF ChimeraX 1.6 (ref. 54) and the PyMOL Molecular Graphics System version 2.5 (Schrödinger) were used to create **a**–**d**.

## Results

### Design and biophysical characterization

To enable packaging and pH-dependent release of molecular cargoes, we sought to computationally design pH-dependent 'plugs' for antibody nanoparticles that, in environments above pH 7, close off the apertures in the original designs, but dissociate at the lower pH values that are characteristic of the endosome and tumor microenvironment[27]. We focused on octahedral antibody nanoparticles (O42.1) constructed from a $C_4$-symmetric designed tetramer and $C_2$-symmetric IgG dimers[26] (Fig. 1a). The eight $C_3$ axes in the nanoparticle are unoccupied and feature triangular pores that are 13 nm in length. On these $C_3$ axes, we aimed to incorporate a designed pH-dependent $C_3$ trimer[28] and tune it to disassemble at the pH of the endosomal environment (Fig. 1b). We reasoned that such three-component nanoparticles could (1) encapsulate molecular cargoes without leakage, (2) selectively enter target cells and (3) disassemble in the acidic environment of the endosome.

The previously designed pH-dependent trimer is much smaller than the aperture along the $C_3$ axis of the octahedral nanoparticle (Fig. 1b); hence, filling the $C_3$ axis with the pH-dependent trimer requires extending the backbone such that it makes shape-complementary interactions with the designed tetramer. To enable this, we combined helical fusion[29,30] and protein-docking[31] approaches into a single design pipeline. We extended the pH-dependent trimer by fusing combinations of helical-repeat-protein building blocks onto each subunit to generate more than 80,000 distinct $C_3$ fusions with helical repeats of variable geometry extending outwards from the $C_3$ axis (Fig. 1b and Methods). The resultant diverse set of $C_3$ building blocks was docked into the three-fold-symmetric pore by aligning both $C_3$ axes and sampling translational and rotational degrees of freedom along this axis[31] and varying the lengths of the repeat protein arms (Fig. 1c and Methods). The resulting 'plugged' octahedral assemblies (O432) contain

12 IgG1 crystallizable fragment (Fc) domains along the octahedral two-fold axes, six tetramers along the octahedral four-fold axes and eight trimeric plugs along the octahedral three-fold axes (Fig. 1d).

The newly generated interfaces between the pH-dependent plugs and the octahedral assembly were evaluated for designability using a combination of the residue pair transform (rpx) score—a prediction of interaction energy following sequence design[31,32]—and overall shape complementarity at the interface[33]. For 6,000 docks predicted to have high designability and shape complementarity, the amino acid sequences at the newly formed fusion junctions and at the interface between the trimeric plug and antibody nanoparticle were optimized using Rosetta sequence design calculations[34], generating substitutions on both the trimer and the tetramer subunits. Designed interfaces were evaluated for secondary structure contacts and chemical complementarity, and 45 designed trimeric plug and nanoparticle tetramer pairs were selected for experimental characterization.

Designed trimers and tetramers were expressed bicistronically and co-purified by immobilized metal affinity chromatography (IMAC) purification. SDS–polyacrylamide gel electrophoresis (SDS–PAGE) revealed that 16 out of 45 trimer designs co-eluted with the tetramer, suggesting that the trimeric and tetrameric components are associated (Extended Data Fig. 1). To form the three-component nanoparticle, a superfolder green fluorescent protein (sfGFP)-Fc fusion protein[26] was added to the clarified lysates of co-expressed trimers and tetramers, and the resulting mixtures were subjected to IMAC. For 5 out of the 16 designs, all three components co-purified, as assessed by SDS–PAGE and native PAGE (Extended Data Fig. 2a,b). To enable controllable nanoparticle assembly in vitro, the genes in these designs that encode the trimers and tetramers were subcloned into separate expression vectors containing an amino- or carboxy-terminal 6×-histidine tag, and the independently expressed oligomers were purified separately

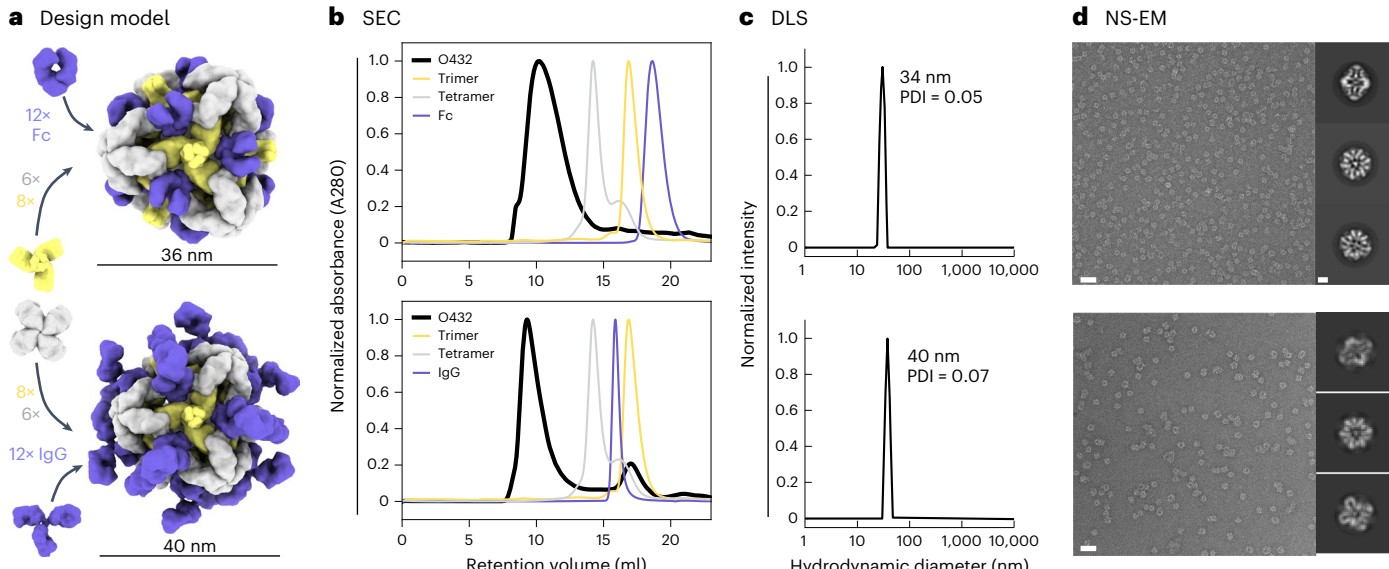

**Fig. 2 | Efficient assembly of three-component nanoparticles from independently purified components for both Fc and IgG. a**, Design models with Fc or IgG (purple), designed nanoparticle-forming tetramers (gray) and pH-dependent plug-forming trimers (yellow). **b**, Overlay of representative SEC traces of the full assembly formed by mixing designed tetramers, trimers and Fc or IgG (black) with those of the single components in gray (tetramer), yellow (trimeric plug) or purple (Fc or IgG). **c**, Representative DLS of fractions collected from the O432 assembly peak, showing average hydrodynamic diameters of 34 nm (polydispersity index (PDI), 0.05) and 40 nm (PDI, 0.07) for the O432 assemblies with Fc and full-length IgG, respectively. **d**, Negatively stained electron micrographs with reference-free 2D class averages along each axis of symmetry in inset; electron microscopy images were collected prior to SEC purification. Scale bars, 100 nm and 10 nm for the micrograph and 2D averages, respectively. The PyMOL Molecular Graphics System version 2.5 (Schrödinger) was used to create **a**.

by size-exclusion chromatography (SEC) (Extended Data Fig. 3a). The SEC elution profiles of the redesigned tetramers were similar to those of the parent O42.1 tetramer, O42.1 C4 (Extended Data Fig. 3b)[26]. Three-component assemblies were prepared by mixing the three purified proteins—the trimeric plug, designed tetramer and Fc of human IgG1—in a 1:1:1 stoichiometric ratio, followed by overnight incubation and SEC purification (Extended Data Fig. 3c; owing to tetramer insolubility, insufficient material was produced to prepare a three-component assembly reaction for one design, O432-43). Mixing the three components in a 1:1:1 stoichiometric ratio yielded elution peaks in the void volume, with a shoulder peak between 9 and 11 ml at the elution volume of the original O42.1 antibody nanoparticle[26] (Extended Data Fig. 3c,d).

For one of the designed assemblies, O432-17, both peaks contained all three protein components (Extended Data Fig. 3e,f). Optimization of the stoichiometric ratios of the components during in vitro assembly increased the yield at the expected O42.1 elution volume (Fig. 2b; the optimal assembly ratio per protomer of purified trimeric plug, tetramer and Fc was 1.1:1:1). Assembly of the designed nanoparticle was cooperative and required all three components: mixing any two of the three O432-17 components stoichiometrically did not result in fully assembled nanoparticles (Extended Data Fig. 3g–i). This cooperativity simplifies preparation of the three-component nanoparticles because it prevents incomplete assembly or assembly hysteresis of nanoparticles containing two out of the three components, thus eliminating the need for further purification steps to separate these species from the intended three-component assembly. Dynamic light scattering (DLS) of the optimized O432 peak indicated that the hydrodynamic diameter was 34 nm (Fig. 2c), and negative-stain electron microscopy (NS-EM) revealed the presence of monodisperse nanoparticles (Fig. 2d). Two-dimensional (2D) class averages of negatively stained micrographs revealed plug-like density in the three-fold views compared with the original two-component antibody nanoparticle.

For downstream delivery applications, we tested whether the O432-17 nanoparticle would assemble when the designed trimer and tetramer were co-incubated with full-length α-EGFR IgG antibodies containing both Fc and Fab domains (Fig. 2a). The O432-17 design eluted in the void volume, owing to the increased diameter from the additional Fab domains (Fig. 2b). DLS of this void volume peak revealed a monodisperse hydrodynamic diameter of 40 nm, in line with the expected diameter of the IgG-containing O432-17 assembly (Fig. 2c). NS-EM micrographs and 2D class averages of the peak fraction exhibited plug-like density in the three-fold view following 2D classification, as well as Fab-like density in the 2-fold, 3-fold and 4-fold views (Fig. 2d). Owing to the inherent flexibility of the antibody Fc–Fab junction[35], the Fab domain density was not well resolved (Fig. 2d).

**Cryo-EM analysis of O432-17**

We sought to determine the structure of the O432-17 nanoparticles using single-particle cryogenic electron microscopy (cryo-EM) (Fig. 3a–d). Following data collection and preprocessing of raw micrographs, a subset of the 2D averages with fully assembled O432-17 nanoparticles (Extended Data Fig. 4a,b) was used to generate an ab initio three-dimensional (3D) reconstruction in the absence of applied symmetry (Extended Data Fig. 4c). 3D heterogeneous refinement yielded four classes (Extended Data Fig. 4d), all corresponding to fully plugged O432 antibody nanoparticles, with six designed tetramers at the four-fold octahedral symmetry axes forming interfaces with eight designed trimers at the three-fold symmetry axes and 12 dimeric Fc fragments at the two-fold symmetry axes (Table 1). A subsequent 3D refinement containing particles from all four classes was generated after applying octahedral symmetry, resulting in a final map with an estimated global resolution of 7 Å (Extended Data Fig. 4e).

We fit the O432-17 design model into the experimentally determined cryo-EM map using backbone refinement in Rosetta, yielding a density-refined model. The model contains the plug, tetramer and Fc, and is in close agreement with the original design (Fig. 3d). The helices of the tetramers at the four-fold axes are clearly evident in the cryo-EM density, with each helical repeat extending from the four-fold axis of the octahedral architecture to bind the Fc, which exhibits little or no structural distortion. The designed trimeric plugs are also clearly

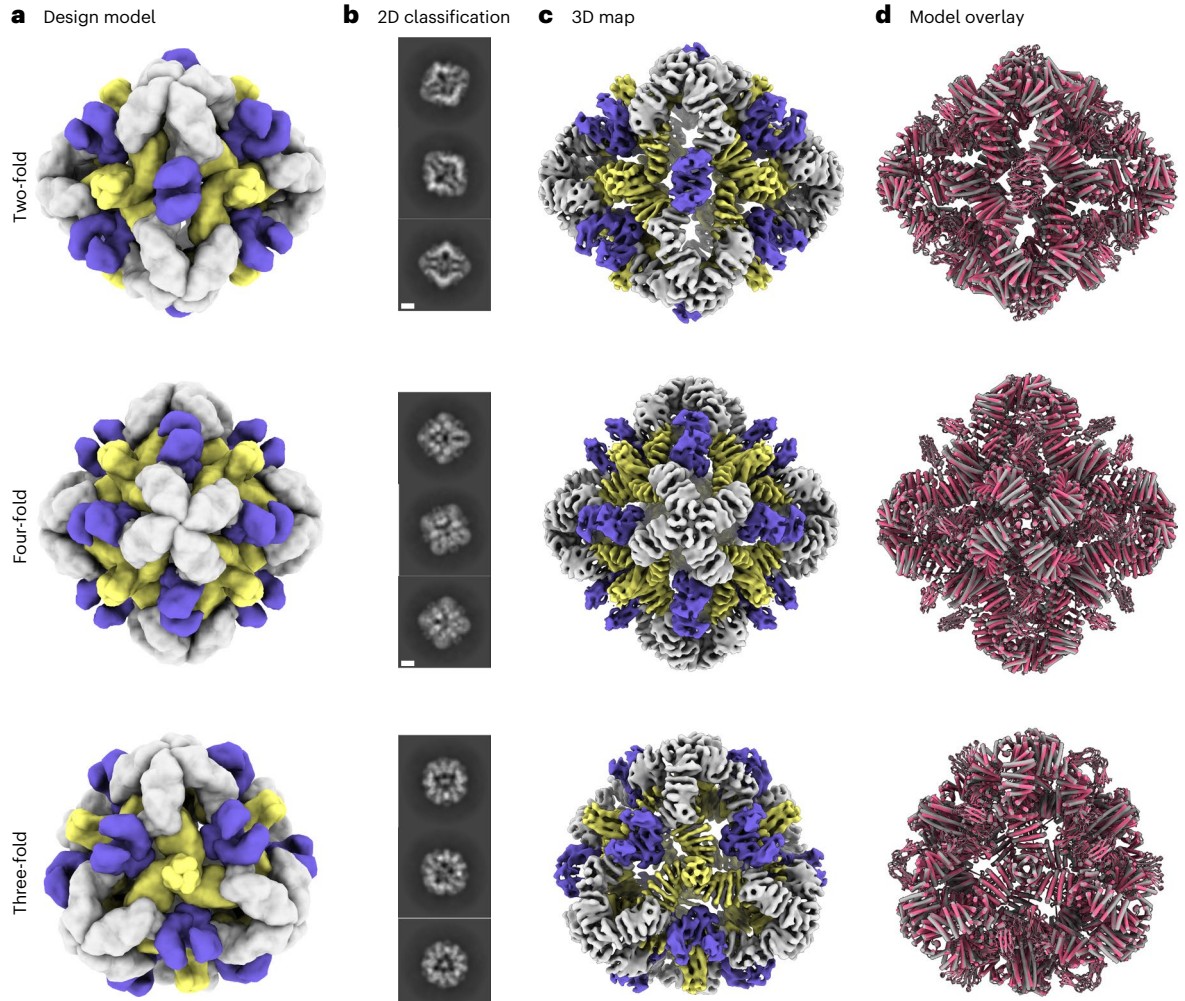

**a** Design model  **b** 2D classification  **c** 3D map  **d** Model overlay

**Fig. 3 | Cryo-EM analysis of 72-subunit nanoparticles composed of three structural components. a**, Cryo-EM characterization of the O432 assembly with Fc before SEC purification. Computational design models viewed along each axis of symmetry of the octahedral architecture are shown. **b**, Representative 2D class averages along each axis of symmetry. Scale bars, 10 nm. **c**, 3D EM density map with 7 Å resolution, reconstructed from the collected dataset. **d**, Overlay of the design model (gray) and the model refined into the 3D EM density map (pink), showing high agreement. UCSF ChimeraX 1.6 (ref. 54) was used to create **a**.

evident in the cryo-EM density, with helices along the three-fold axis of the octahedral architecture that extend to form an interface with the tetramers. The addition of the trimeric plug reduces the porosity of the antibody nanoparticle, as intended: the largest diameter pore is 3 nm, compared with 13 nm in the original O42.1 nanoparticle[26]. The helices forming the trimer and tetramer interface are largely consistent between the design and the model refined into density (Extended Data Fig. 4f). The density-refined model of the plugged O432-17 nanoparticle is substantially closer to the original O42.1 design model than to the previously determined O42.1 cryo-EM structure: the average Cα root mean squared deviation (r.m.s.d.) of the asymmetric unit was 1.6 Å between the O432-17 density-refined model and the design model, 1.9 Å between the O432-17 design model and O42.1 density-refined model, and 4.2 Å between the O42.1 density-refined model and the O42.1 design model (Extended Data Fig. 5a–c). These results suggest that the addition of the trimeric plug buttresses the tetramer into a conformation that more closely matches that of the original design model, and could also reduce the overall flexibility of the system, as compared with the original O42.1 design.

## Trimeric-plug-dependent packaging of molecular cargoes

We next set out to redesign the nanoparticles to package and protect molecular cargoes through electrostatic interactions[17,36,37]. We generated variants with either highly positively or highly negatively charged interiors, and focused on changing interior surface residues of the trimeric plug. Substitutions to amino acids with the desired charge (or no charge) were preferred.

We screened for packaging of nucleic acids by assembling positively charged trimeric plug variants, the designed tetramer and cetuximab (CTX), an antibody to EGFR, with a 154-nucleotide (nt) prime editing guide RNA (pegRNA) cargo[38–40] (Fig. 4a). Packaging and protection of nucleic acid cargo in the presence or absence of nuclease was assessed through non-denaturing electrophoresis, staining with SYBR Gold and Coomassie to detect RNA and protein, respectively. One variant, O432-17(+), showed co-migration of the nucleic acid and nanoparticle, both with and without treatment with the Benzonase nuclease (Fig. 4b). The RNA-packaging three-component assembly migrated similarly to an assembly reaction lacking RNA cargo, and negatively stained electron micrographs and corresponding 2D class averages also closely resembled those of empty three-component nanoparticles (Extended Data Figs. 6a,b and 7a–f). Excess nucleic acid that did not co-migrate with the protein was degraded in the Benzonase-treated sample, demonstrating that the nanoparticles protected the nucleic acid that co-migrated with O432-17(+). We did not observe co-migration of O432-17(+) and pegRNA after RNAse A treatment (Fig. 4b,c); the remaining pore size after the addition of the plug is likely still large enough to admit the 14-kDa enzyme but small enough to exclude the 60-kDa Benzonase[19]. Nucleic acids that co-migrated with the plugless

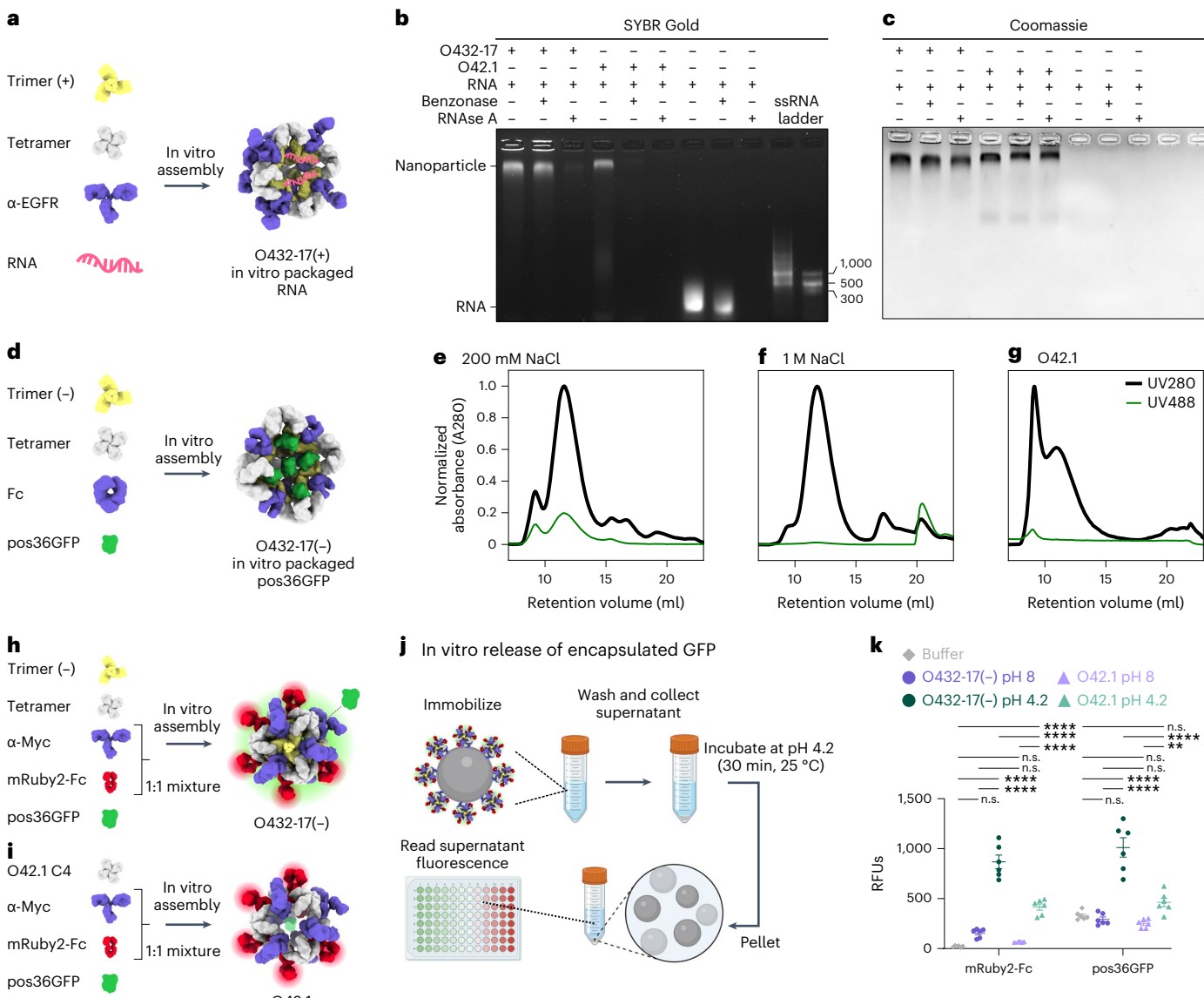

**Fig. 4 | Plugged antibody nanoparticles electrostatically package protein and nucleic acid cargoes. a**, Designed positively charged trimer variants were assembled by incorporating the designed tetramer, pegRNA and antibody to EGFR (α-EGFR) to form O432-17(+) nucleocapsids. **b,c**, O432-17(+) nucleocapsids and O42.1 assembled in vitro with RNA were treated with Benzonase or RNAse A for 1 h, electrophoresed on non-denaturing 0.8% agarose gels and stained with SYBR Gold (**b**; nucleic acid) and Coomassie (**c**; protein). **d**, Designed negatively charged trimer variants were assembled with designed tetramer, pos36GFP and human Fc to form O432-17(−) nanoparticles with in vitro-packaged pos36GFP. **e,f**, SEC chromatograms of in vitro packaging reactions involving O432-17(−) and pos36GFP were performed in either 200 mM NaCl (**e**) or 1 M NaCl (**f**). **g**, As a comparison, SEC chromatograms of in vitro-assembled O42.1 with pos36GFP in 200 mM NaCl showed no co-migration of pos36GFP. Absorbance was monitored at 280 nm (black) and 488 nm (green). **h**, O432-17(−) nanoparticles for in vitro release of pos36GFP were assembled with negatively charged trimer, designed tetramer, pos36GFP and a 1:1 mixture of a Myc-targeted monoclonal antibody (α-Myc) and mRuby2-Fc. **i**, As a comparison, O42.1 was assembled in vitro with O42.1 C4, pos36GFP and a 1:1 mixture of antibody to Myc and mRuby2-Fc.

**j**, Experimental design for in vitro release of encapsulated pos36GFP in acidic conditions. Assembled nanoparticles were immobilized on Myc-peptide-coated *S. cerevisiae* yeast cells. Cells and supernatant were collected by centrifugation and resuspended and incubated in acidic conditions. The supernatant containing released components and cargo was buffer exchanged to pH 8, and fluorescence intensity was analyzed. **k**, Fluorescence intensity of the supernatant of O432-17(−) and O42.1 assembled with pos36GFP and mRuby2-Fc before and after acidic incubation. mRuby2-Fc fluorescence was used as an indicator of nanoparticle assembly; a positive fluorescence signal indicates nanoparticle disassembly. Positive pos36GFP fluorescence indicates release of pos36GFP cargo. *$P \le 0.05$, **$P \le 0.01$, ***$P \le 0.001$, ****$P < 0.0001$, two-way ANOVA with Tukey's correction for multiple comparisons. n.s., not significant. Data are presented as mean values ± s.e.m. and measured over three independent samples in duplicate (Supplementary Table 6). UCSF ChimeraX 1.6 (ref. 54) and the PyMOL Molecular Graphics System version 2.5 (Schrödinger) were used to create **a**, **d**, **h** and **i**. **j** was created using BioRender. GraphPad Prism version 9.3.1 (GraphPad Software) was used to create **k**. RFUs, relative fluorescence units.

O42.1 nanoparticle were degraded after both Benzonase and RNAse A treatment, demonstrating that the trimeric plug is required for packaging and protection of nucleic acid cargo (Fig. 4b,c).

We also designed negatively charged plug variants to package positively charged protein cargoes. We screened for packaging of positively charged GFP (pos36GFP)[41] by assembling negatively charged trimeric plug variants with different interior surface charges with tetramer, Fc and pos36GFP (Fig. 4d). For one variant, O432-17(−), both the 488 nm GFP and the 280 nm nanocage signals were detected in the elution profile during SEC, and NS-EM confirmed that three-component nanoparticle

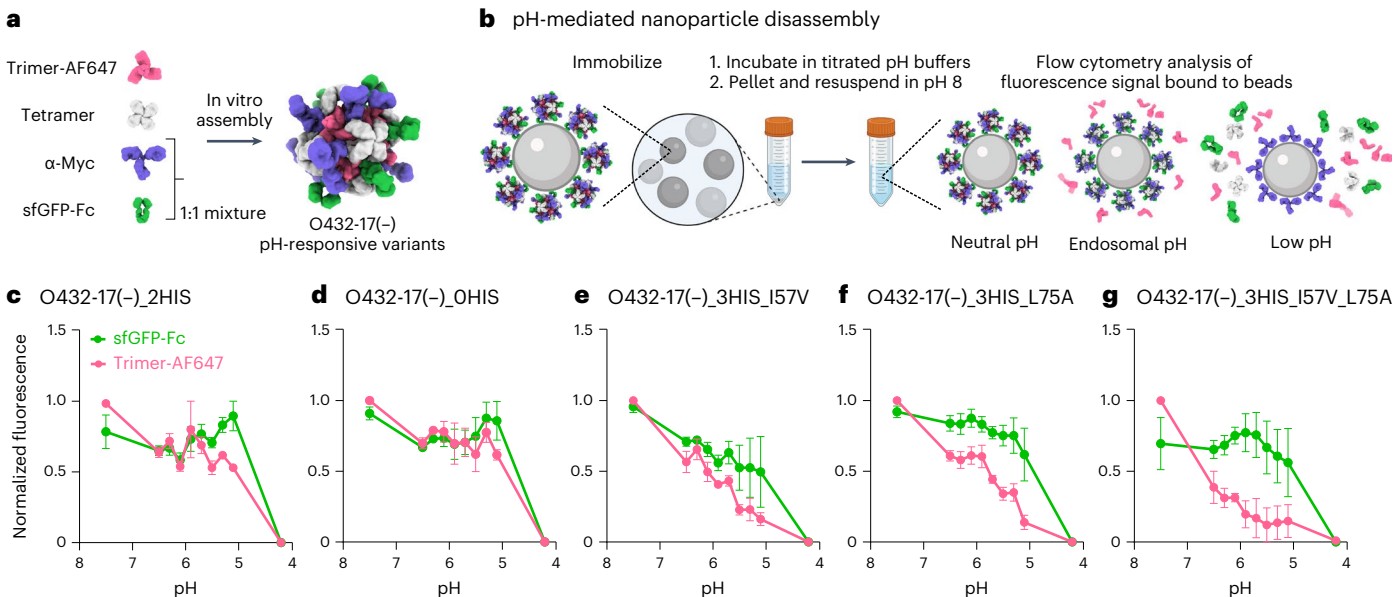

**Fig. 5 | Plugged antibody nanoparticles exhibit tunable dissociation in response to acidification. a**, O432-17 nanoparticles were assembled with AF647-conjugated trimeric plug variants, designed tetramer, sfGFP-Fc and α-Myc antibody. **b**, O432-17 nanoparticles were immobilized on Myc-peptide-coated beads and incubated in titrated pH buffers. Beads were washed by centrifugation and resuspended in 25 mM Tris and 500 mM NaCl, pH 8, and the remaining trimer and sfGFP-Fc fluorescence on the beads was analyzed by flow cytometry. **c–g**, AF647 and sfGFP fluorescence levels were normalized to the minimum and maximum values across the titration and analyzed as a function of the pH for each nanoparticle variant (O432-17(−)_2HIS (**c**), O432-17(−)_0HIS (**d**), O432-17(−)_3HIS_I57V (**e**), O432-17(−)_3HIS_L75A (**f**), O432-17(−)_3HIS_I57V_L75A (**g**)). Data are normalized to the minimum and maximum fluorescence per titration and presented as mean ± s.e.m. for three biologically independent replicates. UCSF ChimeraX 1.6 (ref. 54) and the PyMOL Molecular Graphics System version 2.5 (Schrödinger) were used to create **a**; **b** was created using BioRender; GraphPad Prism version 9.3.1 (GraphPad Software) was used to create **c–g**.

assembly had occurred (Fig. 4e and Extended Data Fig. 8c,d). On the basis of the fluorescence and absorbance of the assemblies, approximately nine to ten pos36GFPs are packaged per nanoparticle in 200 mM NaCl, occupying roughly 40% of the interior volume. SEC revealed that GFP packaging did not occur in the presence of 1 M NaCl (Fig. 4f), indicating that packaging is largely driven by electrostatic interactions between the cargo and nanoparticle interior. In vitro assembly of the unplugged O42.1 (ref. 26) with pos36GFP did not result in cargo packaging (Fig. 4g). SEC, DLS and NS-EM confirmed O432-17(−) assembly in the absence of cargo, suggesting that assembly stability is cargo-independent (Extended Data Fig. 8a–d). Thus, our designed three-component nanoparticles efficiently encapsulate and protect molecular cargoes.

We next sought to release encapsulated pos36GFP cargo through pH-mediated disassembly of O432-17(−) nanoparticles. O432-17(−) trimer, tetramer and pos36GFP were assembled in vitro using a 50:50 mixture of mouse antibody to Myc and mRuby2-Fc, the latter of which enabled monitoring of nanoparticle disassembly (Fig. 4h). As a comparison, we also assembled O42.1 with the same antibody cocktail and pos36GFP (Fig. 4i). Assembled nanoparticles were immobilized on *Saccharomyces cerevisiae* cells expressing the Myc peptide. The cells were collected by centrifugation and then resuspended in citrate–phosphate buffer at low or high pH (4.2 and 8, respectively) for 30 min, after which the supernatant was collected and buffer exchanged to neutral pH before fluorescence was measured (Fig. 4j). Although both O42.1 and O432-17(−) nanoparticles remain intact at high pH and disassemble at low pH, we observed considerably more pH-dependent pos36GFP cargo release from the plugged O432-17(−) nanoparticles than from the O42.1 nanoparticles, because the absence of the plug precluded cargo packaging and retention (Fig. 4k). The designed trimeric plug on O432-17(−) nanoparticles thus enables packaging, retention and pH-mediated release of encapsulated cargo.

## Tuning pH-dependent plug dissociation

We next explored the potential for disassembly of O432-17(−) and O432-17(+) nanoparticles in the pH range of the endosome and lysosome (pH 4.5–6.5)[42]. We monitored the dissociation of fluorescently labeled plug trimers and antibody Fc fused to GFP (sfGFP-Fc) from the nanoparticles at low pH through flow cytometry[43–45]. O432-17 nanoparticles assembled with tetramer, Alexa Fluor 647 (AF647)-conjugated trimeric plug and a 50:50 mixture of mouse monoclonal antibody to Myc and sfGFP-Fc (Fig. 5a) were immobilized onto 3 μm Myc-tag-coated polystyrene beads and incubated in citrate–phosphate buffers ranging from pH 4.2 to 7.5. The beads were collected by centrifugation and resuspended in buffer at pH 8, and then were analyzed using flow cytometry (Fig. 5b). Loss of AF647 and sfGFP signals indicates disassembly and release of plug trimers and sfGFP-Fc from the nanoparticles, respectively (Extended Data Fig. 8a–d).

We found that by varying the number of histidine-containing hydrogen-bond networks and the hydrophobic packing with the trimer, the pH of plug disassembly could be tuned over a remarkably wide range. The O432-17(−) plug design (O432-17(−)_2HIS) with two histidine hydrogen-bond networks disassembled at pH 5.1, similar to O432-17(−)_0HIS, a negative control trimer variant in which all histidines are substituted with asparagine (Fig. 5c,d). O432-17(−)_3HIS_I57V and O432-17(−)_3HIS_L75A, which have three histidine networks and additional core packaging destabilization, assembled robustly (Extended Data Fig. 9a–d) and showed clear pH-dependent release of the pH trimer at pH of above 5.1 (Fig. 5e,f). The apparent dissociation constant ($pK_a$) values of trimer dissociation (the pH at which the fluorescence was 50% of the maximum signal) were pH 6.1 and pH 5.9 for O432-17(−)_3HIS_I57V and O432-17(−)_3HIS_L75A, respectively; nanoparticle disassembly, as measured by release of sfGFP-Fc, occurred only at lower pH (pH 5.3 and 4.7, respectively). For plug release, O432-17(−)_3HIS_I57V_L75A had a remarkably high apparent $pK_a$ of pH 6.7 (Fig. 5g; the nanoparticle only disassembled at pH 5). pH-mediated release of encapsulated cargo was observed for O432-17(−)_2HIS (Fig. 4k), but could not be investigated for the other O432-17(−) variants, because pos36GFP was not well packaged, perhaps owing to altered interactions with the plug. Like O432-17(−), histidine-containing variants of O432-17(+) disassembled at a more basic pH than did the parental construct (Extended Data

**Table 1 | Cryo-EM data collection statistics**

|  | O432-17 (EMD-29602) |
|---|---|
| **Data collection and processing** | |
| Magnification | ×105,000 |
| Voltage (kV) | 300 |
| Electron exposure (e⁻/Å²) | 63.775 |
| Defocus range (µm) | 0.8–1.7 |
| Pixel size (Å) | 0.42 |
| Symmetry imposed | O |
| Initial particle images (no.) | 88,357 |
| Final particle images (no.) | 50,017 |
| Map resolution (Å) FSC threshold | 7.05 Å |
| Map resolution range (Å) | 6–7.5 Å |

Figs. 7a–d and 8e,f). These results demonstrate that nanoparticle dissociation can be tuned over a wide pH range.

### Entering target cells through receptor-mediated endocytosis

We next tested the ability of the O432-17 nanoparticles to enter cells through receptor-mediated endocytosis. We assembled targeted nanoparticles by incubating a 1:1 stoichiometric mixture of CTX and Fc fused to mRuby2 (mRuby2-Fc) with the tetramer and the 3HIS_I57V_L75A trimeric plug variant labeled with AF647 (named O432-17-CTX). We assembled a non-EGFR-targeting nanoparticle (named O432-17-Fc) as a negative control by mixing the tetramer and trimer-AF647 with mRuby2-Fc (Extended Data Fig. 10a). Assembled O432-17-CTX and O432-17-Fc were incubated with A431 cells, which have high levels of EGFR expression, and imaged with epifluorescence microscopy. The percentage of A431 cells expressing O432-17-CTX (44%) was substantially higher than the percentage expressing O432-17-Fc (5%), suggesting that the O432-17(−) nanoparticles require EGFR binding for efficient endocytosis (Extended Data Fig. 10b,c). The EGFR-targeted nanoparticle bound to 45% and 68% of wild-type HeLa cells, which express moderate levels of EGFR with and without serum, respectively; EGFR-knockout HeLa cells were not efficiently labeled in either condition (10% with serum and 7% without serum) (Extended Data Fig. 10f,g; the reduced labeling in the presence of serum likely reflects a slight instability in the nanoparticle structure in the presence of high concentrations of exogenous antibody[26]). Together, these data establish that the antibody component of O432-17 nanoparticles enables targeting of cells expressing a specific receptor.

### Discussion

We describe a general approach for reducing the porosity of protein nanomaterials by designing custom symmetric plugs that fill pores along unoccupied symmetry axes. We used this approach to generate the first designed non-covalently-associated protein nanoparticles with distinct structural components on three axes of symmetry. In contrast to functionalized homomeric and designed two-component protein nanoparticles[1,8,14,17,18,26,36,46–49], each of the three components in our designs has a specific functional role. The trimeric plug enables pH-responsive disassembly and packaging (and release) of cargoes, the antibody provides cell-targeting functionality and the tetramer drives assembly of the three-component nanoparticle by interacting with both the trimer and Fc domain. This division of labor makes our designed system highly modular: targeting specificity can be altered simply by switching the antibody component to target cells of interest, and the pH of disassembly and cargo-packaging specificity can be programmed by choosing the appropriate trimeric plug. Assembly of the nanoparticles in vitro from independently purified components

enables facile inclusion of multiple variants of each component, such as multiple distinct antibodies.

Engineering drug-delivery platforms to induce nanoparticle disassembly at a biologically relevant pH is a critical challenge. Our results represent a test of our understanding of protein disassembly and assembly dynamics[50–53]. We finely tuned the apparent p$K_a$ of disassembly of our O432 system through a combination of histidine-containing hydrogen-bonding networks and cavity-introducing substitutions that weaken hydrophobic interactions at the trimeric component's oligomeric interface. Through this approach, we raised the apparent p$K_a$ of disassembly to a remarkably high pH of 6.7, well above the pH of the endosome and in the range of many tumor microenvironments, and close to the maximum achievable value, given the p$K_a$ of histidine.

Our O432 nanoparticle system is capable of packaging both protein and nucleic acid cargoes, protects nucleic acid cargoes from degradation, disassembles at biologically relevant pH with precise tunability, incorporates a variety of targeting moieties and is readily internalized by target cells. In performing these functions, the O432 nanoparticles resemble viruses. They also resemble the smallest viruses in size, with an internal diameter (25 nm) that is slightly larger than that of adeno-associated viruses (18 nm). However, the architecture of the O432 nanoparticles differs greatly from small icosahedral viruses, many of which are constructed from multiple copies of a single capsid protein that assumes several different conformations and performs several distinct functions at different stages of the viral life cycle. By contrast, the O432 nanoparticles are constructed from three modular protein components that each perform a specific function. The tunable pH dependence makes this system a particularly attractive platform for engineering release and delivery of drugs during early stages of endosomal maturation. However, to be broadly useful, future designs for intracellular biologics-delivery systems will need to incorporate endosomal escape machinery. The nanoparticles are well suited in their current form for conditional delivery of cytotoxic tumor-killing or tumor-modulating drugs into the tumor microenvironment with tumor-specific antibodies; by providing an additional checkpoint on proper localization, the pH-dependent release of cargo in the lower pH tumor microenvironment could minimize off-tumor toxicity and systemic exposure compared with classic direct antibody-conjugation approaches. Overall, the pH-dependent disassembly, programmability and versatility of the O432 platform provide multiple exciting paths forward for biologics delivery.

### Online content

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

## Methods

### Extension of pH-responsive trimeric building blocks

We used the WORMS helical fusion software[29] to generate fusions between the base pH-trimer helical bundle and pairs of 82 helical repeat proteins, designed to cover a broad range of axial and radial displacement between repeat units[55]. Scaffold information was prepared in a json file, which is used as an input parameter in the WORMS software through the --dbfiles option. For each scaffold, the pdb file path ('file'), a geometric class specification ('class') and a connections dictionary ('connections') specifying the chain ('chain'), terminus direction ('direction') and residue positions that were available for fusion ('residues') were included in the json file. Geometric class specification for oligomeric scaffolds required either an N or C following the oligomeric architecture, to define the terminus available for subsequent fusions. A connections dictionary specifying each chain, the terminus direction and residue positions allowed for fusion was required for each chain and terminus direction. The oligomeric scaffold asymmetric unit required one connections dictionary, specifying fusion to one terminus and one chain. As monomers, helical repeat proteins require a list of two connections dictionaries, because they allow fusions to either the N or C terminus. An example json file example is shown below, where the connections dictionary specifies allowable fusions on the C terminus of the pH trimer from residue 50 and over, and anywhere along the helical repeat proteins:

```
{
{"file": "/path/to/ph_trimer.pdb",
  "class": ["C3_C"],
  "connections":[
  {"chain": 1, "direction": "C", "residues":["50:"]}
  ]
},
{"file": "/path/to/helicalrepeatprotein1.pdb",
  "class": ["Monomer"],
  "connections": [
  {"chain": 1, "direction": "N",
"residues":[":210"]},
  {"chain": 1, "direction": "C",
"residues":["-210:"]}
  ]
},
{"file": "/path/to/helicalrepeatprotein2.pdb",
  "class": ["Monomer"],
  "connections": [
  {"chain": 1, "direction": "N",
"residues":[":160"]},
  {"chain": 1, "direction": "C",
"residues":["-160:"]}
  ]
}
}
```

To prepare the WORMS software for fusions, we also included a map for backbone connections (--bbconn), which specifies the order in which scaffolds of each class can be fused. We used the following backbone connection map, which specifies that the N terminus of the fusion of two helical repeat proteins in the monomer class is fused to the C terminus of an oligomer with $C_3$ symmetry:

```
--bbconn C3 C3_C \
  NC Monomer \
  N_ Monomer \
```

Finally, we specified a --geometry NullCriteria() option, which allowed generation of arbitrary fusions of varying radius and length to the C terminus of the pH trimer.

### Computational docking of plugged antibody nanoparticles

The generated fusions served as docking inputs into the antibody nanoparticle through the --inputs1 and --inputs2 options. To prepare the antibody nanoparticle for axle docking along its $C_3$ axis, we generated a context structure PDB file by isolating the chains forming the $C_3$ symmetric aperture. We fed this context structure and each trimer fusion separately into the RPXDock application[31] using the axle docking protocol (--architecture AXLE_C3). To enable trimming, we set a maximum number of trimmed residues (--max_trim) and specified the chains representing the fusion to be trimmed (--trimmable_components). Finally, to enable greater sampling, we allowed the RPXDock application to sample the entire search space (--docking_method) at a cartesian and angular resolution of 0.25 angstroms and degrees, respectively (--grid_resolution_cart_angstroms and --grid_resolution_ori_degrees). An example axle docking executable is shown below.

```
PYTHONPATH = /path/to/rpxdock/site-packages /path/
to/python/environment /path/to/rpxdock/dock.py \

--architecture AXLE_C3 \
--inputs1 /path/to/plugfusion.pdb \
--inputs2 /path/to/contextstructure.pdb \
--max_trim 400 \
--cart_bounds -300 300 \
--docking_method grid \
--grid_resolution_cart_angstroms 0.25 \
--grid_resolution_ori_degrees 0.25 \
--hscore_files ailv_h \
--hscore_data_dir /path/to/rpxdock/hscore \
--trimmable_components 'A' \
--function stnd \
```

We evaluated the docked interfaces between each fusion and antibody nanoparticle using the RPX score and NContact metrics, which are predictive of geometric complementarity and designability. Empirically, we deemed docks with an RPX score greater than 50 and with more than one residue pair contact to be suitable for sequence optimization.

### Computational design of plugged antibody nanoparticles

We used the Rosetta Macromolecular Modeling suite to optimize the sequence residues contributing to either the fusion domains or interface between plug and nanoparticle. With residue selectors, we identified clashing residue positions as a result of the WORMS fusion and residues contributing to the interface between the plug and antibody nanoparticle. The sequence of the selected residues was optimized on the basis of the local positions of each residue defined by secondary structure, solvent accessible surface area (SASA) of the backbone atom and the Cβ atom and the number of amino acid side chains in a cone extending along the Cα–Cβ vector side chain neighbors. An example of the fusion, docking and design of O432-17 is included at https://github.com/erincyang/plug_design.git.

### Small-scale bicistronic bacterial protein expression

Synthetic genes encoding designed trimeric-plug and tetramer sequences bicistronically were purchased from Genscript in pET-29b+ vectors. They were connected by an intergenic region (5′-TAAAGAAGGAGATATCATATG-3′) and a C-terminal 6×histidine tag on the tetramer. Expression plasmids were transformed into BL21(DE3) *Escherichia coli* cells and were grown in LB medium supplemented with 50 mg l$^{-1}$ kanamycin at 37 °C overnight. The overnight culture was diluted into autoinduction medium and incubated at 37 °C overnight. Cells were lysed chemically in BugBuster supplemented with 1 mM PMSF and 20 mM imidazole, and were cleared by centrifugation. Clarified lysates were purified using IMAC with Ni-NTA magnetic beads. The soluble fractions were washed with 25 mM Tris pH 8.0, 300 mM NaCl

and 60 mM imidazole before elution with 25 mM Tris pH 8.0, 300 mM NaCl and 300 mM imidazole. Elution fractions from bicistronic expression plasmids were subsequently subjected to native PAGE and SDS–PAGE to identify slow migrating species that co-eluted two proteins of different molecular weights.

### Small-scale bicistronic bacterial protein expression for trimeric-plug and tetrameric designs for sfGFP-Fc lysate assembly
Synthetic genes encoding designed trimeric plug and tetramer sequences bicistronically were purchased from Genscript in pET-29b+ vectors and connected by an intergenic region (5′-TAAAGAAGGAGATATCATATG-3′) and a C-terminal 6×Histidine tag on the tetramer. Expression plasmids were transformed into BL21(DE3) *E. coli* cells and grown in LB medium supplemented with 50 mg l$^{-1}$ kanamycin at 37 °C overnight. The overnight culture was diluted into auto-induction medium and incubated at 37 °C overnight. Cells were lysed chemically in BugBuster supplemented with 1 mM PMSF and 20 mM imidazole, and were cleared by centrifugation. Clarified lysates were incubated with a 3 μM final concentration of sfGFP-Fc and purified by IMAC with Ni-NTA magnetic beads. The soluble fractions were washed with 25 mM Tris pH 8.0, 300 mM NaCl and 60 mM imidazole before elution with 25 mM Tris pH 8.0, 300 mM NaCl and 300 mM imidazole. Elution fractions from bicistronic expression plasmids were subsequently subjected to native PAGE and SDS–PAGE to identify slow migrating species that co-eluted three proteins of different molecular weights.

### Production of Fc and Fc fusions
Fc, sfGFP-Fc, mRuby2-Fc and CTX IgG were purchased from Genscript in CMVR vectors and transfected into in ExpiHEK293F cells (Thermo Fisher Scientific) and purified by IMAC with 50 mM Tris pH 8.0, 300 mM NaCl and 500 mM imidazole elution buffer, and were then purified further through SEC over a Superdex 200 10/300 GL FPLC column (Cytiva) into 50 mM Tris pH 8.0, 300 mM NaCl and 0.75% CHAPS (Fc) or 50 mM Tris, pH 8.0, 300 mM NaCl and 0.05% glycerol (Cetuximab IgG, sfGFP-Fc, mRuby2-Fc). Stocks were frozen at −80 °C for subsequent analyses.

### Large-scale expression and purification of O432-17 components
Designs that co-purified and yielded slowly migrating species, as determined by native PAGE, were subsequently subcloned into pET-29b+ vectors (Genscript). Each vector encoded either a trimeric plug or a tetramer variant—both with C-terminal hexahistidine tags—that was expressed at a larger scale (1–12 l of culture). Cells were lysed by microfluidization in 25 mM Tris pH 8.0, 300 mM NaCl, 1 mM DTT, 1 mM PMSF and 0.1 mg ml$^{-1}$ DNAse and cleared by centrifugation. Clarified lysates were filtered through 0.7-μm filters and purified by IMAC using gravity columns with nickel-NTA resin or HisTrap HP columns (Cytiva) using 25 mM Tris pH 8.0, 300 mM NaCl and 60 mM imidazole wash buffer and 25 mM Tris pH 8.0, 300 mM NaCl and 300 mM imidazole elution buffer. Elution fractions containing pure proteins were concentrated using centrifugal filter devices (Millipore) and further purified on a Superdex 200 10/300 GL (for large-scale purification) or Superose 6 10/300 GL (for comparison to assembly) gel-filtration column (Cytiva) using 25 mM Tris pH 8.0, 150 mM NaCl and 0.75% CHAPS. Gel filtration fractions containing pure protein in the desired oligomeric state were pooled, concentrated and frozen in aliquots at −80 °C for subsequent analyses.

### In vitro assembly of O432-17 nanoparticles
Purified tetramer and Fc or IgG from gel filtration fractions were assembled in a 1:1 molar ratio and with 1.1× excess trimeric plug. Molar ratios were calculated on the basis of the monomeric extinction coefficient. Assemblies were assembled in a total volume between 100 and 500 μl

and a total concentration of each monomeric component between 5 and 50 μM, dialyzed overnight at 25 °C in 25 mM Tris pH 8.0 and 150 mM NaCl, and purified on a Superose 6 10/300 GL gel filtration column (Cytiva) using 25 mM Tris pH 8.0 and 150 mM NaCl as the running buffer.

### Dynamic light scattering
DLS measurements were performed using the default sizing and polydispersity method on the UNcle (Unchained Labs). O432-17 variants (8.8 μl) were pipetted into glass cuvettes. DLS measurements were run in triplicate at 25 °C with an incubation time of 1 s; results were averaged across ten runs and plotted using Python3.7.

### Negative-stain electron microscopy preparation and data collection of O432-17 nanoparticles and variants
Between 0.1 and 0.2 mg ml$^{-1}$ of pre- or post-SEC O432-17 assemblies in 25 mM Tris pH 8.0 and 150 mM NaCl were applied onto 400- or 200-mesh carbon-coated copper grids and glow discharged for 20 s, followed by three applications of 3.04 μl 2% nano-W or Uranyless stain.

Micrographs were recorded using EPU software (Thermo Fisher Scientific) on a 120 kV Talos L120C transmission electron microscope (Thermo Fisher Scientific) at a nominal magnification of ×45,000 (pixel size, 3.156 Å per pixel) and a defocus range of 1.0–2.5 μm.

### Negative-stain electron microscopy data analysis of O432-17 nanoparticles
Negative-stain electron microscopy datasets were processed using Relion3.0 software[56]. Micrographs were imported into the Relion3.0 software, and a contrast transfer function was estimated using GCTF[57]. Around 500 particles were manually picked and 2D classified, and selected classes were used as templates for particle picking in all images. Approximately 100,000 picked particles were 2D classified for 25 iterations into 50 classes.

### Negative-stain electron microscopy data analysis of O432-17 nanoparticle variants
Negative-stain electron microscopy datasets were processed by CryoSPARC software v4.0.3 (ref. [58]). Micrographs were imported into the CryoSparc software. Particles were picked using the blob picker in CryoSPARC and 2D classified, or 2D classes from blob picking were used as templates for particle picking in all images. All the picked particles were 2D classified for 20 iterations into 100 classes.

### Cryo-electron microscopy preparation and data collection of O432-17 nanoparticles
Two microliters of pre-SEC-purified O432-17-Fc sample at 0.5 mg ml$^{-1}$ in 25 mM Tris pH 8.0 and 150 mM NaCl were applied onto C-flat 1.2/1.3 holey carbon grids. Grids were then plunge-frozen into liquid ethane and cooled with liquid nitrogen using a Thermo Fisher Vitrobot Mk IV, with a 6.5-s blotting time and blot force of 0. The blotting process took place inside the vitrobot chamber at 22 °C and 100% humidity. Data acquisition was performed with Leginon on a Titan Krios electron microscope operating at 300 kV using a K3 summit direct electron detector equipped with an energy filter and operating in super-resolution mode. The nominal magnification for data collection was ×105,000 with a calculated pixel size of 0.42 Å per pixel. The final dose was 63.775 e$^-$ per A$^2$ for 2,223 videos.

### Cryo-electron microscopy data analysis of O432-17 nanoparticles
All data processing was done using CryoSPARC v3.0.0 (ref. [58]). Alignment of video frames was performed using Patch Motion with an estimated $B$ factor of 500 Å$^2$, with the maximum alignment resolution set to 5. During alignment, all videos were Fourier cropped by one-half. Defocus and astigmatism values were estimated using patch CTF with default parameters. An initial 1,391 nanoparticle particles were

manually picked and extracted with a box size of 576 pixels. This was followed by a round of 2D classification and subsequent template-picking using the best 2D class averages low-pass filtered to 20 Å. Particles were next picked with Template Picker and were manually inspected before extraction with a box size of 660 pixels, and were then further Fourier cropped to a final box size of 330 pixels, for a total of 66,904 particles. A round of reference-free 2D classification was performed in CryoSPARC, with a maximum alignment resolution of 6 Å. The best classes that revealed visibly assembled nanoparticles were used for 3D ab initio determination using the $C_1$ symmetry operator. This was followed by a round of 3D heterogeneous refinement using $C_1$ symmetry and sorting into four distinct classes, all of which revealed complete plugging of the octahedral three-component nanoparticle. Thus, all 50,017 of the best particles selected from 2D classification were subjected to non-uniform 3D refinement with octahedral symmetry applied, yielding a final map with an estimated global resolution of 7.06 Å, following per-particle defocus refinement. The final maps were deposited in the EMDB under accession code EMD-29602. Refined models were generated by rigid-body docking followed by refinement of the backbone of the design model into the final cryo-EM density map in Rosetta[59].

## Computational design of O432-17 electrostatically charged variants

A consensus design approach was used to first identify interior surface positions predicted to be the most robust to surface substitutions. These positions were divided into three tiers on the basis of the predicted enhancement to the design's stability and/or solubility. Using the Rosetta modeling suite, the trimeric plug design model was redesigned, allowing optimization of the identities of interior surface residues that did not contribute to the interface between plug monomers or between the trimeric plug and antibody nanoparticle. We deployed three tiers of sequence-optimization strategies per charged variant. The first strategy enabled optimization of residue positions toward only charged amino acids (arginine or lysine for the positively charged variants, glutamate and aspartate for the negatively charged variants). The second strategy included the charged amino acids but also polar, non-charged amino acids, such as asparagine, glutamine and alanine, to maintain helical propensity. The third strategy included the residue identities in the first and second strategy, but also enabled reversion to the original identity in the trimeric plug design model. Substitutions that resulted in substantial losses of atomic packing interactions or side chain–side chain or side chain–backbone hydrogen bonds were discarded. The top-scoring designs for each design strategy and surface position tier were selected for inclusion as variant proteins.

## Expression and purification of electrostatically charged trimeric plug variants

Plasmids encoding electrostatically charged trimeric plug variants and containing a C-terminal hexahistidine tag were cloned into pET-29b+ vectors (Genscript) and expressed overnight at 37 °C in autoinduction medium[60]. Cells were lysed by microfluidization or sonication in 25 mM Tris pH 8.0, 500 mM NaCl, 1 mM DTT, 1 mM PMSF and 0.1 mg ml⁻¹ DNAse and cleared by centrifugation. Clarified lysates were filtered through 0.7-μm filters and purified by IMAC using gravity columns with nickel-NTA resin or HisTrap HP columns (Cytiva) using 25 mM Tris pH 8.0, 500 mM NaCl and 60 mM imidazole wash buffer and 25 mM Tris pH 8.0, 500 mM NaCl and 300 mM imidazole elution buffer. Elution fractions containing pure proteins were concentrated using centrifugal filter devices (Millipore) and further purified on a Superdex 200 10/300 GL (for large-scale purification) or Superose 6 10/300 GL (for comparison with assembly) gel-filtration column (Cytiva) using 25 mM Tris pH 8.0, 500 mM NaCl and 0.75% CHAPS. Gel filtration fractions containing pure protein in the desired oligomeric state were pooled, concentrated and frozen in aliquots at −80 °C for subsequent analyses.

## Expression and purification of pos36GFP

pos36GFP plasmid (Addgene no. 62937) was transformed into BL21(DE3) bacteria cells in lysogeny broth and induced with isopropyl ß-ᴅ-1-thiogalactopyranoside (IPTG) at an optical density of 0.6 and grown for 16 h at 20 °C. Cells were lysed by microfluidization or sonication in 25 mM Tris pH 8.0, 500 mM NaCl, 1 mM DTT, 1 mM PMSF and 0.1 mg ml⁻¹ DNAse and cleared by centrifugation. Clarified lysates were filtered through 0.7-μm filters and purified by IMAC using gravity columns with nickel-NTA resin or HisTrap HP columns (Cytiva) with 25 mM Tris pH 8.0, 500 mM NaCl and 60 mM imidazole wash buffer and 25 mM Tris pH 8.0, 500 mM NaCl and 300 mM imidazole elution buffer. Elution fractions containing pure proteins were concentrated using centrifugal filter devices (Millipore) and further purified on a Superdex 200 10/300 GL gel filtration columns (Cytiva) using 25 mM Tris pH 8.0, 500 mM NaCl and 0.75% CHAPS. Gel filtration fractions containing pure protein were pooled, concentrated and frozen in aliquots at −80 °C for subsequent analyses.

## In vitro packaging of pos36GFP

O432-17(−) nanoparticles with pos36GFP cargoes were assembled in the same molar ratio of purified 1× tetramer, 1× Fc and 1.1× negatively charged trimeric plug from gel filtration fractions with 1× purified pos36GFP. Mixtures were dialyzed at 25 °C for 16 h into assembly buffer containing 25 mM Tris pH 8.0 and 200 mM NaCl, or 25 mM Tris pH 8.0 and 1 M NaCl. The ability of nanoparticles to package pos36GFP cargo was assessed using SEC on a Superose 6 10/300 GL column (Cytiva) in either buffer. Packaged GFP was quantified using the absorbance measurements at 280 and 488 nm obtained using a NanoDrop 8000 spectrophotometer. The absorbance of pure pos36GFP at 280 and 488 nm in 200 mM NaCl was used to calculate the absorbance at 280 nm due to pos36GFP in gel filtration fractions containing pos36GFP packaged in O432-17(−) nanoparticles. The relative absorbance due to pos-36GFP and O432-17(−) nanoparticles was used to calculate the molar ratio of each protein using calculated extinction coefficients and was quantified against a standard curve generated using pure pos36GFP[36].

## In vitro packaging of pegRNA

O432-17(+) nucleocapsids with pegRNA (Integrated DNA Technologies) were assembled in a molar ratio of purified 1× tetramer, 1× CTX IgG and 1.1× positively charged trimeric plug from gel filtration fractions with 3× pegRNA per nucleocapsid. Mixtures were dialyzed at 25 °C for 16 h into 25 mM Tris pH 8.0 and 150 mM NaCl. Nucleocapsids were screened for in vitro encapsidation by native gel electrophoresis before and after a 1 h treatment with Benzonase or RNAse A.

## Gel electrophoresis

Native agarose gels were prepared using 0.8% ultrapure agarose (Invitrogen) in TAE buffer (Thermo Fisher Scientific) containing SYBR Gold (Invitrogen). Nine microliters of O432-17(+) nucleocapsids were treated with 1 μl Benzonase (Invitrogen, diluted to 10 units per μl) or 1 μl RNAse A (Thermo Fisher Scientific, diluted to 10 units per μl) at 25 °C for 60 min, followed by mixing with 2 μl 6× loading dye (New England Biosciences, no SDS), and electrophoresed at 120 V for 30 min[17,19]. RNA in gels was imaged, and gels were subsequently stained with Gelcode Blue to analyze protein (Thermo Fisher Scientific).

Protein SDS–PAGE gels were performed using anyKD polyacrylamide gels (Bio-Rad) in Tris-glycine buffer and were electrophoresed at 180 V for 120 min.

## In vitro assembly of O42.1 with pos36GFP

For each component, 1× O42.1 C4, 1× Fc and 1× pos36GFP per monomeric concentration were assembled and dialyzed for 16 h at 25 °C into 25 mM Tris pH 8.0 and 200 mM NaCl. The resulting assembly was screened using SEC on a Superose 6 10/300 GL column (Cytiva) in 25 mM Tris pH 8.0 and 200 mM NaCl.

## Yeast display of Myc peptide

EBY100 *S. cerevisiae* cells were grown at 30 °C for 16 h in C-Trp-Ura medium supplemented with 2% (w/v) glucose, and were allowed to grow at 30 °C for 16 h in SGCAA expression medium to display myc-3×YPG peptide, followed by two washes in PBS + 3% BSA before incubation with O432-17 nanoparticle variants targeting the Myc peptide.

## In vitro release of pos36GFP

O432-17(–) or O42.1 nanoparticles that had been in vitro assembled with pos36GFP were incubated with EBY100 *S. cerevisiae* cells expressing Myc-3×YPG peptide for 1 h at 25 °C. The cells were washed in 500 µl 25 mM Tris pH 8.0 and 500 mM NaCl until the supernatant fluorescence plateaued (after seven wash cycles) and were resuspended into citrate–phosphate buffer, pH 4.2, for 30 min at 25 °C. The supernatant was then collected and buffer exchanged using 25 mM Tris pH 8.0 and 500 mM NaCl before a fluorescence readout was generated using a Synergy Neo2 plate reader, with excitation wavelengths of 460 and 559 nm and emission wavelengths of 509 and 600 nm, and a gain of 120. Fluorescence was analyzed using GraphPad Prism version 9.3.1 for Windows, GraphPad Software (www.graphpad.com) and statistical significance was determined by two-way ANOVA with a 95% confidence interval and an alpha of 0.5.

## Rational design of O432-17 pH-responsive variants

Bulky hydrophobic residues such as isoleucine and leucine amino acid positions within the pH-responsive trimeric interface were selected for optimization to either alanine or valine using the Rosetta software suite. The point substitution with the best scoring interface energy (ddG) at each position was selected as a trimeric plug variant. Combinatorial pairs of each point substitution variant were included as trimeric plug variants. We generated four O432-17 pH-responsive variants for both O432-17(–) and O432-17(+), two containing a third histidine hydrogen-bond network and either the p.I57V or p.L75A point substitution (3HIS_I57V and 3HIS_L75A), one containing a third histidine hydrogen-bond network and both p.I57V and p.L75A point substitutions (3HIS_I57V_L75A) and one negative control containing zero histidine hydrogen-bond networks (0HIS), in which all histidines were mutated to asparagine.

## AF647 conjugation to O432-17 trimeric plug variants

Trimeric plug variants containing a p.T359C substitution were generated by site-directed mutagenesis PCR. Alexa Fluor 647 C2 maleimide (Thermo Fisher Scientific) dissolved in DMSO and trimeric plug variants containing 10× TCEP were incubated in a 5:1 molar ratio with respect to the trimeric plug monomer for 16 h (overnight) at 4 °C in PBS + 0.75% CHAPS titrated to pH 7.2. The final reaction mixture contained 75 µM Alexa Fluor, 15 µM trimeric plug and 150 µM TCEP. The maleimide reaction was quenched with 1 mM DTT and buffer exchanged into 25 mM Tris pH 8.0 and 150 mM NaCl using PD-10 desalting columns with Sephadex-25 resin (Cytiva), according to the manufacturer's protocol. It was then dialyzed for 3–4 days in 25 mM Tris pH 8.0, 150 mM NaCl and 0.75% CHAPS at 4 °C. Positively charged trimeric plug variants were buffer exchanged and dialyzed into 25 mM Tris pH 8.0, 500 mM NaCl and 0.75% CHAPS. The degree of labeling was estimated from a fluorophore extinction coefficient of $265,000 \, M^{-1} cm^{-1}$, and absorbance measurements were taken at 280 and 650 nm using a NanoDrop 8000 spectrophotometer.

## Flow-cytometry-based pH titration

Linear Myc peptide with an N-terminal lysine side chain and 3× glycine linker (KGGGEQKLISEEDL) was produced through solid-phase peptide synthesis and biotinylated through amide formation. The resulting biotinylated Myc peptide was purified by reversed-phase high-performance liquid chromatography and quality checked for the proper molecular weight using liquid chromatography–mass spectrometry, lyophilized and dissolved in 100% DMSO for long-term storage. Then, 3.0–3.4 µm streptavidin-coated polystyrene particles (Spherotech) were incubated with biotinylated Myc peptide diluted in PBS to 5% DMSO 1 h at 25 °C. The antibody to Myc was purchased from Cell Signaling Technologies (cat. no. 9B11). The coated particles were washed in PBS + 3% BSA twice and split equally into pH-titrated citrate–phosphate buffer for 30 min at 25 °C. Coated particles were washed twice with PBS + 3% BSA and resuspended for flow cytometry. For O432-17(+) experiments, we prevented non-specific association with the polystyrene particles by using EBY100 *S. cerevisiae* cells.

All flow-cytometry experiments were performed on a LSR II Flow Cytometer (BD Biosciences). Lasers calibrated with coated particles were stained with either FITC-labeled anti-Myc-tag antibody (Abcam cat. no. 9E10), APC-labeled anti-Myc-tag antibody (Abcam cat. no. 9E10) or no antibody. Ten thousand events were collected per sample on three biological replicates. All flow-cytometry results were analyzed using FlowJo. Singlet beads were first isolated before analysis for AF647 or sfGFP signal. Normalization to the minimum and maximum fluorescence of each channel with each titration sample was performed in Python3.7, and the apparent $pK_a$ of AF647 and sfGFP fluorescence was estimated with a four-parameter non-linear logistic regression fit in GraphPad Prism version 9.3.1 for Windows (GraphPad Software).

## Cells

Wild-type HeLa (ATCC CCL-2), EGFR-knockout HeLa (Abcam ab255385) and A431 cells (ATCC CRL-1555) were cultured at 37 °C with 5% $CO_2$ in flasks with Dulbecco's modified Eagle medium (DMEM) (Gibco) supplemented with 1 mM L-glutamine (Gibco), 4.5 g l$^{-1}$ D-glucose (Gibco), 10% fetal bovine serum (FBS) (Hyclone) and 1% penicillin–streptomycin (PenStrep) (Gibco). Wild-type HeLa and EGFR-knockout HeLa cells were also cultured with 1× nonessential amino acids (Gibco) supplemented in the medium. Cells were passaged twice per week. For passaging, cells were dissociated using 0.05% trypsin EDTA (Gibco) and split 1:5 or 1:10 into a new tissue-culture-treated T75 flask (Thermo Fisher Scientific cat. no. 156499). For serum starvation, cells were cultured in their respective cell medium without FBS for 16 h.

## Immunostaining

For immunostaining, 35-mm glass-bottom dishes were seeded at a density of 20,000 cells per dish. A final monomeric concentration of 10 nM of O432-17-CTX or O432-17-Fc nanoparticles (417 pM final nanoparticle concentration) were incubated with cultured cells in DMEM or DMEM supplemented with 10% FBS. Cells were fixed in 4% paraformaldehyde, permeabilized with 100% methanol and blocked with PBS + 1% BSA. Cells were immunostained with anti-LAMP2A antibody (Abcam cat. no. ab18528) and/or anti-NaK ATPase (Abcam cat. no. ab76020, Abcam cat. no. ab283318), followed by Alexa Fluor 488-conjugated goat anti-rabbit-IgG secondary antibody (Thermo Fisher Scientific cat. no. A-11034) or Alexa Fluor 488-conjugated goat anti-mouse-IgG secondary antibody (Thermo Fisher Scientific cat. no. A-11029) and DAPI (Thermo Fisher Scientific cat. no. D1306), and were stored in the dark at 4 °C until imaging.

## Nanoparticle uptake epifluorescence microscopy (A431 cells)

Cells were washed twice with FluoroBrite DMEM imaging medium and subsequently imaged in the same medium in the dark at room temperature. Epifluorescence imaging was performed on a Yokogawa CSU-X1 spinning dish confocal microscope with either a Lumencor Celesta light engine with seven laser lines (408, 445, 473, 518, 545, 635 and 750 nm) or a Nikon LUN-F XL laser launch with four solid state lasers (405, 488, 561 and 640 nm). A 40× objective with a numerical aperture (NA) of 0.95 and a Hamamatsu ORCA-Fusion scientific CMOS camera, both controlled by NIS Elements software (Nikon), were used. The following excitation (EX)/emission (EM) filter combinations (center/bandwidth in nm) were used: BFP, EX408 EM443/38; GFP, EX473 EM525/36; RFP,

EX545 EM605/52; far red, EX635 EM705/72. The exposure times were 100 ms for the acceptor direct channel and 500 ms for all other channels, with no EM gain set and no neutral density filter added.

## Nanoparticle uptake image acquisition in wild-type and EGFR-knockout HeLa cells

Four-color, 3D images were acquired with a commercial OMX-SR system (GE Healthcare). Toptica diode lasers with excitations of 405 nm, 488 nm and 640 nm were used. Emission was collected on three separate PCO.edge sCMOS cameras using an Olympus 60× 1.420-NA Plan-Apochromat oil immersion lens; 1024 × 1024 images (pixel size, 6.5 μm) were captured with no binning. Acquisition was controlled with AcquireSR Acquisition control software. Z-stacks were collected with a step size of 125 nm. Images were deconvolved in SoftWoRx 7.0.0 (GE Healthcare) using the enhanced ratio method and 200 nm noise filtering. Images from different color channels were registered in SoftWoRx using parameters generated from a gold-grid registration slide (GE Healthcare).

## Nanoparticle uptake quantification

To quantify the percentage of cells that were bound to nanocages, the Multi-point tool in Fiji (ImageJ)[61] was used to count nuclei in each image. The mRuby2 channel was used to identify and count cells, again using the multi-point tool, showing nanoparticle localization above the background signal using the identified nuclei as a guide for cell position. For cells positive for the LAMP2A nanoparticle signal, staining was used to aid in determining cell shape, and the freehand selection tool was used to draw manual regions of interest around the cell circumference. All regions of interest for positive cells were measured for cell area and integrated intensity in the mRuby2 channel for integrated intensity–cell area quantification. Cellular data were plotted and statistical tests were done in GraphPad Prism version 9.3.1 for Windows, GraphPad Software.

## Statistics and reproducibility

Details on electron microscopy experiments can be found in Supplementary Table 4. All gel electrophoresis experiments were replicated in at least two biologically independent experiments.

## Reporting summary

Further information on research design is available in the Nature Portfolio Reporting Summary linked to this article.

## Data availability

Source data for all images and data generated and analyzed by the authors are provided with this paper. Density maps have been deposited in the Electron Microscopy Data Bank under the accession number EMD-29602. Source data are provided with this paper.

## Code availability

Source code for the fusion, docking and design of non-porous pH-responsive antibody nanoparticles is available at https://github.com/erincyang/plug_design. The protocol requires compilation of the worms and rpxdock repositories, which have been made available at https://github.com/willsheffler/worms (ref. 29) and https://github.com/willsheffler/rpxdock (ref. 31), respectively. Source code for generation of the figures in this manuscript was written by the authors and is provided in the Supplementary Information.

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

## Acknowledgements

We thank J. Lazarovitz, W. L. White, S. Boyken, C. Richardson, D. Chmielewski, I. Yousif, H. Choi, S. J. Hendel, Y. Hsia, M. Wu, G. Reggiano, A. Olshefsky, K. Van Wormer, R. Krishnamurty, K. Herrera, L. Goldschmidt and L. Stewart for helpful discussions and general tasks related to the study, L. Carter and M. Miranda for help with SEC-MALS, M. Murphy and C. Ogohara for assistance with mammalian proteins and H. Bai and K. Wu for providing EBY100 cells displaying Myc-3×YPG. This research was funded by the NSF Grant CHE-1629214 (N.P.K. and D.B.), Defense Threat Reduction Agency Grants HDTRA1-18-1-0001 and HDTRA1-19-1-0003 (N.P.K. and D.B.), the grant DE-SC0018940 funded by the U.S. Department of Energy, Office of Science (D.B.), National Institutes of Health's National Institute on Aging grants R01AG063845 and 1R01CA240339 (N.P.K. and D.B.), the Bill and Melinda Gates Foundation no. INV-010680 (N.P.K. and D.B.), the Audacious Project at the Institute for Protein Design, Washington Research Foundation and Translational Research Fund, National Science Foundation Graduate Research Fellowship program under the grant number DGE-1762114 (E.C.Y.), the Helen Hay Whitney Foundation (J.Z.Z.), the DAAD PROMOS program (N.G.) and the Howard Hughes Medical Institute (D.B.).

## Author contributions

E.C.Y., R.D., N.P.K. and D.B. designed the research. E.C.Y., R.D., M.C.M, A.J.B., W.S., J.Z.Z., J.D., A.S., M. Abedi, N.G., M. AAhlrichs, C.D., A.H., S. Cheng, M.L., P.M.L., S. Chan, R.S., J.F., G.U., J.L., M.S. and A.K acquired, analyzed or interpreted the data. E.C.Y., W.S. and D.B. created software for the research. E.C.Y., N.P.K. and D.B. wrote the paper.

## Competing interests

A provisional patent application has been filed (63/493,252) on the plugged antibody nanoparticle sequences by the University of Washington, listing D.B., E.C.Y., N.P.K., R.D., J.L., W.S., G.U., and J.F. as inventors. The other authors declare no competing interests.

## Additional information

**Extended data** is available for this paper at https://doi.org/10.1038/s41594-024-01288-5.

**Correspondence and requests for materials** should be addressed to Neil P. King or David Baker.

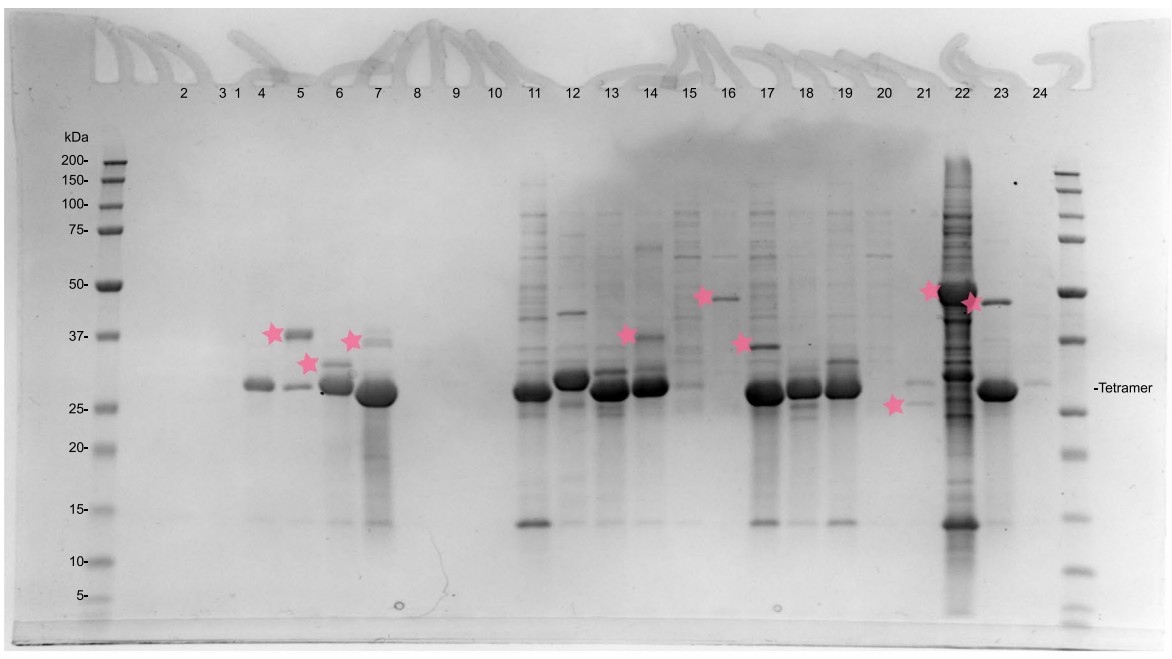

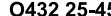

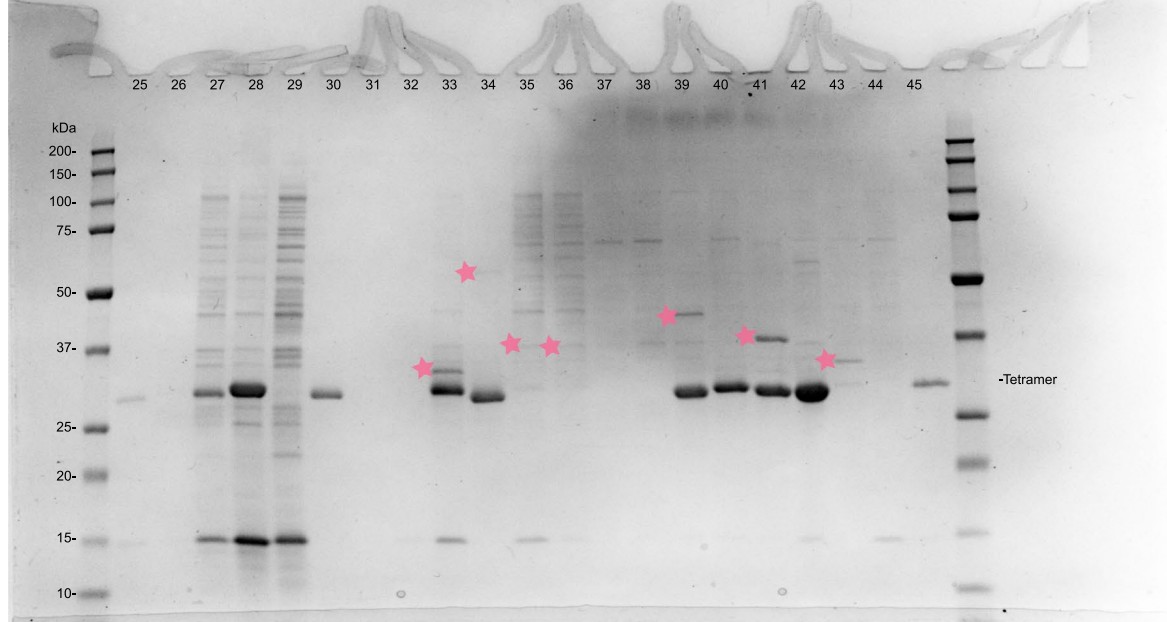

**Extended Data Fig. 1 | Reducing SDS–PAGE screening gels of co-expressed trimer-tetramer variants 1–45.** Tetramer protein band is marked (right), designs where the designed trimers and tetramers co-eluted are marked with pink stars next to the trimer band. Gel samples were derived from the same experiment and electrophoresis was processed in parallel.

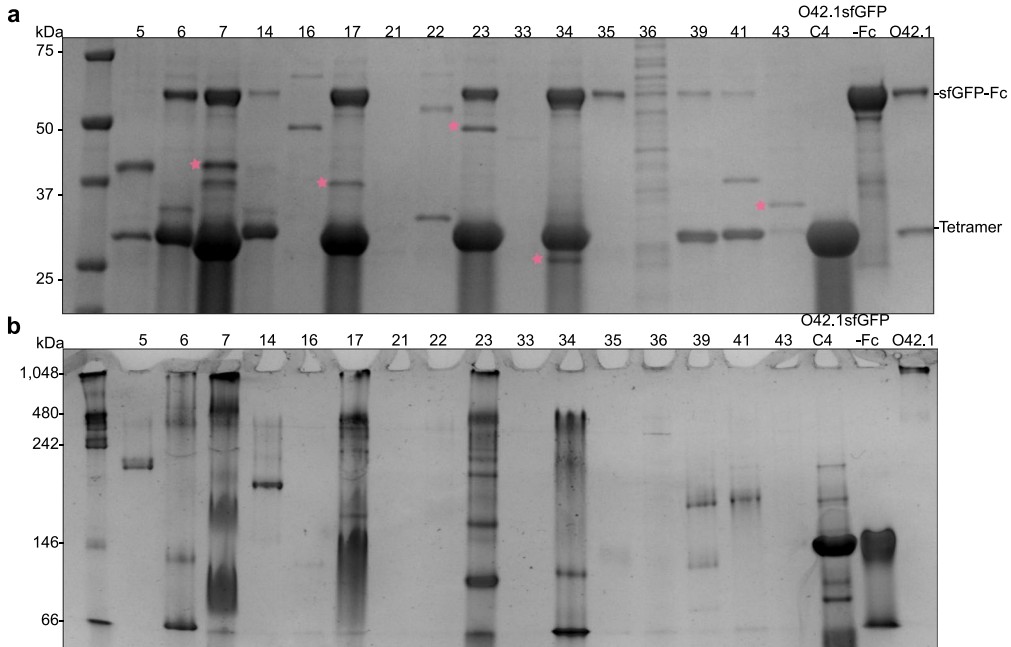

**Extended Data Fig. 2 | Example experimental screen of O432 three–component nanoparticles. a**, Clarified lysates of 16 designs where the designed trimer co-eluted with the tetramer were supplemented with a purified sfGFP-Fc fusion protein, purified by IMAC, and subject to reducing SDS-PAGE. Trimer protein bands selected for subsequent characterization are marked with pink stars. **b**, Non-denaturing native PAGE gel electrophoresis of clarified lysates from 16 designs, supplemented with purified sfGFP-Fc fusion protein and purified by IMAC. All designs were compared against the previously designed, two-component antibody nanoparticle (O42.1), O42.1 tetrameric component (O42.1 C4) and sfGFP-Fc (Divine et al.[26]). Gel samples were derived from the same experiment and electrophoresis was processed in parallel.

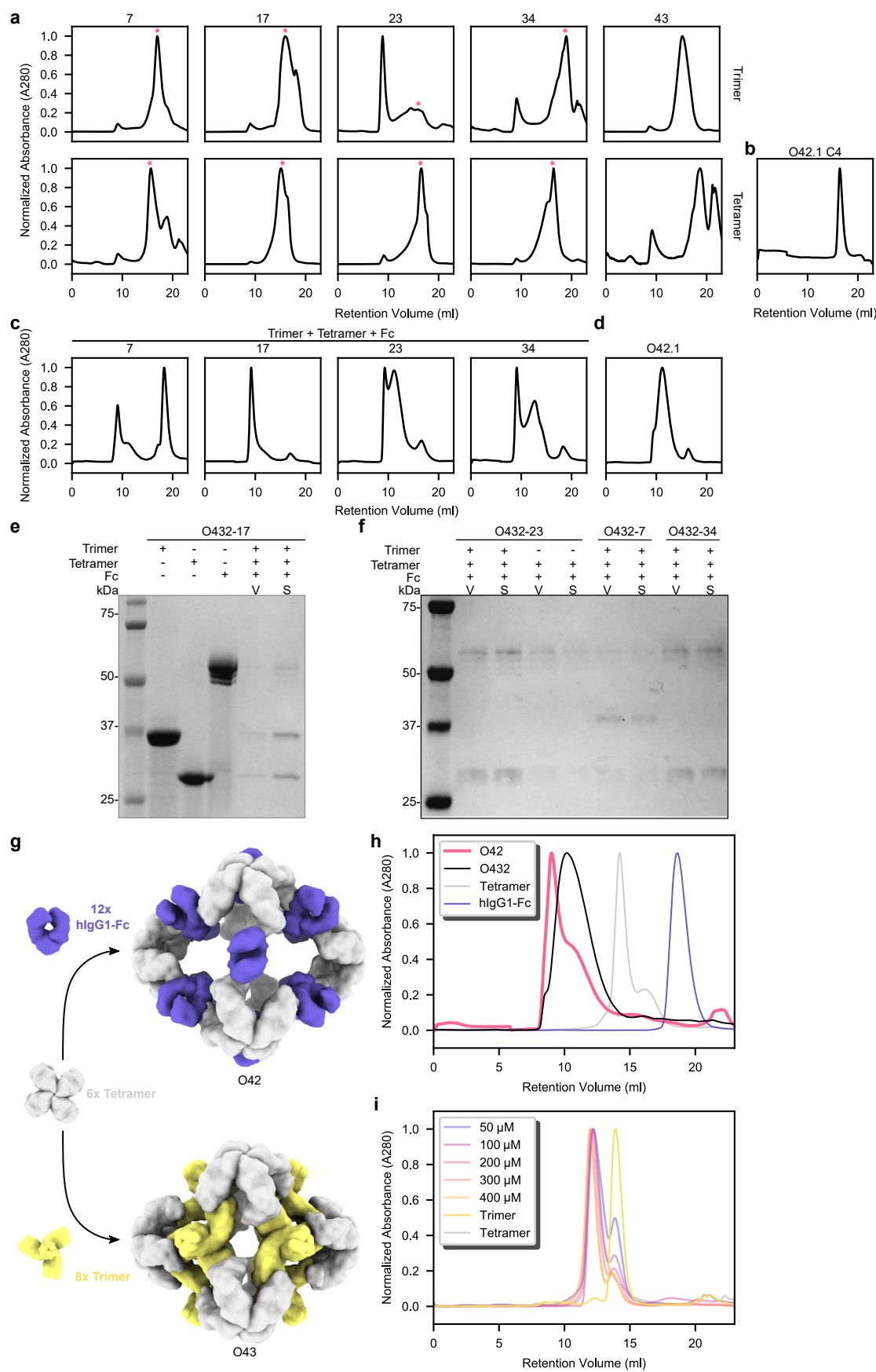

**Extended Data Fig. 3 | See next page for caption.**

**Extended Data Fig. 3 | Large-scale assembly competency screen of O432 three-component nanoparticles. a**, Subcloned components from putative three-component assemblies were purified separately. Peaks marked with a pink star were collected for stoichiometric *in vitro* assembly of three-component O432 nanoparticles. **b**, Elution profiles of tetrameric components were compared to the original O42.1 tetramer (O42.1 C4). **c**, Material permitting, stoichiometric *in vitro* assembly of trimer, tetramer, and Fc were purified by SEC, **d**, Elution profiles of resulting assemblies were compared to the SEC elution profile of previously designed O42.1 (Divine et al.[26]). **e**, Non-reducing SDS-PAGE of the void (V) and shoulder (S) peaks from SEC of O432-17 with purified trimer, tetramer, and Fc components as controls. **f**, Non-reducing SDS-PAGE of the void (V) and shoulder (S) peaks from SEC purification of O432-23, O432-7, and O432-34 *in vitro*

assemblies. **g**, 6× designed tetramers form protein-protein interfaces with both 12× Fc and 8× designed trimers. A schematic depicts a hypothetical nanoparticle assembly from the designed tetramer with only the Fc (top) or only the trimer (bottom). **h**, Representative SEC traces on the Superose 6 10/300 GL of assembly reactions containing only the designed tetramer and Fc (pink), compared to the full three-component assembly (black) and the individual components (gray and purple). **i**, Representative SEC traces on the Superdex 200 10/300 GL of assembly reactions containing only the designed tetramer and trimer at concentrations of 50 µM, 100 µM, 200 µM, 300 µM, and 400 µM, compared to the individual components (gray and yellow). UCSF ChimeraX 1.6 and the PyMOL Molecular Graphics System, version 2.5 (Schrödinger) was used to create **g**.

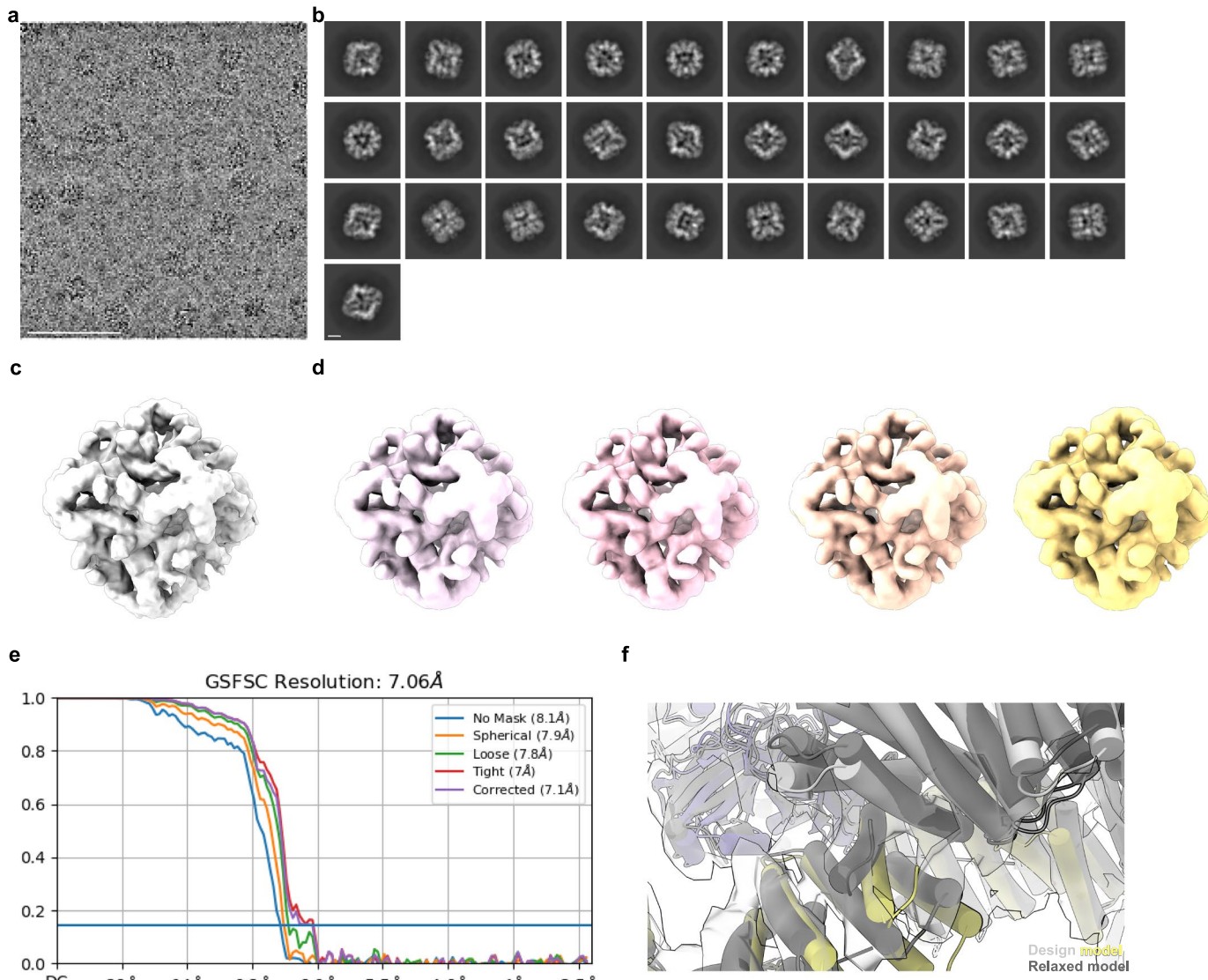

**Extended Data Fig. 4 | Cryo-EM processing shows newly designed trimeric plug occupying all 3-fold symmetry axes of the nanoparticle octahedral architecture. a**, Representative micrograph of cryo-EM sample. Scale bar, 100 nm. **b**, Reference-free two-dimensional class averages. Scale bar, 10 nm. **c**, *Ab initio* three-dimensional reconstruction without applied octahedral symmetry. **d**, 3D reconstructions generated following a heterogeneous refinement in the absence of applied octahedral symmetry. All four classes show trimeric plugs occupying all facets of the designed nanoparticle. **e**, Gold-standard Fourier shell correlation curves for the O432-Fc EM density map with octahedral symmetry applied. **f**, Close up of plug interface between design model (light gray and yellow) and model built from the 3D reconstruction (dark gray). UCSF ChimeraX 1.6 and the PyMOL Molecular Graphics System, version 2.5 (Schrödinger) was used to create **f**.

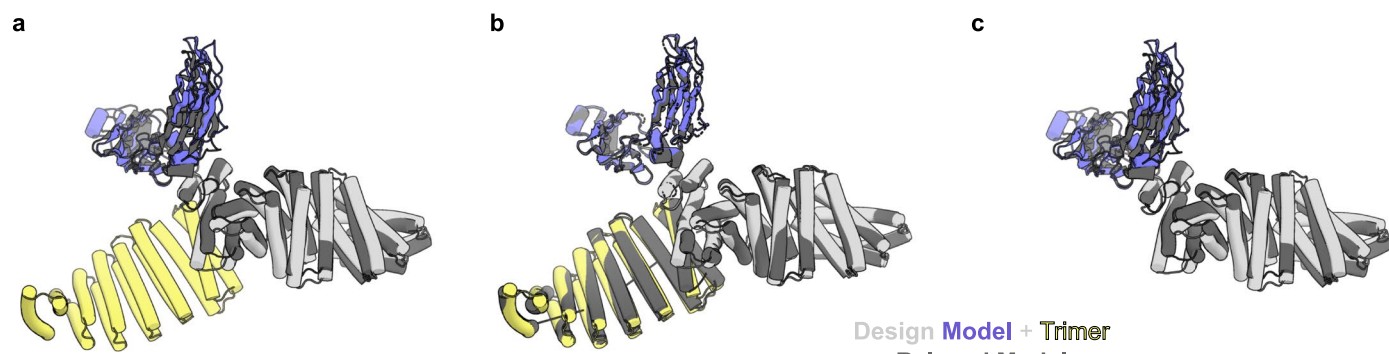

**Design** **Model** + **Trimer**
**Relaxed Model**

**Extended Data Fig. 5 | Comparison of design models of O42.1 and O432-17 with models fit into Cryo-EM density. a**, The three-component O432-17 design model (light gray, purple, and yellow) overlaid on the two-component O42.1 density-refined model (dark gray) shows an RMSD of 1.9 Å. **b**, The O432-17 design model (light gray, purple, and yellow) deviates from its density-refined model (dark gray) by 1.6 Å. **c**, The O42.1 design model deviates from its density-refined model (dark gray) by 4.2 Å. The PyMOL Molecular Graphics System, version 2.5 (Schrödinger) was used to create **a-c**.

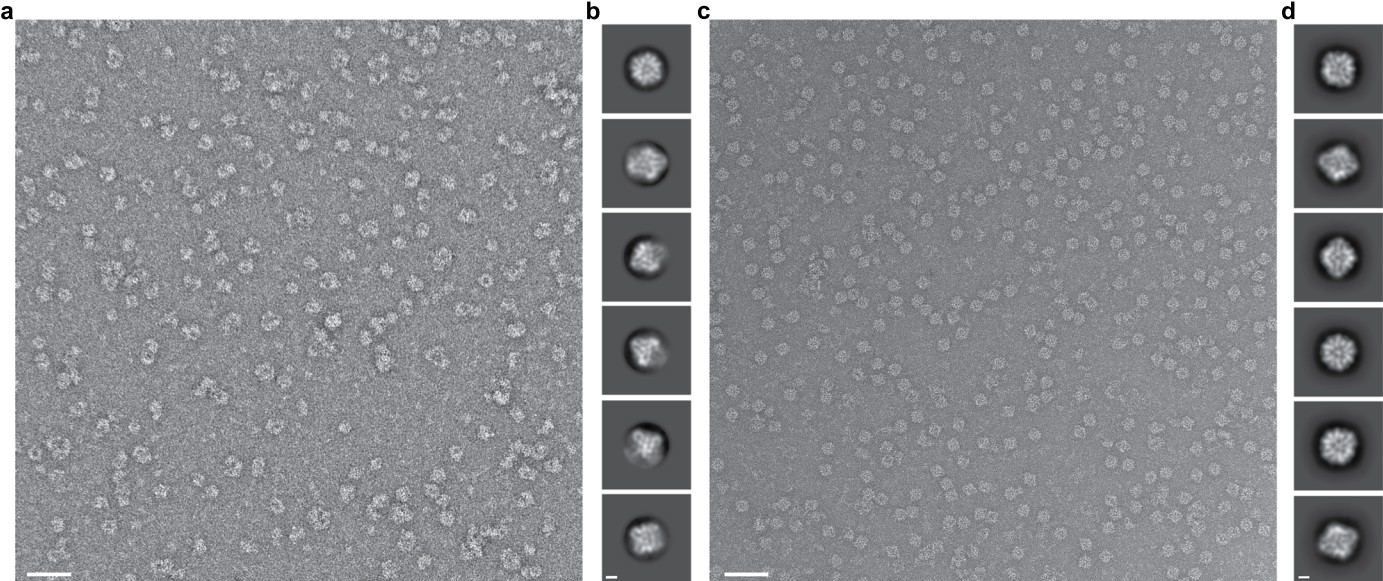

**Extended Data Fig. 6 | O432-17 nanoparticles form assemblies in the presence of protein and nucleic acid cargo. a**, Representative negative stain electron micrograph of O432-17(+) in the presence of RNA. Scale bar, 100 nm. **b**, Reference-free two-dimensional class averages showing multiple views of O432-17(+) nanoparticles. Scale bar, 10 nm. **c**, Representative negative-stain electron micrograph of O432-17(-) in the presence of pos36GFP. Scale bar, 100 nm. **d**, Reference-free two-dimensional class averages showing multiple views of O432-17(-) nanoparticles. Scale bar, 10 nm.

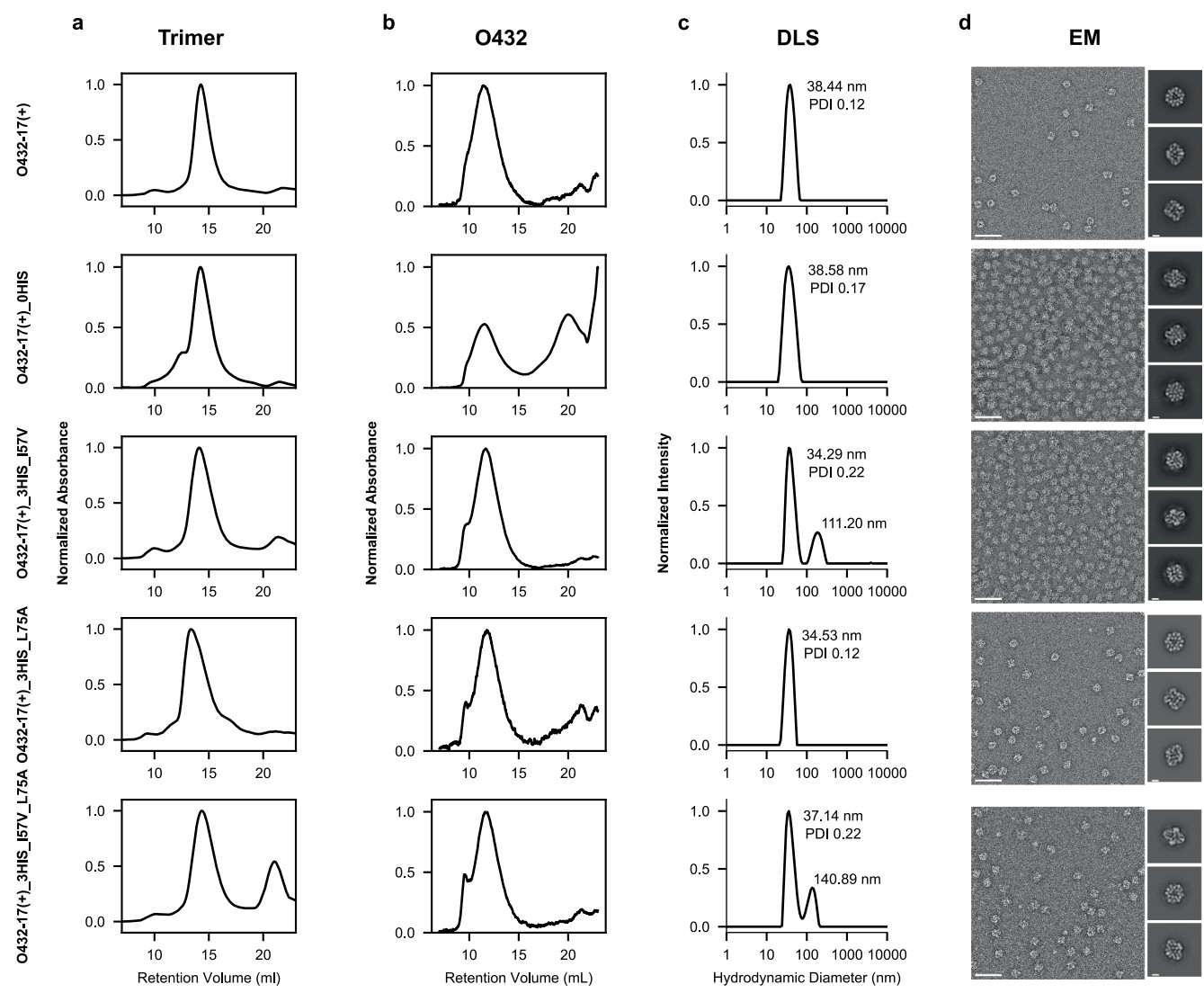

**Extended Data Fig. 7 | Assembly competency of O432-17(+) trimer variants.**
**a**, SEC elution profile of the trimer component on a Superdex 200 GL 10/300
chromatography column. **b**, SEC elution profile of the full O432-17 assembly
containing each trimer variant assembled with tetramer and Fc and purified on
a Superose 6 GL 10/300 chromatography column. **c**, DLS profile of the resulting
assembly as normalized intensity. **d**, Representative nsEM and reference-free
two-dimensional class averages of each assembly variant. Scale bar 100 nm and
10 nm, respectively.

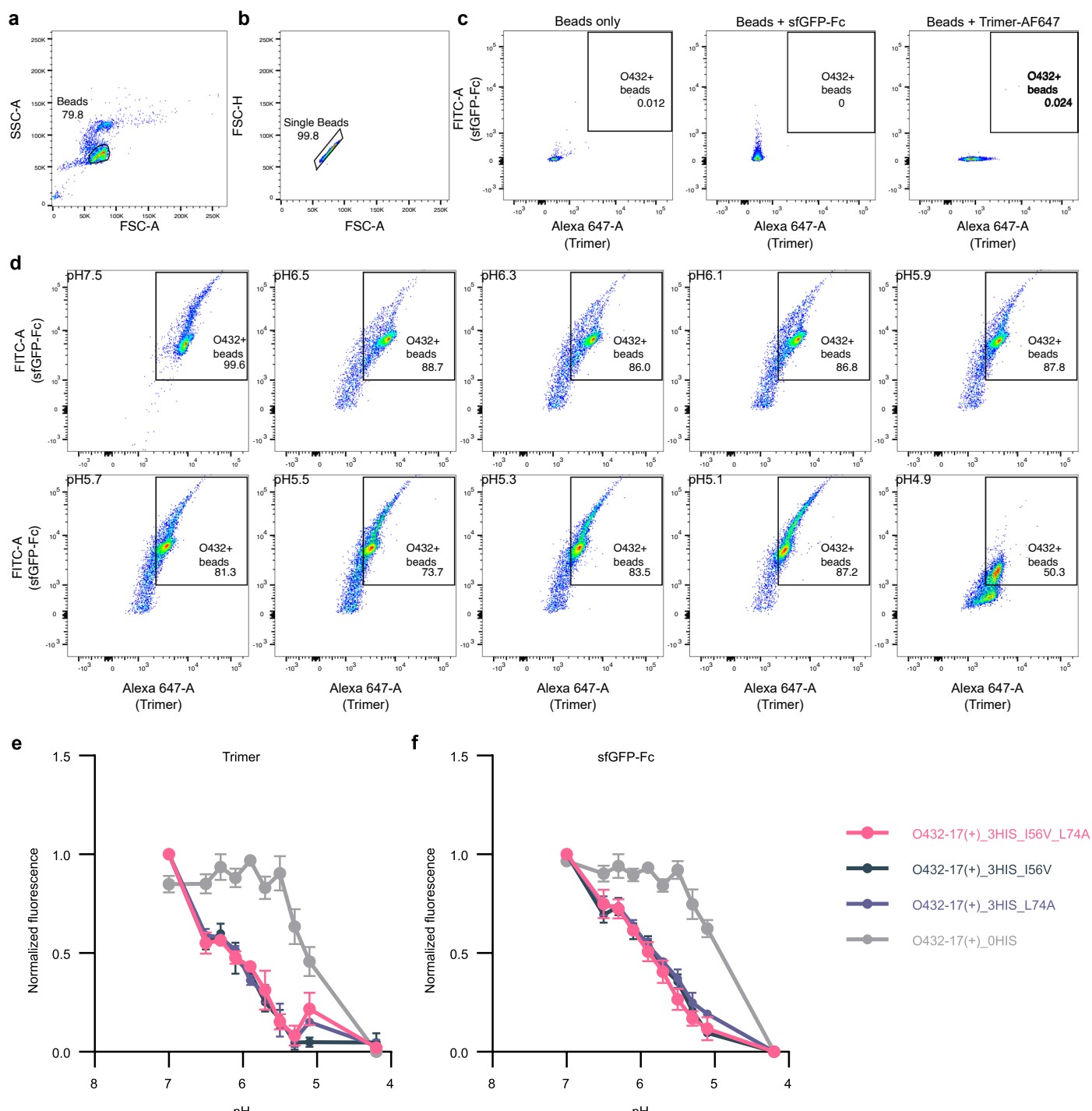

**Extended Data Fig. 8 | Flow cytometry analysis of O432-17 nanoparticle variants. a**, Forward and side scattering intensity isolated all beads. **b**, Isolated beads were selected for singlet scattering. **c**, Singlet beads were gated for both AF647 and sfGFP-Fc signal with reference to negative controls: beads only (left), beads incubated with sfGFP-Fc (middle), and beads incubated with AF647-labeled trimeric plug (right). **d**, The mean fluorescence intensity of beads positive for AF647-labeled trimer and sfGFP-Fc signal was taken at each pH. This gating strategy was applied across all trimeric plug variants: 0HIS, 2HIS, 3HIS_I57V, 3HIS_L75A, 3HIS_I57V_L75A. **e-f**, Mean fluorescence intensity of O432-17(+)

nanoparticles was measured as a function of pH for the trimeric plug variants and sfGFP-Fc. Histidine-containing O432-17(+) variants: O432-17(+)_3HIS_I57V, O432-17(+)_3HIS_L75A, O432-17(+)_3HIS_I57V_L75A; negative control: O432-17(+)_0HIS. Apparent pKas for O432-17(+)_3HIS_I57V, O432-17(+)_3HIS_L75A, and O432-17(+)_3HIS_I57V_L75A: pH 6.1 (AF647); pH 5.8 (sfGFP). O432-17(+)_0HIS apparent pKa: pH 5.1 (AF647); pH 5.0 (sfGFP). Data are presented as mean values +/- SEM over 3 biologically independent replicates. FlowJo v10.8.1(BD Biosciences) was used to create **a, b, c**, and **d**. GraphPad Prism version 9.3.1 (GraphPad Software) was used to create **e** and **f**.

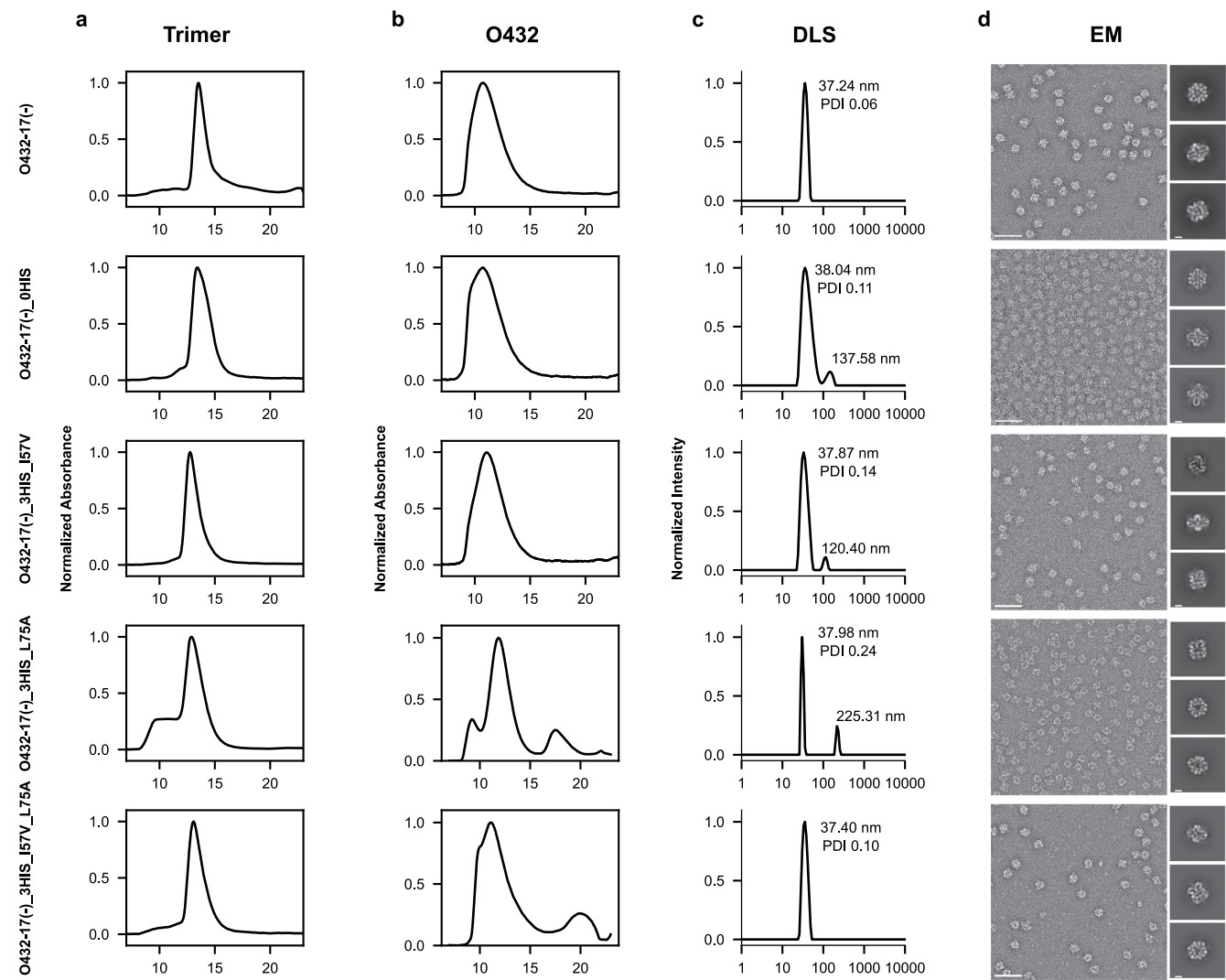

**Extended Data Fig. 9 | Assembly competency of O432-17(-) trimer variants.**
**a**, SEC elution profile of the trimer component on a Superdex 200 GL 10/300 chromatography column. **b**, SEC elution profile of the full O432-17 assembly containing each trimer variant assembled with tetramer and Fc and purified on a Superose 6 GL 10/300 chromatography column. **c**, DLS profile of the resulting assembly as normalized intensity. **d**, Representative nsEM and reference-free two-dimensional class averages of each assembly variant. Scale bar 100 nm and 10 nm, respectively.

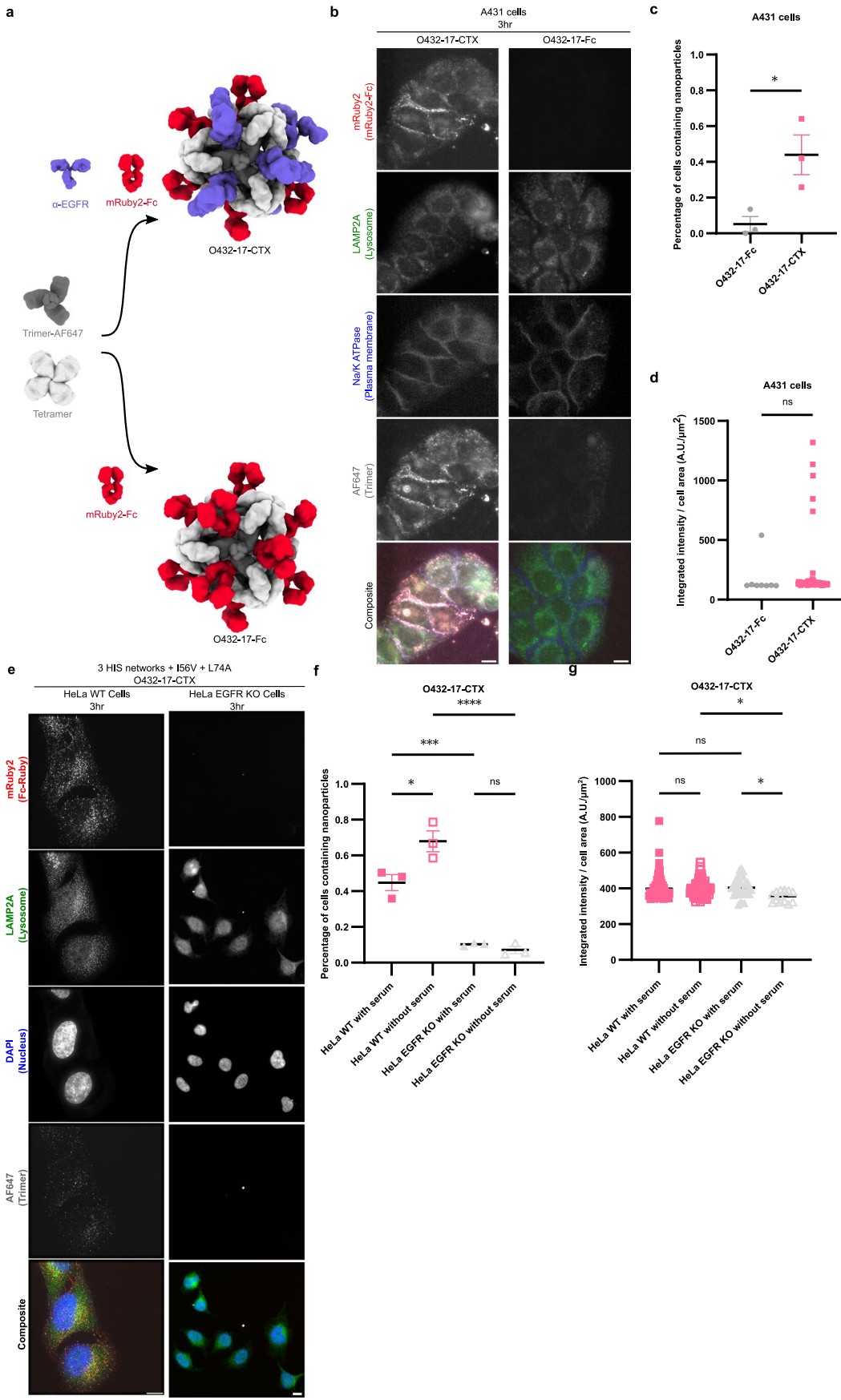

**Extended Data Fig. 10 | See next page for caption.**

**Extended Data Fig. 10 | Quantification of targeted receptor-mediated uptake of O432 nanoparticles. a**, Targeted and non-targeted variants of O432-17(-) nanoparticles were assembled *in vitro* with AF647-conjugated trimeric plug, designed tetramer, and either premixed mRuby2-Fc and α-EGFR mAb (top) or mRuby2-Fc alone (bottom). **b**, Cellular uptake of O432-17-CTX and O432-17-Fc nanoparticles was measured by the AF647 and mRuby2 fluorescence within the cell area after 3 hours of incubation in A431 cells. Single confocal plane images; grayscale panels for lysosomal membranes (green), mRuby2-Fc (red), and Trimer-AF647 (gray). Scale bar, 10 µm. **c**, Percentage of A431 cells containing nanoparticles. Statistics: *P = 0.0306, two-tailed unpaired t-test. Data are presented as mean values +/- SEM over 3 images per condition. **d**, Integrated intensity per cell area among A431 cells containing nanoparticles. Statistics: P = 0.6025, ns = not significant, two-tailed, unpaired T-test. Data are presented as mean values +/- SEM across all cells in 3 images per condition. **e**, Targeted receptor-mediated uptake of O432-17-CTX nanoparticles in WT and EGFR KO HeLa cells after 3 hours. Scale bar, 10 µm. **f**, Percentage of HeLa WT and HeLa EGFR KO cells containing O432-17-CTX nanoparticles with and without serum. Statistics: *P = 0.0116, ***P = 0.001, ****P < 0.0001, ns (P = 0.9349) = not significant, One-way ANOVA with Tukey's correction for multiple comparisons. Data are presented as mean values +/- SEM over 3 images per condition. **g**, Integrated intensity per cell area among HeLa WT and HeLa EGFR KO cells containing O432-17-CTX nanoparticles with and without serum. Statistics: *P = 0.0359 (HeLa WT without serum vs. HeLa EGFR KO without serum), *P = 0.0161 (HeLa EGFR KO with serum vs. HeLa EGFR KO without serum), P = 0.9598 (HeLa WT with serum vs. HeLa WT without serum), P = 0.9298 (HeLa WT with serum vs. HeLa EGFR KO with serum), One-way ANOVA with Tukey's correction for multiple comparisons. Data are presented as mean values +/- SEM across all cells in 3 images per condition; ImageJ v2 was used to create **b,e**; GraphPad Prism version 9.3.1 (GraphPad Software) was used to create **c-d, f-g**.

# Reporting Summary

## Statistics

For all statistical analyses, confirm that the following items are present in the figure legend, table legend, main text, or Methods section.

| n/a | Confirmed | |
|---|---|---|
| ☐ | ☒ | The exact sample size (*n*) for each experimental group/condition, given as a discrete number and unit of measurement |
| ☐ | ☒ | A statement on whether measurements were taken from distinct samples or whether the same sample was measured repeatedly |
| ☐ | ☒ | The statistical test(s) used AND whether they are one- or two-sided<br>*Only common tests should be described solely by name; describe more complex techniques in the Methods section.* |
| ☐ | ☒ | A description of all covariates tested |
| ☐ | ☒ | A description of any assumptions or corrections, such as tests of normality and adjustment for multiple comparisons |
| ☐ | ☒ | A full description of the statistical parameters including central tendency (e.g. means) or other basic estimates (e.g. regression coefficient) AND variation (e.g. standard deviation) or associated estimates of uncertainty (e.g. confidence intervals) |
| ☐ | ☒ | For null hypothesis testing, the test statistic (e.g. *F*, *t*, *r*) with confidence intervals, effect sizes, degrees of freedom and *P* value noted<br>*Give P values as exact values whenever suitable.* |
| ☒ | ☐ | For Bayesian analysis, information on the choice of priors and Markov chain Monte Carlo settings |
| ☒ | ☐ | For hierarchical and complex designs, identification of the appropriate level for tests and full reporting of outcomes |
| ☒ | ☐ | Estimates of effect sizes (e.g. Cohen's *d*, Pearson's *r*), indicating how they were calculated |

*Our web collection on statistics for biologists contains articles on many of the points above.*

## Software and code

Policy information about availability of computer code

| Data collection | Both the worms (https://github.com/willsheffler/worms) and RPXDock (https://github.com/willsheffler/rpxdock) software codes are available publicly on GitHub. Code was developed as part of this project and provided in separate publications; there are no version accession numbers for these repositories. An example design protocol used to generate O432-17 nanoparticles is provided in the following github link: https://github.com/erincyang/plug_design<br>Electron microscopy processing was done using Relion 3.1 or CryoSPARC4.2. Particles for Fig 1D (top) were picked with CisTEM and imported into Relion for subsequent 2D averaging<br>Images were created in UCSF ChimeraX 1.6 and the PyMOL Molecular Graphics System, version 2.5 (Schrödinger), and Biorender |
|---|---|
| Data analysis | Flow cytometry analyses were performed in Flowjo, LLC softwarev10.10<br>Statistical analyses were performed in GraphPad Prism v9.3.1<br>Cell images were analyzed in ImageJ v2.<br>Electron microscopy analysis was performed by CryoSPARC software v4.0.3 or Relion3.0 software |

For manuscripts utilizing custom algorithms or software that are central to the research but not yet described in published literature, software must be made available to editors and reviewers. We strongly encourage code deposition in a community repository (e.g. GitHub). See the Nature Portfolio guidelines for submitting code & software for further information.

## Data

Policy information about [availability of data](availability of data)

All manuscripts must include a [data availability statement](data availability statement). This statement should provide the following information, where applicable:

- Accession codes, unique identifiers, or web links for publicly available datasets
- A description of any restrictions on data availability
- For clinical datasets or third party data, please ensure that the statement adheres to our [policy](policy)

> Source data for all images and data generated and analyzed by the authors are provided with this manuscript. Density maps have been deposited in the Electron Microscopy Data Bank under the accession number EMD-29602.

## Human research participants

Policy information about [studies involving human research participants and Sex and Gender in Research.](studies involving human research participants and Sex and Gender in Research.)

| | |
|---|---|
| Reporting on sex and gender | N/A |
| Population characteristics | N/A |
| Recruitment | N/A |
| Ethics oversight | N/A |

Note that full information on the approval of the study protocol must also be provided in the manuscript.

# Field-specific reporting

Please select the one below that is the best fit for your research. If you are not sure, read the appropriate sections before making your selection.

☒ Life sciences        ☐ Behavioural & social sciences        ☐ Ecological, evolutionary & environmental sciences

For a reference copy of the document with all sections, see [nature.com/documents/nr-reporting-summary-flat.pdf](nature.com/documents/nr-reporting-summary-flat.pdf)

# Life sciences study design

All studies must disclose on these points even when the disclosure is negative.

| | |
|---|---|
| Sample size | In vitro release experiments were performed with three biologically independent replicates and measured in duplicate. All cell uptake experiments were performed in duplicate. Three images were acquired per condition and analyses were carried out for all cells in each image. |
| Data exclusions | Outliers with low bead or cell counts were excluded from the data analysis in the flow cytometry data. |
| Replication | Details on electron microscopy experiments can be found in Supplementary Table 4. All gel electrophoresis experiments were replicated in at least two biologically independent experiments. All experiments were executed successfully. Electron micrographs with poor CTF estimations were excluded from data analysis. |
| Randomization | Cell and bead experiments were randomized such that replicate within a condition received a biased subset of cells |
| Blinding | All cell and bead experiments were performed with blinding study design |

# Reporting for specific materials, systems and methods

We require information from authors about some types of materials, experimental systems and methods used in many studies. Here, indicate whether each material, system or method listed is relevant to your study. If you are not sure if a list item applies to your research, read the appropriate section before selecting a response.

## Materials & experimental systems

| n/a | Involved in the study |
|-----|----------------------|
| ☐ | ☒ Antibodies |
| ☐ | ☒ Eukaryotic cell lines |
| ☒ | ☐ Palaeontology and archaeology |
| ☒ | ☐ Animals and other organisms |
| ☒ | ☐ Clinical data |
| ☒ | ☐ Dual use research of concern |

## Methods

| n/a | Involved in the study |
|-----|----------------------|
| ☒ | ☐ ChIP-seq |
| ☐ | ☒ Flow cytometry |
| ☒ | ☐ MRI-based neuroimaging |

## Antibodies

| | |
|---|---|
| Antibodies used | Myc-Tag (9B11) Mouse mAb (Cell Signaling Technologies, #2276), Anti-LAMP2A antibody (Abcam ab18528, dilution: 1:200), goat anti-rabbit- IgG Alexa Fluor™ 488 secondary antibody (Thermo Fisher A-11034, dilution 1:200), Cetuximab (Table 3, Methods), Anti-NaK ATPase (Abcam ab76020, Abcam ab283318, dilution: 1:200x), goat anti-mouse-IgG Alexa Fluor™ 488 secondary antibody (Thermo Fisher A-11029, dilution: 1:200x) |
| Validation | Commercial antibodies validated by manufacturers and validated via a certificate of analysis with lot number, measured concentration, and approved applications for Flow Cytometry and Immunoprecipitation<br>Cetuximab antibody was produced in house following the materials and methods protocol and SEC and assembly data validation is shown in the manuscript (Figure 2) |

## Eukaryotic cell lines

Policy information about cell lines and Sex and Gender in Research

| | |
|---|---|
| Cell line source(s) | All cell lines are reported in materials and methods. ExpiHEK293F (Thermo Fisher A14527), WT HeLa (ATCC CCL-2), EGFR KO HeLa (Abcam ab255385), and A431 cells (ATCC CRL-1555) |
| Authentication | Authentication was performed by the manufacturer for growth rate, mycoplasma contamination, STR profiling, and cell viability, and tested routinely during culturing for cell viability, cell morphology, and mycoplasma contamination. |
| Mycoplasma contamination | Cell lines tested negative for mycoplasma contamination upon thawing and were tested routinely during culturing for negative mycoplasma contamination |
| Commonly misidentified lines (See ICLAC register) | No commonly misidentified lines were used in this study. |

## Flow Cytometry

### Plots

Confirm that:

☒ The axis labels state the marker and fluorochrome used (e.g. CD4-FITC).

☒ The axis scales are clearly visible. Include numbers along axes only for bottom left plot of group (a 'group' is an analysis of identical markers).

☒ All plots are contour plots with outliers or pseudocolor plots.

☒ A numerical value for number of cells or percentage (with statistics) is provided.

### Methodology

| | |
|---|---|
| Sample preparation | Linear myc peptide with an N terminal lysine side chain and 3× glycine linker (KGGGEQKLISEEDL) was produced via solid-phase peptide synthesis and biotinylated via amide formation. The resulting biotinylated myc peptide was purified by RP-HPLC and quality checked for the proper molecular weight via LC-MS, lyophilized, and dissolved in 100% DMSO for long term storage. 3.0-3.4 µm streptavidin coated polystyrene particles (Spherotech) were incubated with biotinylated myc peptide diluted in PBS to 5% DMSO for 20 minutes at 25°C. Following two washes in PBS + 3% BSA by centrifugation at 3000×G for 5 minutes, coated polystyrene particles were incubated with α-myc-O432-17 nanoparticle variants for one hour at 25°C. α-myc antibody was purchased from Cell Signaling Technologies (9B11). The coated particles were washed in PBS + 3% BSA twice and split equally into pH-titrated citrate-phosphate buffers for 30 minutes at 25°C. Coated particles were washed twice with PBS + 3% BSA and resuspended for flow-cytometry. For O432-17(+) experiments, we prevented non-specific association to the polystyrene particles by substituting with EBY100 S. cerevisiae cells.<br><br>All flow-cytometry experiments were performed on a LSR II Flow Cytometer (BD Biosciences). Lasers were calibrated with coated particles stained with either FITC Anti-Myc tag antibody (Abcam 9E10), APC Anti-Myc tag antibody (Abcam 9E10), or no antibody. 10,000 events were collected per sample on three biological replicates. All flow-cytometry results were analyzed using the FlowJo, LLC software. Singlet beads were first isolated before analyzing for AF647 or sfGFP signal. Normalization to the minimum and maximum fluorescence of each channel with each titration sample was performed in Python3.7, and the |

apparent pKa of AF647 and sfGFP fluorescence was estimated with a four parameter nonlinear logistic regression fit in GraphPad Prism version 9.3.1 for Windows, GraphPad Software, San Diego, California USA, www.graphpad.com.

Instrument BD LSR II Flow Cytometer

Software Flowjov10.8

Cell population abundance Cell population abundance is shown in the supplementary materials (Supplementary Figure 8). 10,000 events were collected per well. Approximately 80% of the bead population was gated for bead morphology and purity, nearly all gated beads (99%) were singlet beads which were analyzed for Alexa-647 and FITC fluorescence.

Gating strategy The example gating strategy is shown in Supplementary Figure 8. Beads were initially gated on SSC-A and FSC-A, identifying a dense population of beads without abnormal morphologies or contamination. Singlet beads were identified using FSC-H and FSC-A. The cutoff for positive Alexa 647-A fluorescence was determined using the negative control for beads incubated with Alexa-647-conjugated trimer. The FITC cutoff was determined using a negative control for beads incubated with sfGFP-Fc.

☒ Tick this box to confirm that a figure exemplifying the gating strategy is provided in the Supplementary Information.

