## [Peer Review File · Nature Structural & Molecular Biology]

Peer Review Information

Manuscript Title: Computational design of non-porous pH-responsive antibody nanoparticles

Corresponding author name(s): David Baker, Neil King

Reviewer Comments & Decisions:

Decision Letter, initial version:

Message: 1st June 2023

Dear Dr Yang,

Thank you again for submitting your manuscript "Computational design of non-porous, pH-responsive antibody nanoparticles". I apologize for the delay in responding, which resulted from the difficulty in obtaining suitable referee reports. Nevertheless, we now have comments (below) from the 3 reviewers who evaluated your paper. In light of those reports, we remain interested in your study on the condition that the reviewers' concerns are critically addressed, and we would like to see your response to the comments of the referees, in the form of a revised manuscript.

You will see that while the reviewers appreciate the design strategy, Reviewers #1 and #3 raise major concerns about the lack of direct demonstration of pH-responsive release of cargo, or the cellular delivery of encapsulated cargo, and Reviewer #3 additionally points out the lack of testing of the effect of pore size on cargo retention. Seeing as these are central aspects of the study, we would like to see these points experimentally addressed before we can decide to send the manuscript back to the referees. Please be sure to address/respond to all concerns of the referees in full in a point-by-point response and highlight all changes in the revised manuscript text file.

We appreciate the requested revisions are extensive. We thus expect to see your revised manuscript within 6 months. If you cannot send it within this time, please let us know. We will be happy to consider your revision as long as nothing similar has been accepted for publication at NSMB or published elsewhere. Should your manuscript be substantially delayed without notifying us in advance and your article is eventually published, the received date would be that of the revised, not the original, version.

Reporting Summary:

When submitting the revised version of your manuscript, please pay close attention to our [href="https://www.nature.com/nature-portfolio/editorial-policies/image-integrity">Digital Image Integrity Guidelines. and to the following points below:](https://www.nature.com/nature-portfolio/editorial-policies/image-integrity)

Please note that all key data shown in the main figures as cropped gels or blots should be presented in uncropped form, with molecular weight markers. These data can be aggregated into a single supplementary figure. While these data can be displayed in a relatively informal style, they must refer back to the relevant figures. These data should be submitted with the last revision, prior to acceptance, but you may want to start putting it together at this point.

SOURCE DATA: we request that authors provide, in tabular form, the data underlying the graphical representations used in figures. This is to further increase transparency in data reporting, as detailed in this editorial (<http://www.nature.com/nsmb/journal/v22/n10/full/nsmb.3110.html>). Spreadsheets can be submitted in excel format. Only one (1) file per figure is permitted; thus, for multi-paneled figures, the source data for each panel should be clearly labeled in the Excel file; alternately the data can be provided as multiple, clearly labeled sheets in an Excel file. When submitting files, the title field should indicate which figure the source data pertains to. We encourage our authors to provide source data at the revision stage, so that they are part of the peer-review process.

While we encourage the use of color in preparing figures, please note that this will incur a charge to partially defray the cost of printing. Information about color charges can be

found at <http://www.nature.com/nsmb/authors/submit/index.html#costs>

We require deposition of coordinates (and, in the case of crystal structures, structure factors) into the Protein Data Bank with the designation of immediate release upon publication (HPUB). Electron microscopy-derived density maps and coordinate data must be deposited in EMDB and released upon publication. Deposition and immediate release of NMR chemical shift assignments are highly encouraged. Deposition of deep sequencing and microarray data is mandatory, and the datasets must be released prior to or upon publication. To avoid delays in publication, dataset accession numbers must be supplied with the final accepted manuscript and appropriate release dates must be indicated at the galley proof stage. Please find the complete NRG policies on data availability at <http://www.nature.com/authors/policies/availability.html>.

[redacted]

Sincerely,
Sara

Sara Osman, Ph.D.
Associate Editor
Nature Structural & Molecular Biology

Referee expertise:

Referee #1: Protein engineering, nanoparticles

Referee #2: Protein design

Referee #3: Rational design, biomolecular engineering

Reviewers' Comments:

Reviewer #1:

Remarks to the Author:

The manuscript by Yang et al. describes the combination of a previously reported pH-responsive trimeric protein with O42 antibody protein cages. The aim is to generate nanocapsules that can disassemble in acidic environments to release cargo. The authors extend the designed pH-responsive trimer, enabling it to bind to the O42 cages and reduce the size of the pores. The authors do a good job of outlining the problem that these designed nanoparticles aim to solve – endosomal release of molecular payloads from targeted delivery systems. Unfortunately, pH-responsive release of cargo is not demonstrated, neither is delivery of encapsulated cargo into cells and the indirect evidence provided for effect of pH on cage structure and fidelity requires considerable interpretation to come to the conclusions drawn. Additionally, the concepts of antibody-targeted protein cages and charge-driven cargo loading have been previously investigated in much more detail. In several instances, the experiments presented leave open questions and more thorough characterisation of the system is required. As such, this manuscript does not meet the standards of novelty, impact or scientific rigour that one expects from 'Nature Structural and Molecular Biology'.

Detailed comments:

No data is provided to show that the engineered pH responsive variants with point mutations (e.g. O432-17(-)_3HIS_I56V_L74A etc.) actually assemble into cages, with or without cargo. Moreover, no basic characterisation data is provided for these trimeric proteins.

pH response. The flow cytometry analysis is indirect and does not suffice to understand the effect of pH on these proteins without supporting techniques. Assessing the fluorescence of the supernatant after treatment in acidic conditions would be helpful to shown which components have been released from the cages. SDS-PAGE analysis to see the change on the ratios of the components would also be an easy way to directly assess effect on particle composition.

Porosity. The title refers to these protein capsules as non-porous. However, the authors point out on page 11 that the particles have pores of 3nm in diameter. While the reduction in pore size may be helpful for preventing release of cargoes, it is not explored in sufficient detail. For example, no data are provided for the stability of the cargo in biological serum and although the authors state (page21, para 2) that protein cargo was protected, there is no evidence provided to support this. Additionally, the experiment shown in Figure 4d does not thoroughly test the ability of the cages to protect RNA cargo. As in relevant studies (Butterfield et al Nature 2017, Tetter et al. Science 2021) RNase A enzyme should be used, and multiple timepoints longer than 30 minutes should be tested to provide a proper assessment.

Antibody-mediated cell targeting. Non-covalent recruitment of antibodies for cell-specific targeting has been extensively explored with different protein nanoparticles (Iijima et al Int. J. Cancer 1999, 80: 110-118; Ried et al. J. Virol. 2002, 76; Volpers et al. J. Virol 2003, 77; Kickhoefer et al. ACS Nano 2009, 3. are a few examples) and should be cited appropriately. One important questions to answer is - How stable are the bound antibodies

in the presence of other circulating ones? The cell experiments are carried out in serum-free media – how does this look in the presence of other antibodies as has been shown before?

Fidelity of assembly. On page 8 (para 2) the authors state that mixing tetramer and Fc results in partially formed particles. Indeed the SEC trace in Fig S2B shows a considerable fraction with identical retention time to the O432 particle. This fact contradicts the subsequent claim that the cooperativity prevents incomplete assembly of particles, as although the fully assembled particles containing all 3 components are favoured, it is possible that one or more trimeric units could be missing. The SEC trace shown in Fig 2B supports this case, as there is clearly unbound trimer after assembly, whereas there is no free tetramer or antibody. Assaying for cargo loading after cage formation would be one way to answer this question.

Cargo loading. Besides the fact that the negatively-charged pH-responsive particles do not load cargo (page 14, para 1) and no data or information is provided for the positively charged variants, the characterisation of the cargo loading described in Figure 4 is lacking. Details of the assembly process are not provided in the methods and only one technique of characterisation is used to demonstrate the encapsulation of either RNA or GFP. Many questions remain relating to the rate and efficiency of packaging, cargo capacity, stability and integrity of the encapsulation complexes. The TEM image provided for RNA loading cages looks very heterogeneous and none is provided for the negatively charged cages with GFP. Overall the characterisation and analysis of the cargo loading is not up to the standard expected in the field.

Charged variants. The description and characterisation of the positively and negatively charged trimer variants are lacking. The sequences and variant names provided in the SI further confuse the reader as the mutations I56V and L74A should read I57V and I75A based on the parent sequence (O432-17-C3). Additionally, some sequences appear to have an extra Met residue at the N-terminus. Furthermore, it is not shown whether O432 particles can be formed using the charged trimer variants without any cargo. This is an interesting question to answer about the stability of the system.

Confocal microscopy. The images should show a nuclear stain or a brightfield image to provide a reference to the cell structure. Displaying the AF647 signal as white in the composite also makes co-localisation more difficult to identify. Although difficult to properly judge from the images provided, there seems to be considerable overlap of the mRuby2 and AF647 signal, suggesting that particles may be intact. Additionally, it is unclear why there would be so much LAMP2A signal in the nuclei? A FRET-based approach may be helpful to better interrogate the behaviour of these assemblies in cellulo.

Additional comments:

Fig. S1A – It would be helpful to add some arrows indicating the trimer and tetramer proteins. The gel is currently hard to interpret and it is unclear where the trimeric protein band is.

Fig. S1B - Why do O42.1 particles show >4 bands in the non-denaturing PAGE? Where is the expected mobility of the band corresponding to a complete particle?

Fig. S1C – One would expect the tetrameric proteins to have very similar size/shape and

elute with the same retention time. How do these compare to a known reference, e.g. O42.1 C4. The authors should also identify which peaks were isolated and used in the subsequent experiments.

Fig. S1D/F – The Fc fragment used in the experiments looks to be partially truncated (S1F).

Fig. S2C – Why is the tetramer retention time different between panels B and C?

An SDS-Page gel for the 16 designs where tetramer & trimer co-elute (before sfGFP-Fc addition) is missing.

Fig. 4d – is missing a ladder to reference the RNA length.

Fig. 4E – The coomassie staining looks rather odd with empty bands, is the gel overloaded?

Page 16 – The references to Fig. S6D-E should be to S6F-G

Page 18 – The following points are contradictory - incubated up to 16 hours...after 3 hours the cells were fixed?

Fig. S5 – This sample definitely looks quite heterogeneous. The caption title states the wrong protein variant.

Table S2 - What does the star refer to in the SI for some of the amino acid sequences?

Abstract – ‘variety of molecular payloads’, two is not really a variety.

Abstract – ‘designed nanoparticles’ is rather vague and therefore incorrect (e.g. LNPs) Should specify that these are designed protein-based nanoparticles.

Intro – “require assembly and encapsulation of molecules outside the target cell”. This is unnecessarily confusing – simply stating in vitro encapsulation may be clearer.

Reviewer #2:

Remarks to the Author:

This manuscript describes the design and characterization of a protein nanoparticle that assembles from tetrameric, trimeric, and dimer (Fc/antibody) components; the trimer furthermore is designed to disassemble at low pH based on histidine residues arranged at interfaces. The idea is to confer opening of large pores, or disassembly, under endosomal conditions. The study represents a substantial design effort, requiring a high level of expertise. Overall the experimental work to guide and confirm a successful design result is thorough and sound. The writing is clear. The manuscript will provide a notable contribution to the timely area of designing complex protein assemblies. I have the following points to consider before publications.

1) The specific particle that results could be useful in applications, but arguably what will be of most utility to others are the ideas and methods presented. In that regard, a point

that should be made clearer at the abstract level is that many (45) experimental candidates formed the basis for the eventual successful outcome. Success rates for protein design efforts remain a key concern for the field, and it's important to emphasize what a challenging effort it is (and what expertise is required) to realize success in these kinds of endeavors.

2) Following the point above, later in the paper it would be helpful to add a note or two about how the intermediate screening of designs should be interpreted, and perhaps what led to attrition. If my reading is correct, the initial pulldown results aren't shown but it is reported that there was some evidence for physical association of designed tetramer and trimer in 16 candidates, but no tests for geometrically correct assembly at that stage. From those, 4 (or 5? -- #43 doesn't seem to show the expected tetramer band?) appeared to also associate with the dimer. And from those 4 where association of three components seemed to occur, only one appeared to assemble (predominantly) into the expected cage structure based on SEC. Something to that effect should be explained if that's the right interpretation.

And what might be said about the distinction between associating vs assembling precisely as designed? Is it likely that this reflects irregular (i.e. geometrically incorrect) or partial forms of assembly? Could kinetic or entropic trapping effects be at play? Answers might not be known but it seems worth bringing up ways of thinking about what might be limiting high success rates.

3) The designed architecture here is indeed unique, but the strong emphasis (abstract and discussion) on being the first to include three structural/symmetry elements needs to be re-thought. Cannon et al. (2020) demonstrated a designed icosahedral assembly based on pentameric + trimeric + dimeric components. That paper is included in passing in group citations, but otherwise goes unnoted. That previous work was based on genetically fusing three symmetric components while the present study relies on non-covalent associations, so one could talk about the distinction. But the more broadly phrased claim of priority without connection to the prior study doesn't seem right. [Referencing in the paper seems good otherwise.]

4) The design logic for the flow cytometry experiments near the top of page 11 is hard to understand as described. After the topic sentence for that paragraph, it would be helpful to try to elaborate in more plain language what is being devised here.

Minor comments:

- The earlier O42.1 assembly comes up by name at the top of page 5 but without a reference or further description. I think this is Divine 2021? Please clarify.

- Fig S1 A. The reader doesn't have a way of evaluating this gel without the MW of the trimers. At least for the cases that are being asserted to be successes, the trimer MW should be given, perhaps alongside arrow markers for the bands on the gel that are being ascribed to the trimer components.

- Where OMAC experiments are being mentioned, please indicate/remind the reader which subunit is carrying the tag.

- The figure legends are constructed strangely. E.g. "separately, D. and, " and so on. Having a period followed by a lower case letter makes it difficult to parse what is what.
- Top of page 11. It would be an improvement to avoid jargon: "relaxing"

Reviewer #3:

Remarks to the Author:

Key results

The manuscript by Yang et al. reports on the rational design of antibody-decorated hollow protein nanocages that encapsulate cargo and can controllably disassemble upon dropping the pH. The architecture of the nanocage is largely based on the previous Science paper by Divine et al. 2021, in which an octahedral porous cage was formed from de novo-designed protein tetramers and Fc fragments of antibodies. In the present study, the pores in the walls of the cage were controllably blocked by a third, trimeric protein plug component. The interface between the trimer and the tetramers features pH-tunable ionic bonds that help controllably disassemble the plug from the porous cage.

The protein plug was designed by expanding in size a small trimeric core using a combination of helical fusion and protein docking, followed by optimization with Rosetta sequence design calculations. Thousands of design variants were generated and screened to check whether the protein plug fits the size and shape of pores within the nanocage. 45 of the design variants were then experimentally produced and tested with PAGE and SEC to check for the assembling of the nanocage from the plug, tetramers and antibody components. For a smaller subset, the nanocages were characterized by DLS and cryo-EM. The experimental structures of the nanocages were compared to the initial designs and found to be in good agreement. A selected protein nanocage was then used to encapsulate RNA inside the cage structure to protect the nucleic acid cargo from nuclease digestion. The pH-responsive disassembly of the plugs from the hollow nanocage was tuned by point mutations in the subunits, and the corresponding pK_as for disassembly were measured with flow cytometry. Finally, the antibody-decorated nanoparticles were added to eucaryotic cells to demonstrate antigen-specific cellular uptake into endosomes using confocal fluorescence microscopy as read-out.

Originality and significance

The rational design of a hollow protein nanocage with a pH-removable protein plug is novel, as is the protective encapsulation of RNA within a designed nanocage. The study is significant in protein design as helical fusion is demonstrated to help obtain the protein plug by expanding a small core to fit in size and shape to the nanocage pores. As other significant point, the designed protein architecture integrates three functions, namely encapsulation within a nanocage, specific molecular recognition, and pH-dependent disassembly. Finally, the structure may be used for targeted delivery of bioactive cargo inside cells, even though more work would be required to demonstrate the benefit for research or drug delivery. While the findings are novel and significant, more could be done to place them within the wider context of biomimetic design, nanobiotechnology, drug delivery, and other protein designs, as detailed under 'Conclusions'.

Methodology and data

The methodology uses the appropriate tools for answering the scientific questions. Most of the claims in the manuscript are strongly supported by experimental evidence. Two points should be addressed with additional experiments described under 'Improvements'. The use of statistics and treatment of uncertainties is good.

Conclusions

While the findings are novel and significant, more could be done to place them into a wider context along the following points.

In terms of biomimetic design, protein shells that encapsulate nucleic acids clearly replicate the overall principle and function of viral particles. How does the designed structure compare to other viral particles? For example, are the designed protein nanocages smaller than any viral template? What about the encapsulated volume? The symmetry of the components may also be compared.

Within the wider context of nanobiotechnology, non-protein multimeric nanocages have previously been constructed via DNA nanotechnology. Examples include Sigl, C. et al. (2021) 'Programmable icosahedral shell system for virus trapping', *Nature Materials*, 20(9), pp. 1281–1289. doi:10.1038/s41563-021-01020-4 and the environmentally sensitive assembly and encapsulation Ijäs, H. et al. (2019) 'Reconfigurable DNA origami nanocapsule for pH-controlled encapsulation and display of cargo', *ACS Nano*, 13(5), pp. 5959–5967. doi:10.1021/acsnano.9b01857 and Hu, Y. et al. (2020) 'Dynamic DNA assemblies in biomedical applications', *Advanced Science*, 7(14), p. 2000557. doi:10.1002/advs.202000557. The revised introduction or discussion of the manuscript should mention that nanocages and stimulus responsiveness can also be designed with DNA.

Relating to drug delivery, the biomedical application potential of the protein cages needs to be discussed. Firstly, which cargo would benefit most from cellular delivery by the new nanocages? Second, protein cage disassembly within the acidic endosomes is not sufficient as the cargo must also be released from the endosome. Indeed, accumulation of the fluorescent protein seems to appear in the microscopy images of the cells in Fig. 5. Furthermore, what are benchmark delivery methods against which the protein cages should be compared to in terms of uptake, release, cargo size, a.o.?

Finally, the principle of creating a plug for a protein hole may be applied for designing other protein structures such as porous and reversibly closing 2D lattices or gated membrane channels. The outlook of the manuscript should mention the above points to better appreciate the significance of the findings.

Improvements:

1. The effect of porosity on cargo retention

A major claim of the paper is that adding a plug into a porous nanocage can encapsulate cargo inside the protein carrier. By adding the plug, the pore size decreases from 13 nm to 3 nm, based on cryo-EM data. However, the cargoes tested are bound during the octahedron assembly and -from the reviewer's impression- the effect on pore size on cargo retention is never explicitly tested in the publication. The effect of pore size on cargo

retention should be experimentally tested to better appreciate the usefulness of the nanocage (European Journal of Pharmaceutical Sciences, 120, pp. 199–211. doi:10.1016/j.ejps.2018.05.004.).

2. The pH-dependent release of the cargo

Another claim of the paper is how the pH-dependent detachment of the trimer-plug can control cargo release from the nanocage lumen. The manuscript only demonstrates the detachment of the trimer from the nanocage which is, however, not filled with cargo. To close this gap, the authors should test the pH-dependent release of the RNA (or other) cargo.

Minor points: Most experiments are clearly described and expanded in the supplementary information. However, there are a few points that need clarification.

1. What is the particle formation yield for the selected design? Would it be possible to give a % estimate normalized by the concentration of Fc added?
2. Figure 5: The data on cellular uptake lack more quantitative analysis of the results. For example, data on average fluorescence per cell and number of cells with fluorescence signals should be presented.
3. Figure S1-F. There is no band for the condition Tetramer+Fc. However, Tetramer + Fc should produce the cage structure or at least just some aggregates. Why is there no band?
4. Figure S1 caption: E is used to refer to subfigure F and subfigure G is not referred to.
5. The nanoparticles were incubated with cells in serum-free media. What happens when cells are incubated with serum-containing media? The latter condition would mimic biomedical applications.
6. It is clear that the packaging/loading of molecular cargos is highly dependent on the electrostatic interaction between the plug and the cargo. Could there be other factors which affect the packaging of cargo in the nanocages, such as size/molecular weight of the cargo? What is the expected maximum cargo size? What is the expected correlation between cargo size and the assembly efficiency of the nanocages?

Clarity and context:

The text is overall very clearly written.

Minor points:

1. Several places in the publication refer to “incubations at a concentration per monomer”. What is defined as monomer? Does this include all elements in the nanoparticle including the antibodies or only the monomers forming the tetramer and trimer?
2. Figure 1B: Which of the fused trimer was chosen in the experimental screening process?
3. Page 8, line 12: “The optimal assembly ratio per protomer of purified trimeric plug, tetramer, and Fc was determined to be 1.1:1.1:1.” Is this the stoichiometric ratio? How was this ratio obtained? The following reference should be considered. Nature Communications, 2015 6(1). doi:10.1038/ncomms7203.
4. Page 9: “This cooperativity simplifies preparation of the three component nanoparticles as it prevents incomplete assembly of nanoparticles containing two out of the three components”. Would not Fc and the tetramer interact on their own?
5. Page 14 “After 30 minutes, the beads were resuspended and brought back to pH 7.5 to...” Did the authors mean: “beads were collected by centrifugation and resuspended

in...”?

6. Page 14/15. At the beginning of the section, the target pKa is not mentioned. This is slightly confusing.

Author Rebuttal to Initial comments

Reviewers' Comments:

Reviewer #1:

Remarks to the Author:

The manuscript by Yang et al. describes the combination of a previously reported pH-responsive trimeric protein with O42 antibody protein cages. The aim is to generate nanocapsules that can disassemble in acidic environments to release cargo. The authors extend the designed pH-responsive trimer, enabling it to bind to the O42 cages and reduce the size of the pores. The authors do a good job of outlining the problem that these designed nanoparticles aim to solve – endosomal release of molecular payloads from targeted delivery systems. Unfortunately, pH-responsive release of cargo is not demonstrated, neither is delivery of encapsulated cargo into cells and the indirect evidence provided for effect of pH on cage structure and fidelity requires considerable interpretation to come to the conclusions drawn. Additionally, the concepts of antibody-targeted protein cages and charge-driven cargo loading have been previously investigated in much more detail. In several instances, the experiments presented leave open questions and more thorough characterisation of the system is required. As such, this manuscript does not meet the standards of novelty, impact or scientific rigour that one expects from 'Nature Structural and Molecular Biology'.

Detailed comments:

No data is provided to show that the engineered pH responsive variants with point mutations (e.g. O432-17(-)_3HIS_I56V_L74A etc.) actually assemble into cages, with or without cargo. Moreover, no basic characterisation data is provided for these trimeric proteins.

We thank the reviewer for their feedback. We now provide SEC, DLS, and NS-EM validation of the trimer component and assemblies of all pH-responsive variants with point mutations without cargo in new Supplementary Figures 8 and 10, and NS-EM validation of both O432-17(-) and O432-17(+) assemblies containing cargo in the revised Supplementary Figure 7 (formerly Supplementary Figure 5). We refer to them in the following text, which we have added to the Results section:

“SEC of the trimer and the resulting assembly, DLS, and NS-EM micrographs and two-dimensional class averages confirmed O432-17(-) assembly in the absence of cargo, suggesting that assembly stability is cargo-independent (Supplementary Fig. 8a-d).”

“Each negatively-charged pH-responsive variant was validated by SEC of the trimeric plug component and for assembly competency with the tetramer and Fc by SEC, DLS, and NS-EM (Supplementary Fig. 8a-d).”

“SEC of the trimer and the resulting assembly, DLS, and NS-EM confirmed three-component nanoparticle assembly of O432-17(+) with and without the presence of the pegRNA (Supplementary Fig. 7a,b; Supplementary Fig. 10a-d).”

“All pH-responsive O432-17(+) trimeric plug variants were validated by SEC and for assembly competency with tetramer and Fc and showed an apparent pKa of pH 6.1 for AF647 and pH 5.8 for sfGFP fluorescence, respectively (Supplementary Fig. 9e,f; Supplementary Fig. 10a-d)”

We also provide NS-EM validation of O432-17(-) and O432-17(+) assemblies with cargo in a revised Supplementary Figure 7 and refer to this figure in the text:

SEC of the trimer and the resulting assembly, DLS, and NS-EM confirmed three-component nanoparticle assembly of O432-17(+) with and without the presence of the pegRNA (Supplementary Fig. 7a,b; Supplementary Fig. 10a-d).”

“One variant, which will be referred to as O432-17(-), interacted with pos36GFP in these conditions, as shown by co-elution of the 488 nm signal and the 280 nm signal during SEC, and NS-EM confirmed three-component nanoparticle assembly of O432-17(-) with pos36GFP (Fig. 4e; Supplementary Fig. 7c,d).”

Finally, we included in the original manuscript a sentence describing that only the original O432-17(-) and O432-17(+) were capable of encapsulating cargo and assembling properly:

“These pH-responsive, O432-17(-) variants did not encapsulate pos36GFP, perhaps due to differences in assembly efficiency of the fully plugged nanoparticle in the presence of molecular cargo, and hence we were only able to investigate the pH-mediated release of encapsulated cargo for the O432-17(-) variant (O432-17(-)_2HIS (Fig. 4k).”

pH response. The flow cytometry analysis is indirect and does not suffice to understand the effect of pH on these proteins without supporting techniques. Assessing the fluorescence of the supernatant after treatment in acidic conditions would be helpful to shown which components have been released from the cages. SDS-PAGE analysis to see the change on the ratios of the components would also be an easy way to directly assess effect on particle composition.

We thank the reviewer for these helpful suggestions. To check for which components have been released from the cages, we looked for pH-mediated release of a) fluorescent protein cargo and b) a fluorescently labeled component into the supernatant after treatment in acidic conditions. We compared the plugged O432-17(-) nanoparticle to the original unplugged O42.1 antibody nanoparticle. We found that the fluorescence signal of both pos36GFP (cargo) and mRuby2-Fc (nanoparticle component) in the supernatant after each wash following the first wash at pH 8 plateaued at wash 7 (see Figures A and B below). The fluorescence of the cargo in the supernatant after acidic incubation was much greater in the O432-17(-) sample than the O42.1 sample, and there was no significant difference of pos36GFP fluorescence on resuspended yeast cells following acidic incubation between the nanoparticle samples. We include the data

for pH-dependent release of cargo in a revised Figure 4 and refer to it in the following new text in the Results section:

“We next sought to release encapsulated pos36GFP cargo via pH-mediated disassembly of O432-17(-) nanoparticles. O432-17(-) trimer, tetramer, and pos36GFP were assembled in vitro using a 50/50 mixture of α -Myc mouse antibody and mRuby2-Fc, the latter of which enabled us to monitor nanoparticle disassembly (Fig. 4h). As a comparison, we also assembled O42.1 with the same antibody cocktail and pos36GFP (Fig. 4i). Assembled nanoparticles were immobilized on *S. cerevisiae* cells displaying the myc peptide, the cells were collected by centrifugation, and the fluorescence of the resulting supernatant was measured before resuspending the remaining cells in fresh buffer. The cells were then resuspended in citrate-phosphate buffer at low or high pH (4.2 and 8, respectively) for 30 minutes, after which the supernatant was collected and buffer exchanged to neutral pH before fluorescence readout (Fig. 4j). At pH 8, we did not observe a substantial difference in mRuby2-Fc fluorescence between the two nanoparticles (133.5 relative fluorescence units (RFU) for O432-17(-) and 45.17 RFU for O42.1), suggesting that both nanoparticles remained intact in neutral pH conditions (Fig. 4k, Supplementary Table 8). At pH 4.2, we observed a mean mRuby2-Fc fluorescence difference of 846 RFU for O432-17(-) and 394.3 RFU for O42.1 relative to buffer (23 RFU), suggesting that both nanoparticles disassemble at low pH.

We observed pH-dependent pos36GFP cargo release from the plugged O432-17(-) nanoparticles (Fig. 4k). Incubation of the O42.1 nanoparticles at either low or high pH (4.2 and 8, respectively) did not result in substantial release of pos36GFP cargo into the supernatant, as little cargo was left after washing since there is no plug present to prevent cargo leakage from the nanoparticle. Incubation of O432-17(-) at pH 8 also failed to release substantial fluorescence into the supernatant. By contrast, O432-17(-) released considerably more posGFP cargo into the supernatant after incubation at pH 4.2 than O42.1 (1013 RFU and 463.8 RFU, respectively). The designed trimeric plug on O432-17(-) nanoparticles thus enables packaging, retention, and pH-mediated release of encapsulated cargo.”

Figure A: pos36GFP fluorescence signal of O432-17(-) and O42.1 nanoparticles measured at an excitation of 460 nm and an emission wavelength of 509 nm in the supernatant following each bead wash (following wash 2), in the supernatant after pH 4 incubation, and remaining in the pellet following pH 4 incubation.

Figure B: pos36GFP fluorescence signal of O432-17(-) and O42.1 nanoparticles measured at an excitation of 559 nm and an emission wavelength of 600 nm in the supernatant following each bead wash (following wash 2), in the supernatant after pH 4 incubation, and remaining in the pellet following pH 4 incubation.

Porosity. The title refers to these protein capsules as non-porous. However, the authors point out on page 11 that the particles have pores of 3nm in diameter. While the reduction in pore size may be helpful for preventing release of cargoes, it is not explored in sufficient detail. For example, no data are provided for the stability of the cargo in biological serum and although the authors state (page21, para 2) that protein cargo was protected, there is no evidence provided to support this. Additionally, the experiment shown in Figure 4d does not thoroughly test the ability of the cages to protect RNA cargo. As in relevant studies (Butterfield et al Nature 2017, Tetter et al. Science 2021) RNase A enzyme should be used, and multiple timepoints longer than 30 minutes should be tested to provide a proper assessment.

We thank the reviewer for their feedback. We have now added RNase A to the RNA challenge assay and modified the incubation time from 30 minutes to 1 hour, as in relevant studies (Butterfield et al. Nature 2017, Tetter et al. Science 2021). We also now demonstrate the effect of the reduction in pore size by including the plugless O42.1 particle as a control for both RNA

and GFP packaging and also included a ssRNA ladder in this new gel. These data are included in the revised Figure 4 and described as follows in the Results section:

“We screened for packaging of nucleic acid by assembling positively charged trimeric plug variants, the designed tetramer, and the α -EGFR antibody Cetuximab (CTX) with a 154-nt prime editing guide RNA (pegRNA) cargo (Hsu et al. 2021; Anzalone et al. 2019; Nelson et al. 2022) (Fig. 4a). Packaging and protection of nucleic acid cargo in the presence and absence of nuclease was assessed via non-denaturing electrophoresis, staining with SYBR Gold and Coomassie to detect RNA and protein, respectively. One variant, referred to as O432-17(+), showed comigration of nucleic acid and nanoparticle with and without Benzonase treatment (Fig. 4b). SEC of the trimer and the resulting assembly, DLS, and NS-EM confirmed three-component nanoparticle assembly of O432-17(+) with and without the presence of the pegRNA (Supplementary Fig. 7a,b; Supplementary Fig. 10a-d). Excess nucleic acid that did not comigrate with the protein was degraded in the Benzonase-treated sample, demonstrating nanoparticle protection of the nucleic acid that comigrated with O432-17(+). We did not observe comigration of O432-17(+) and pegRNA after RNase A treatment (Fig. 4b,c); the remaining pore size after the addition of the plug is likely still large enough to admit the 14 kDa enzyme, while small enough to exclude the 60 kDa Benzonase (Tetter et al. 2021). Nucleic acid comigrating with the plugless O42.1 nanoparticle was degraded after both Benzonase and RNase A treatment, demonstrating that the trimeric plug is required for packaging and protection of nucleic acid cargo (Fig. 4b,c).

We also designed negatively charged plug variants to package positively charged protein cargoes. We screened for packaging of positively charged GFP (pos36GFP) (Zuris et al. 2015) by assembling negatively charged trimeric plug variants with different interior surface charge with tetramer, Fc, and pos36GFP (Fig. 4d). One variant, which will be referred to as O432-17(-), interacted with pos36GFP in these conditions, as shown by co-elution of the 488 nm signal and the 280 nm signal during SEC, and NS-EM confirmed three-component nanoparticle assembly of O432-17(-) with pos36GFP (Fig. 4e; Supplementary Fig. 7c,d). Based on measurements of fluorescence intensity and UV/Vis absorbance, we estimate approximately 9 to 10 pos36GFPs are packaged per nanoparticle in 200 mM NaCl, occupying roughly 40% of the interior volume. Including 1 M NaCl in the buffers used during packaging and SEC prevented packaging (Fig. 4f), demonstrating that GFP packaging is largely driven by electrostatic interactions between the cargo and nanoparticle interior. In vitro assembly of O42.1 (Divine et al. 2021) with pos36GFP revealed no cargo packaging, indicating that presence of the designed trimeric plug is necessary for cargo packaging (Fig. 4g). SEC of the trimer and the resulting assembly, DLS, and NS-EM micrographs and two-dimensional class averages confirmed O432-17(-) assembly in the absence of cargo, suggesting that assembly stability is cargo-independent (Supplementary Fig. 8a-d). Thus, our designed three-component nanoparticles efficiently encapsulate and protect molecular cargoes.”

Antibody-mediated cell targeting. Non-covalent recruitment of antibodies for cell-specific targeting has been extensively explored with different protein nanoparticles (Iijima et al. *Int. J. Cancer* 1999, 80: 110-118; Ried et al. *J. Virol.* 2002, 76; Volpers et al. *J. Virol* 2003, 77; Kickhoefer et al. *ACS Nano* 2009, 3. are a few examples) and should be cited appropriately. One important question to answer is - How stable are the bound antibodies in the presence of other circulating ones? The cell experiments are carried out in serum-free media – how does this look in the presence of other antibodies as has been shown before?

We thank the reviewer for their feedback. We've now included the recommended citations in our Introduction and provide new data on receptor-mediated cellular uptake of nanoparticles with and without serum starvation in HeLa WT and HeLa EGFR knockout cells. The results for these experiments are included in the new Supplementary Figure 12 and referred to as follows in the Results section:

“There is considerable interest in tailoring nanoparticle platforms for targeted delivery of therapeutic molecules. Effective nanoparticle platforms for targeted delivery require in vitro cargo encapsulation followed by target recognition, triggered nanoparticle disassembly, and controlled cargo release once inside the cell (Edwardson et al. 2018; Mitchell et al. 2020; Banskota et al. 2022; Wang et al. 2019; Hou et al. 2021; Douglas and Young 2006; Seo et al. 2021; Van de Steen et al. 2021; Azuma et al. 2018; Sigl et al. 2021; Ijäs et al. 2019; Hu et al. 2020; Ried et al. 2002). Cellular uptake of extracellular molecules such as protein-based nanoparticles via endocytosis involves traversal of membrane-bound organelles, including the low-pH endosome and lysosome (Dimitrov 2004; Martens et al. 2014; Lönn et al. 2016; Czapar and Steinmetz 2017). While a number of self-assembling protein nanoparticles with customized structures have been designed, they are composed of just one or two unique, static building blocks and efforts to adapt them for cargo packaging and delivery applications are still in their infancy (Cannon et al. 2020; Lai, King, and Yeates 2012; King et al. 2014; Butterfield et al. 2017; Votteler et al. 2016; Tetter et al. 2021; Levasseur et al. 2021; Edwardson, Mori, and Hilvert 2018). *Antibodies are particularly attractive as targeting moieties for delivery applications, and several previous studies have described various ways of incorporating antibodies into nanoparticle delivery platforms (Iijima et al. 1999; Volpers et al. 2003; Kickhoefer et al. 2009; Kim et al. 2016; Rujas et al. 2021).* We recently reported the computational design of antibody-incorporating nanoparticles in which a designed homooligomer drives the assembly of any antibody of interest into bounded, multivalent, polyhedral architectures (Divine et al. 2021) (Fig. 1a). While such antibody nanoparticles can activate signaling through a variety of cell surface receptors, they are quite porous, which complicates the packaging and retention of molecular cargoes.”

“We next tested the ability of the O432-17 nanoparticles to enter cells through receptor-mediated endocytosis. We assembled targeted nanoparticles by incubating a 1:1 stoichiometric mixture of α -EGFR antibody (CTX) and Fc fused to mRuby2 (mRuby2-Fc) with the tetramer and the 3HIS_I57V_L75A trimeric plug variant labeled with AF647 (named O432-17-CTX). As negative controls, we assembled a non-EGFR

targeting nanoparticle by mixing the tetramer and trimer-AF647 with mRuby2-Fc (named O432-17-Fc) (Supplementary Fig. 11a). Assembled O432-17-CTX and O432-17-Fc were incubated with A431 cells which display high EGFR expression and imaged with epifluorescence microscopy. We observed a significantly higher percentage of A431 cells positive for O432-17-CTX (44%) than O432-17-Fc (5%), suggesting that the O432-17(-) nanoparticles require EGFR binding for efficient endocytosis (Supplementary Fig. 11b,c). The EGFR-targeted nanoparticle bound to 45% and 68% of WT HeLa cells, which express moderate levels of EGFR, with and without serum, respectively, while HeLa EGFR KO cells were not efficiently labeled in either condition (10% with serum and 7% without serum) (Supplementary Fig. 12b,c; the reduced labeling in the presence of serum likely reflects a slight instability in the nanoparticle structure in the presence of high concentrations of exogenous antibody (Divine et al. 2021)). Together, these data establish that the antibody component of O432-17 nanoparticles enables targeting of cells expressing a specific receptor.”

Fidelity of assembly. On page 8 (para 2) the authors state that mixing tetramer and Fc results in partially formed particles. Indeed the SEC trace in Fig S2B shows a considerable fraction with identical retention time to the O432 particle. This fact contradicts the subsequent claim that the cooperativity prevents incomplete assembly of particles, as although the fully assembled particles containing all 3 components are favoured, it is possible that one or more trimeric units could be missing. The SEC trace shown in Fig 2B supports this case, as there is clearly unbound trimer after assembly, whereas there is no free tetramer or antibody. Assaying for cargo loading after cage formation would be one way to answer this question.

We thank the reviewer for their feedback. We previously demonstrated completeness of assembly in Figure 3 and Supplementary Figure 5, in which we show octahedral assemblies containing components on all axes of symmetry without imposing any symmetry during electron microscopy data processing. We only impose octahedral symmetry in the refinement of the helical resolution of the final model. We also now clarify in the text that the Fc and tetramer alone formed predominantly aggregate and add our hypothesis that the newly designed hydrophobic interface on the tetramer contributed to off-target interactions, resulting in either nanoparticle instability (in the fraction that elutes at the same time as the O432 particle), or visible aggregate. The revised passage in the text now reads (additions in italics):

“Optimization of the stoichiometric ratios of each protomer present during *in vitro* assembly resulted in the O432-17 SEC peak shifting from the void volume toward the expected O42.1 elution volume (Fig. 2b). The optimal assembly ratio per protomer of purified trimeric plug, tetramer, and Fc was determined to be 1.1:1:1; the slight excess of trimeric plug favors complete assembly. From the area under the SEC peak, we estimate a total particle yield of 95% (see methods). We found that the assembly of the designed nanoparticle was cooperative and required all three components: mixing any two of the three O432-17 components stoichiometrically did not result in fully assembled

nanoparticles (Supplementary Fig. 4a). As expected, mixing the trimeric plug and Fc resulted in no assembly, as there is no designed interface between those two components. Mixing the tetramer with Fc resulted in visible aggregate (Supplementary Fig. 4b), likely due to the formation of off-target interactions by the newly designed hydrophobic plug interface. *Although we observe a fraction of the tetramer and Fc mixture with the same retention time as the O432 particle, the dominant species eluted in the void volume, suggesting that without the third component, the tetramer and Fc particle is more unstable than the previously designed O42.1 particle. Mixing the tetramer and trimeric plug did not result in association even at high concentration (400 μ M per monomer of each component) (Supplementary Fig. 4c). The hierarchy of interface affinities between the three components appears to drive cooperative assembly, such that the trimeric plug and tetramer only interact in the presence of the third Fc component (Murugan et al. 2015; Wargacki et al. 2021).* This cooperativity simplifies preparation of the three-component nanoparticles as it prevents incomplete assembly or assembly hysteresis of nanoparticles containing two out of the three components, and thus eliminates the need for additional purification steps to separate these species from the intended three-component assembly. Dynamic light scattering (DLS) of the optimized O432 peak indicated a hydrodynamic diameter of 34 nm and a polydispersity index (PDI) of 0.05 (Fig. 2c) and negative-stain electron microscopy (NS-EM) revealed monodisperse nanoparticles (Fig. 2d). Two-dimensional class averages of negatively stained micrographs revealed plug-like density in the 3-fold views compared to the original two-component antibody nanoparticle. ”

Cargo loading. Besides the fact that the negatively-charged pH-responsive particles do not load cargo (page 14, para 1) and no data or information is provided for the positively charged variants, the characterisation of the cargo loading described in Figure 4 is lacking. Details of the assembly process are not provided in the methods and only one technique of characterisation is used to demonstrate the encapsulation of either RNA or GFP. Many questions remain relating to the rate and efficiency of packaging, cargo capacity, stability and integrity of the encapsulation complexes. The TEM image provided for RNA loading cages looks very heterogeneous and none is provided for the negatively charged cages with GFP. Overall the characterisation and analysis of the cargo loading is not up to the standard expected in the field.

We appreciate the reviewer’s constructive feedback. We did not observe nanoparticle assembly of the pH-responsive positively charged variants in the presence of RNA cargo, an observation we have now added to the text for clarity. We also added NS-EM micrographs and 2D class averages of the RNA-containing and GFP-containing nanoparticles in Supplementary Figure 7, showing that the nanoparticles assemble as intended with cargo. Finally, we now describe the details of the assembly processes in the Methods section as follows:

“In vitro packaging of pos36GFP

O432-17(-) nanoparticles with pos36GFP cargoes were assembled in the same molar ratio of purified 1 \times tetramer, 1 \times Fc, and 1.1 \times negatively charged trimeric plug from gel filtration fractions with 1 \times purified pos36GFP. Mixtures were dialyzed at 25°C for 16

hours into assembly 25 mM Tris pH 8.0, 200 mM NaCl, or 25 mM Tris pH 8.0, 1 M NaCl. Nanoparticles packaging pos36GFP were screened for cargo packaging by SEC on a Superose 6 10/300 GL column (Cytiva) in either buffer. Packaged GFP was quantified using the absorbance measurements at 280 and 488 nm obtained using a NanoDrop 8000 spectrophotometer. The absorbance of pure pos36GFP at 280 and 488 nm at 200 mM NaCl was used to calculate the absorbance at 280 nm due to pos36GFP in gel filtration fractions containing pos36GFP packaged in O432-17(-) nanoparticles. The relative absorbance due to pos36GFP and O432-17(-) nanoparticles was used to calculate the molar ratio of each protein using calculated extinction coefficients and quantified against a standard curve generated using pure pos36GFP (Bale et al. 2016).

In vitro packaging of pegRNA

O432-17(+) nucleocapsids with pegRNA (Integrated DNA Technologies) were assembled in a molar ratio of purified 1× tetramer, 1× Cetuximab IgG, and 1.1× positively charged trimeric plug from gel filtration fractions with 3× pegRNA per nucleocapsid. Mixtures were dialyzed at 25°C for 16 hours into 25 mM Tris pH 8.0, 150 mM NaCl. Nucleocapsids were screened for in vitro encapsidation by native gel electrophoresis before and after a 1 hour treatment with Benzonase or RNase A.”

Charged variants. The description and characterisation of the positively and negatively charged timer variants are lacking. The sequences and variant names provided in the SI further confuse the reader as the mutations I56V and L74A should read I57V and I75A based on the parent sequence (O432-17-C3). Additionally, some sequences appear to have an extra Met residue at the N-terminus. Furthermore, it is not shown whether O432 particles can be formed using the charged trimer variants without any cargo. This is an interesting question to answer about the stability of the system.

We thank the reviewer for their feedback. We have updated the naming scheme of the variants with point mutations and removed the extra Met residue in some of the sequences as well as any stop codons (“*”). We also demonstrate assembly competency of O432 particles with and without cargo in new Supplementary Figures 8 and 10, which show the assembly competency of all the negative and positively charged pH-responsive variants without cargo, and new panels in Supplementary Figure 7, which shows assembly competency of O432-17(-) and O432-17(+) with cargo.

Confocal microscopy. The images should show a nuclear stain or a brightfield image to provide a reference to the cell structure. Displaying the AF647 signal as white in the composite also makes co-localisation more difficult to identify. Although difficult to properly judge from the images provided, there seems to be considerable overlap of the mRuby2 and AF647 signal, suggesting that particles may be intact. Additionally, it is unclear why there would be so much LAMP2A signal in the nuclei? A FRET-based approach may be helpful to better interrogate the behaviour of these assemblies in cellulo.

We thank the reviewer for their helpful feedback. We've now included a plasma membrane stain for the A431 cells in Supplementary Figure 11 (formerly Figure 5) and a nuclear stain on the WT HeLa and HeLa EGFR KO cells in a new Supplementary Figure 12. We specifically do not make judgements on colocalization, as we believe substantially more data and images of higher resolution would be required to support such claims. Based on our efforts to use confocal microscopy to measure colocalization (described below), we agree that a FRET-based approach would be appropriate, but is beyond the scope of this work.

We attempted to estimate colocalization between HeLa and A431 cells at 3 hrs and at 16 hrs between the plug and nanoparticle markers (Figure C) and between the nanoparticle marker with LAMP2A (data not shown). We observe a decrease in colocalization percentage between 3 and 16 hours for the pH-responsive O432-17(-)_3HIS_I57V_L75A in A431 cells but not in HeLa cells, although the result is not statistically significant. We do not see a change in colocalization percentage between time points for the non-pH-responsive O432-17(-)_0HIS in either cell line. We speculate that the potential lower colocalization of the components of O432-17(-)_3HIS_I57V_L75A may be attributed to the pH decrease inside of the endosome, but without a functional or higher resolution readout, we cannot definitively conclude this.

Figure C: Colocalization percentage between nanoparticle (mRuby2-Fc) and trimeric plug (AF647) fluorescence markers incubated for 3 and 16 hours in A431 or WT HeLa cells. We compare the colocalization percentage across both time points for the pH-responsive nanoparticle (O432-17(-)_3HIS_I57V_L75A) and the less pH-responsive nanoparticle (O432-17(-)_0HIS), using the plug-less O42.1 nanoparticle as a negative control.

Additional comments:

Fig. S1A – It would be helpful to add some arrows indicating the trimer and tetramer proteins. The gel is currently hard to interpret and it is unclear where the trimeric protein band is.

We thank the reviewer for this advice. We've added markers for the sfGFP-Fc and Tetramer in Supplementary Figure 2 (formerly Supplementary Figure 1). Given that the trimer extensions result in trimer proteins with different molecular weights, we identified trimer protein bands that were selected for downstream characterization with pink stars.

Fig. S1B - Why do O42.1 particles show >4 bands in the non-denaturing PAGE? Where is the expected mobility of the band corresponding to a complete particle?

We thank the reviewer for these questions. The expected particle is over 2 megadaltons in MW. We've updated the native gel with a new native mark ladder and reassembled O42.1 particles, resulting in a cleaner band in Supplementary Figure 2 (formerly Supplementary Figure 1). The resulting O42.1 C4 and sfGFP-Fc show evidence of degradation with more than one band, although the prominent band migrates at the expected molecular weight in both native and SDS-PAGE.

Fig. S1C – One would expect the tetrameric proteins to have very similar size/shape and elute with the same retention time. How do these compare to a known reference, e.g. O42.1 C4. The authors should also identify which peaks were isolated and used in the subsequent experiments.

We thank the reviewer for their advice. We highlighted the peaks used for downstream *in vitro* assembly with pink stars in Supplementary Figure 3. We also added a new elution profile of the O42.1 C4 to Supplementary Figure 3 as well.

Fig. S1D/F – The Fc fragment used in the experiments looks to be partially truncated (S1F).

We agree that there is a very minor species of truncated sfGFP-Fc present that can be detected when a large amount of the protein is analyzed by SDS-PAGE.

Fig. S2C – Why is the tetramer retention time different between panels B and C?

We observed a shift in tetramer retention time between panels B and C in Supplementary Figure 4 (formerly Supplementary Figure 2) because the SEC was run on two different columns. Panel B was run on a Superose 6 10/300 GL while panel C was run on a Superdex 200 10/300 GL. We include these details in the figure caption for clarity.

An SDS-Page gel for the 16 designs where tetramer & trimer co-elute (before sfGFP-Fc addition) is missing.

We agree and have added a new SDS-PAGE gel in Supplementary Figure 1 and also identified the 16 designs where the trimer and tetramer co-elute and were analyzed further with pink stars.

Fig. 4d – is missing a ladder to reference the RNA length.

We agree and have added an ssRNA and low range ssRNA ladder to a new native gel in a revised Figure 4. As the native gel is non-denaturing, the ladder does not run far apart. We ran the native gel for 30 minutes at 120V to see migration of the nanoparticles.

Fig. 4E – The coomassie staining looks rather odd with empty bands, is the gel overloaded?

We thank the reviewer for pointing this out. We repeated our native gel so that the coomassie bands are more clear and not overloaded.

Page 16 – The references to Fig. S6D-E should be to S6F-G

These references have been updated.

Page 18 – The following points are contradictory - incubated up to 16 hours...after 3 hours the cells were fixed?

This text has been updated to only include a 3 hour incubation.

Fig. S5 – This sample definitely looks quite heterogeneous. The caption title states the wrong protein variant.

The caption states the correct protein variant of data obtained by negative stain electron microscopy (NS-EM). We updated the caption for clarity: **“O432-17 nanoparticles form assemblies in the presence of protein and nucleic acid cargo.”**

Table S2 - What does the star refer to in the SI for some of the amino acid sequences?

We've removed the stop codon in the amino acid sequences.

Abstract – ‘variety of molecular payloads’, two is not really a variety.

This has been updated to “protein and nucleic acid payloads”.

Abstract – ‘designed nanoparticles’ is rather vague and therefore incorrect (e.g. LNPs) Should specify that these are designed protein-based nanoparticles.

We feel that the reference to “the designed nanoparticles” discussed in the preceding sentences makes it clear that the nanoparticles are protein-based.

Intro – “require assembly and encapsulation of molecules outside the target cell”. This is unnecessarily confusing – simply stating in vitro encapsulation may be clearer.

We thank the reviewer for this feedback and updated the sentence for clarity:

“Effective nanoparticle platforms for targeted delivery require in vitro cargo encapsulation followed by target recognition, triggered nanoparticle disassembly, and controlled cargo release once inside the cell”

Reviewer #2:

Remarks to the Author:

This manuscript describes the design and characterization of a protein nanoparticle that assembles from tetrameric, trimeric, and dimer (Fc/antibody) components; the trimer furthermore is designed to disassemble at low pH based on histidine residues arranged at interfaces. The idea is to confer opening of large pores, or disassembly, under endosomal conditions. The study represents a substantial design effort, requiring a high level of expertise. Overall the experimental work to guide and confirm a successful design result is thorough and sound. The writing is clear. The manuscript will provide a notable contribution to the timely area of designing complex protein assemblies. I have the following points to consider before publications.

1) The specific particle that results could be useful in applications, but arguably what will be of most utility to others are the ideas and methods presented. In that regard, a point that should be made clearer at the abstract level is that many (45) experimental candidates formed the basis for the eventual successful outcome. Success rates for protein design efforts remain a key concern for the field, and it’s important to emphasize what a challenging effort it is (and what expertise is required) to realize success in these kinds of endeavors.

We thank the reviewer for highlighting this important point. We have added a sentence describing the potential of improving design success rates in our Discussion:

“Although we obtained one successful design from 45 experimentally tested, recent advances in computational design methodologies have significantly improved upon design success rates and should enable greater design efficiency of these types of designed structures and functionalities in the future (Anand et al. 2022; Baek et al. 2021; Jumper et al. 2021; Wang et al. 2021; Anishchenko et al. 2021; Ingraham et al. 2022; Dauparas et al. 2022; Bennett et al. 2023; de Haas et al. 2023).”

2) Following the point above, later in the paper it would be helpful to add a note or two about how the intermediate screening of designs should be interpreted, and perhaps what led to attrition. If my reading is correct, the initial pulldown results aren't shown but it is reported that there was some evidence for physical association of designed tetramer and trimer in 16 candidates, but no tests for geometrically correct assembly at that stage. From those, 4 (or 5? -- #43 doesn't seem to show the expected tetramer band?) appeared to also associate with the dimer. And from those 4 where association of three components seemed to occur, only one appeared to assemble (predominantly) into the expected cage structure based on SEC. Something to that effect should be explained if that's the right interpretation.

And what might be said about the distinction between associating vs assembling precisely as designed? Is it likely that this reflects irregular (i.e. geometrically incorrect) or partial forms of assembly? Could kinetic or entropic trapping effects be at play? Answers might not be known but it seems worth bringing up ways of thinking about what might be limiting high success rates.

We thank the reviewer for this feedback. As noted above in response to a request from Reviewer 1, we have added SDS-PAGE from our initial screens. We note that these screens did not include validation of geometrically correct assembly. Although we agree that it is intriguing and useful to consider failure modes, we do not currently have data that can support one hypothesis over another. Therefore we use the term “associate” as opposed to “assemble” when we describe these new data:.

“SDS-polyacrylamide gel electrophoresis (SDS-PAGE) identified 16 of 45 trimer designs that co-eluted with the tetramer, suggesting association between the trimeric and tetrameric components (Supplementary Fig. 1).”

3) The designed architecture here is indeed unique, but the strong emphasis (abstract and discussion) on being the first to include three structural/symmetry elements needs to be re-thought. Cannon et al. (2020) demonstrated a designed icosahedral assembly based on pentameric + trimeric + dimeric components. That paper is included in passing in group citations, but otherwise goes unnoted. That previous work was based on genetically fusing three symmetric components while the present study relies on non-covalent associations, so one could talk about the distinction. But the more broadly phrased claim of priority without

connection to the prior study doesn't seem right. [Referencing in the paper seems good otherwise.]

We thank the reviewer for this feedback and agree that Cannon et al. (2020) is an important precedent for this study. We add a clarifying detail in the Abstract and elaborate on this distinction in the Discussion:

“Designed *non-covalent* interfaces guide cooperative nanoparticle assembly from independently purified components, and a cryo-EM density map reveals that the assembled structure is very close to the computational design model.”

“We describe a general approach for reducing the porosity of protein nanomaterials by designing custom symmetric plugs that fill pores present along unoccupied symmetry axes. We used this approach to generate the first designed non-covalently associated protein nanoparticles with distinct structural components on three different symmetry axes. *An earlier study reported the design of a self-assembling protein comprising three genetically fused oligomeric domains (Cannon et al. 2020); a key distinction is that the non-covalent interactions between the components of our nanoparticles enable precise control over their assembly in vitro.*”

4) The design logic for the flow cytometry experiments near the top of page 11 is hard to understand as described. After the topic sentence for that paragraph, it would be helpful to try to elaborate in more plain language what is being devised here.

We thank the reviewer for this feedback. We streamlined and clarified this passage to read:

“We next explored the potential for disassembly of O432-17(-) and O432-17(+) nanoparticles in the pH range of the endosome and lysosome (pH 4.5–6.5; Hu et al. 2015). We developed a medium-throughput, low-concentration flow cytometry-based assay (Meanor et al. 2022; Schröter et al. 2015; Gaggero et al. 2022) that monitors the dissociation of fluorescently labeled plug trimers and antibody Fc fused to GFP (sfGFP-Fc) from the nanoparticles at low pH. We assembled O432-17 nanoparticles using the tetramer, an Alexa Fluor 647 (AF647)-conjugated trimeric plug, and a 50/50 mixture of α -myc mouse mAb and sfGFP-Fc (Fig. 5a). The assembled nanoparticles were loaded onto 3 μ m myc tag-coated polystyrene beads and incubated in citrate-phosphate buffers ranging from pH 4.2 to pH 7.5. The beads were collected by centrifugation and resuspended in 25 mM Tris, 500 mM NaCl, pH 8 to normalize the fluorescence intensity of AF647 and sfGFP, and then analyzed by flow cytometry (Fig. 5b). In this assay, loss of AF647 and sfGFP signal indicate disassembly and release of plug trimers and sfGFP-Fc from the nanoparticles, respectively. The O432-17(-) plug design (named O432-17(-)_2HIS), which was based on a pH-responsive helical bundle with two histidine hydrogen bond networks that dissociates into monomers at pH 4.9 (Boyken et al. 2019), did not show reduction in AF647 or sfGFP fluorescence until pH

5.1 or below (Fig. 5c). The observed pH of disassembly of the O432-17(-)_2HIS trimer is similar to that of O432-17(-)_0HIS, a negative control trimer variant with all histidines substituted with asparagine (Fig. 5d). We hypothesize that the dampened sensitivity of the trimeric plug to pH relative to the original pH-responsive helical bundle is due to stabilization afforded by nanoparticle assembly (Wargacki et al. 2021).”

Minor comments:

- The earlier O42.1 assembly comes up by name at the top of page 5 but without a reference or further description. I think this is Divine 2021? Please clarify.

This reference has been updated to (Divine et al., 2021).

- Fig S1 A. The reader doesn't have a way of evaluating this gel without the MW of the trimers. At least for the cases that are being asserted to be successes, the trimer MW should be given, perhaps alongside arrow markers for the bands on the gel that are being ascribed to the trimer components.

We thank the reviewer for this helpful suggestion. We've added markers for the sfGFP-Fc and tetramer in Supplementary Figure 2. Given that the trimer extensions result in trimer proteins with different molecular weights, we identified trimer protein bands that were selected for downstream characterization with pink stars.

- Where OMAC experiments are being mentioned, please indicate/remind the reader which subunit is carrying the tag.

We thank the reviewer for this feedback. We updated the text to include that the tetramer contains a C-terminal His tag when the trimer and tetramers are co-expressed. When the designs were subcloned into separate expression vectors, we clarified that each trimer and tetramer contained their own N- or C-terminal affinity tag.

“To form the three-component nanoparticle, an sfGFP-Fc fusion protein (Divine et al. 2021) was added to the clarified lysates of co-expressed trimers and tetramers with only the tetrameric component containing a 6×-histidine tag, and the resulting mixtures were subjected to IMAC. All three components co-purified as assessed by SDS-PAGE and native PAGE for 5 out of the 16 designs (Supplementary Fig. 2a,b). To enable controllable nanoparticle assembly in vitro, the trimer and tetramer genes for these designs were subcloned into separate expression vectors containing an N- or C-terminal 6×-histidine tag, and the independently expressed oligomers were purified separately by size exclusion chromatography (SEC) on a Superdex 200 10/300 GL column (Supplementary Fig. 3a). The SEC elution profiles of the redesigned tetramers were

similar to those of the parent O42.1 tetramer, O42.1 C4 (Supplementary Fig. 3b) (Divine et al. 2021).”

- The figure legends are constructed strangely. E.g. “separately, D. and, “ and so on. Having a period followed by a lower case letter makes it difficult to parse what is what.

We thank the reviewer for this feedback and have updated the figures and figure captions to NSMB standards.

- Top of page 11. It would be an improvement to avoid jargon: “relaxing”

We thank the reviewer for this feedback. We have replaced this jargon with the correct phrasing: “We fit the O432-17 design model into the experimentally determined cryo-EM map using backbone refinement in Rosetta, yielding a density-refined model.”

Reviewer #3:

Remarks to the Author:

Key results

The manuscript by Yang et al. reports on the rational design of antibody-decorated hollow protein nanocages that encapsulate cargo and can controllably disassemble upon dropping the pH. The architecture of the nanocage is largely based on the previous Science paper by Divine et al. 2021, in which an octahedral porous cage was formed from de novo-designed protein tetramers and Fc fragments of antibodies. In the present study, the pores in the walls of the cage were controllably blocked by a third, trimeric protein plug component. The interface between the trimer and the tetramers features pH-tunable ionic bonds that help controllably disassemble the plug from the porous cage.

The protein plug was designed by expanding in size a small trimeric core using a combination of helical fusion and protein docking, followed by optimization with Rosetta sequence design calculations. Thousands of design variants were generated and screened to check whether the protein plug fits the size and shape of pores within the nanocage. 45 of the design variants were then experimentally produced and tested with PAGE and SEC to check for the assembling of the nanocage from the plug, tetramers and antibody components. For a smaller subset, the nanocages were characterized by DLS and cryo-EM. The experimental structures of the nanocages were compared to the initial designs and found to be in good agreement. A selected protein nanocage was then used to encapsulate RNA inside the cage structure to protect the nucleic acid cargo from nuclease digestion. The pH-responsive disassembly of the plugs from the hollow nanocage was tuned by point mutations in the subunits, and the corresponding pK_as for disassembly were measured with flow cytometry. Finally, the antibody-decorated

nanoparticles were added to eucaryotic cells to demonstrate antigen-specific cellular uptake into endosomes using confocal fluorescence microscopy as read-out.

Originality and significance

The rational design of a hollow protein nanocage with a pH-removable protein plug is novel, as is the protective encapsulation of RNA within a designed nanocage. The study is significant in protein design as helical fusion is demonstrated to help obtain the protein plug by expanding a small core to fit in size and shape to the nanocage pores. As other significant point, the designed protein architecture integrates three functions, namely encapsulation within a nanocage, specific molecular recognition, and pH-dependent disassembly. Finally, the structure may be used for targeted delivery of bioactive cargo inside cells, even though more work would be required to demonstrate the benefit for research or drug delivery. While the findings are novel and significant, more could be done to place them within the wider context of biomimetic design, nanobiotechnology, drug delivery, and other protein designs, as detailed under 'Conclusions'.

We thank the reviewer for their positive assessment of our work.

Methodology and data

The methodology uses the appropriate tools for answering the scientific questions. Most of the claims in the manuscript are strongly supported by experimental evidence. Two points should be addressed with additional experiments described under 'Improvements'. The use of statistics and treatment of uncertainties is good.

Conclusions

While the findings are novel and significant, more could be done to place them into a wider context along the following points.

In terms of biomimetic design, protein shells that encapsulate nucleic acids clearly replicate the overall principle and function of viral particles. How does the designed structure compare to other viral particles? For example, are the designed protein nanocages smaller than any viral template? What about the encapsulated volume? The symmetry of the components may also be compared.

We appreciate the reviewer's suggestion and have incorporated these points into a revised Discussion section (added text in italics):

“Our O432 nanoparticle system is capable of packaging and protecting both protein and nucleic acid cargoes, disassembles at biologically relevant pH with precise tunability, incorporates a wide variety of targeting moieties, and is readily internalized by target cells. In performing these functions, the O432 nanoparticles resemble viruses. They also resemble the smallest viruses in size, with an internal diameter (25 nm) slightly larger than adeno-associated viruses (17 nm). However, the architecture of the O432

nanoparticles differs greatly from small icosahedral viruses, many of which are constructed from multiple copies of a single capsid protein that assumes several different conformations and performs several distinct functions at different stages of the viral life cycle. By contrast, the O432 nanoparticles are constructed from three modular protein components that each perform a specific function. The tunable pH dependence makes this system a particularly attractive platform for engineering release and delivery of drugs during early stages of endosomal maturation. However, to be a broadly useful intracellular biologics delivery system, it will be necessary to incorporate endosomal escape machinery in future designs. The nanoparticles also provide a route to conditional delivery of drugs into the tumor microenvironment. Cytotoxic tumor-killing or -modulating cargoes could be packaged within the nanoparticles and directed to the tumor through targeting with tumor-specific antibodies; the pH-dependent release of cargo could minimize off-tumor toxicity and systemic exposure as compared to classic direct antibody conjugation approaches by providing an additional checkpoint on proper localization. The pH-dependent disassembly, programmability, and versatility of the O432 platform provides multiple exciting paths forward for biologics delivery.”

Within the wider context of nanobiotechnology, non-protein multimeric nanocages have previously been constructed via DNA nanotechnology. Examples include Sigl, C. et al. (2021) ‘Programmable icosahedral shell system for virus trapping’, *Nature Materials*, 20(9), pp. 1281–1289. doi:10.1038/s41563-021-01020-4 and the environmentally sensitive assembly and encapsulation Ijäs, H. et al. (2019) ‘Reconfigurable DNA origami nanocapsule for pH-controlled encapsulation and display of cargo’, *ACS Nano*, 13(5), pp. 5959–5967. doi:10.1021/acsnano.9b01857 and Hu, Y. et al. (2020) ‘Dynamic DNA assemblies in biomedical applications’, *Advanced Science*, 7(14), p. 2000557. doi:10.1002/advs.202000557. The revised introduction or discussion of the manuscript should mention that nanocages and stimulus responsiveness can also be designed with DNA.

We thank the reviewer for pointing this out, and have added the suggested references to the revised Introduction.

Relating to drug delivery, the biomedical application potential of the protein cages needs to be discussed. Firstly, which cargo would benefit most from cellular delivery by the new nanocages? Second, protein cage disassembly within the acidic endosomes is not sufficient as the cargo must also be released from the endosome. Indeed, accumulation of the fluorescent protein seems to appear in the microscopy images of the cells in Fig. 5. Furthermore, what are benchmark delivery methods against which the protein cages should be compared to in terms of uptake, release, cargo size, a.o.?

Although we are excited by the prospects of the O432 nanoparticles for applications in drug delivery, we agree with the reviewer’s earlier comment that more work will be required to realize

this goal. As a result, we feel that it is somewhat premature to discuss these topics in the present manuscript.

Finally, the principle of creating a plug for a protein hole may be applied for designing other protein structures such as porous and reversibly closing 2D lattices or gated membrane channels. The outlook of the manuscript should mention the above points to better appreciate the significance of the findings.

We thank the reviewer for this feedback. We added a sentence in the Discussion about extending our plug design method to other architectures:

“We note that our symmetric plug design approach could be extended to designing reversible closures for other architectures, such as porous 2D lattices or gated membrane channels.”

Improvements:

1. The effect of porosity on cargo retention

A major claim of the paper is that adding a plug into a porous nanocage can encapsulate cargo inside the protein carrier. By adding the plug, the pore size decreases from 13 nm to 3 nm, based on cryo-EM data. However, the cargoes tested are bound during the octahedron assembly and -from the reviewer's impression- the effect on pore size on cargo retention is never explicitly tested in the publication. The effect of pore size on cargo retention should be experimentally tested to better appreciate the usefulness of the nanocage (European Journal of Pharmaceutical Sciences, 120, pp. 199–211. doi:10.1016/j.ejps.2018.05.004.).

We thank the reviewer for this feedback, and refer them to our response to Reviewer #1 above.

2. The pH-dependent release of the cargo

Another claim of the paper is how the pH-dependent detachment of the trimer-plug can control cargo release from the nanocage lumen. The manuscript only demonstrates the detachment of the trimer from the nanocage which is, however, not filled with cargo. To close this gap, the authors should test the pH-dependent release of the RNA (or other) cargo.

We thank the reviewer for this suggestion, and again refer them to our response to Reviewer #1 above.

Minor points: Most experiments are clearly described and expanded in the supplementary information. However, there are a few points that need clarification.

1. What is the particle formation yield for the selected design? Would it be possible to give a % estimate normalized by the concentration of Fc added?

We thank the reviewer for raising this question. We added an estimate for particle formation yield by measuring area under the curve of each SEC trace using the Unicorn 7.3 Software and added the following sentences in the Results and Methods sections:

“From the area under the SEC peak, we estimate a total particle yield of 95% (see methods).”

“The O432-17 design eluted in the void volume, owing to the increased diameter from the additional Fab domains, and had an 80% nanoparticle yield with a 15% trimer component peak as measured by the area under the SEC curve (Fig. 2b).”

“Percent yield of nanoparticle formation was estimated via an area under the SEC curve in Unicorn 7.3 Software (Cytiva). A baseline of 10 mAU was used to determine peak height and width.”

2. Figure 5: The data on cellular uptake lack more quantitative analysis of the results. For example, data on average fluorescence per cell and number of cells with fluorescence signals should be presented.

We thank the reviewer for this feedback, and refer them to our response to Reviewer #1 above. We provide the requested quantitative analysis and associated methods in the a new section in the Methods, a new Supplementary Figure 11 and Supplementary Figure 12, and a new Supplementary Table 9:

“Nanoparticle uptake quantification

To quantify the percentage of cells with cage bound the Multi-point tool in Fiji (ImageJ) (Schindelin et al. 2012) was used to count nuclei in each image. The mRuby2 channel was used to identify and count cells, again, using the multi-point tool, showing nanoparticle localization above background signal using the identified nuclei as a guide for cell position. For cells positive for nanoparticle signal LAMP2A staining was used to aid in determining cell shape and the freehand selection tool was used to draw manual ROIs around the cell circumference. All ROIs for positive cells were measured for cell area and integrated intensity in the mRuby2 channel for integrated intensity/cell area quantification. Cellular data was plotted and statistical tests were done in GraphPad Prism version 9.3.1 for Windows, GraphPad Software, San Diego, California USA, www.graphpad.com..”

3. Figure S1-F. There is no band for the condition Tetramer+Fc. However, Tetramer + Fc should produce the cage structure or at least just some aggregates. Why is there no band?

The total soluble concentration of purified Tetramer + Fc was below the detection limit for SDS-PAGE. For clarity, we removed this band from the screening gel as it provided no additional information, as each individual component is already included in the gel.

4. Figure S1 caption: E is used to refer to subfigure F and subfigure G is not referred to.

We thank the reviewer for catching this and have updated the figures and figure captions to NSMB standards.

5. The nanoparticles were incubated with cells in serum-free media. What happens when cells are incubated with serum-containing media? The latter condition would mimic biomedical applications.

Please see our response to a similar query from Reviewer #1 above.

6. It is clear that the packaging/loading of molecular cargos is highly dependent on the electrostatic interaction between the plug and the cargo. Could there be other factors which affect the packaging of cargo in the nanocages, such as size/molecular weight of the cargo? What is the expected maximum cargo size? What is the expected correlation between cargo size and the assembly efficiency of the nanocages?

We thank the reviewer for raising these important questions. We now include an approximation for the number of GFP cargoes per nanoparticle to address the encapsulation volume and amount of volume occupied by the encapsulated cargo:

“Based on measurements of fluorescence intensity and UV/Vis absorbance, we estimate approximately 9 to 10 pos36GFPs are packaged per nanoparticle in 200 mM NaCl, occupying roughly 40% of the interior volume. Including 1 M NaCl in the buffers used during packaging and SEC prevented packaging (Fig. 4f), demonstrating that GFP packaging is largely driven by electrostatic interactions between the cargo and nanoparticle interior. In vitro assembly of O42.1 (Divine et al. 2021) with pos36GFP revealed no cargo packaging, indicating that presence of the designed trimeric plug is necessary for cargo packaging (Fig. 4g).”

Furthermore, as described above in a response to Reviewer #1, to address other factors that can affect the packaging and protection of cargo in the nanoparticles, we added two different nuclease challenges in the packaging of RNA in which we found that the smaller of the nucleases (RNase A) was able to still degrade the cargo while the larger of the nucleases (Benzonase) did not.

Clarity and context:

The text is overall very clearly written.

Minor points:

1. Several places in the publication refer to “incubations at a concentration per monomer”. What is defined as monomer? Does this include all elements in the nanoparticle including the antibodies or only the monomers forming the tetramer and trimer?

We added “of each component” to clarify that assemblies were mixed in monomeric ratios for all components. We first introduce antibodies and Fcs as dimers, so we treat each antibody or Fc as having two monomers in the assembly ratio context.

2. Figure 1B: Which of the fused trimer was chosen in the experimental screening process?

We added a description in the main text to indicate that the trimeric fusions were a diverse set and were all used in the docking and experimental screening process.

3. Page 8, line 12: “The optimal assembly ratio per protomer of purified trimeric plug, tetramer, and Fc was determined to be 1.1:1.1:1.” Is this the stoichiometric ratio? How was this ratio obtained? The following reference should be considered. Nature Communications, 2015 6(1). doi:10.1038/ncomms7203.

The optimized ratio was determined following two-component assembly tests where we discovered the cooperative nature of the O432 assembly in the excess of trimeric plug. The above response shows the updated text.

4. Page 9: “This cooperativity simplifies preparation of the three component nanoparticles as it prevents incomplete assembly of nanoparticles containing two out of the three components”. Would not Fc and the tetramer interact on their own?

Please see our response above to Reviewer #1, where we clarify the Fc and tetramer association as an unstable assembly.

5. Page 14 “After 30 minutes, the beads were resuspended and brought back to pH 7.5 to...” Did the authors mean: “beads were collected by centrifugation and resuspended in...”?

Thank you for catching this oversight, which we have fixed in the revised text.

6. Page 14/15. At the beginning of the section, the target pKa is not mentioned. This is slightly confusing.

We added the following in the text for clarity:

“We next explored the potential for disassembly of O432-17(-) and O432-17(+) nanoparticles in the pH range of the endosome and lysosome (pH 4.5–6.5; Hu et al. 2015).”

Decision Letter, first revision:

Message: 13th Sep 2023

Dear Dr. Yang,

Thank you again for submitting your manuscript "Computational design of non-porous pH-responsive antibody nanoparticles". We now have comments (below) from the 3 reviewers who evaluated your paper. In light of those reports, we remain interested in your study and would like to see your response to the comments of the referees, in the form of a revised manuscript.

You will see that while all three Reviewers found the manuscript improved in revision, Reviewer #1 still has some outstanding concerns regarding the reporting of the methods, some of the interpretations, and the calculation of the assembly yields. Please be sure to address/respond to all concerns of the referees in full in a point-by-point response and highlight all changes in the revised manuscript text file. If you have comments that are intended for editors only, please include those in a separate cover letter.

We expect to see your revised manuscript within 6 weeks. If you cannot send it within this time, please contact us to discuss an extension; we would still consider your revision, provided that no similar work has been accepted for publication at NSMB or published elsewhere.

Reporting Summary:
<https://www.nature.com/documents/nr-reporting-summary.pdf>

When submitting the revised version of your manuscript, please pay close attention to our [href="https://www.nature.com/nature-portfolio/editorial-policies/image-integrity">Digital Image Integrity Guidelines. and to the following points below:](https://www.nature.com/nature-portfolio/editorial-policies/image-integrity)

- that unprocessed scans are clearly labelled and match the gels and western blots presented in figures.
- that control panels for gels and western blots are appropriately described as loading on

sample processing controls

-- all images in the paper are checked for duplication of panels and for splicing of gel lanes.

Please note that all key data shown in the main figures as cropped gels or blots should be presented in uncropped form, with molecular weight markers. These data can be aggregated into a single supplementary figure item. While these data can be displayed in a relatively informal style, they must refer back to the relevant figures. These data should be submitted with the final revision, as source data, prior to acceptance, but you may want to start putting it together at this point.

SOURCE DATA: we request that authors provide, in tabular form, the data underlying the graphical representations used in figures. This is to further increase transparency in data reporting, as detailed in this editorial (<http://www.nature.com/nsmb/journal/v22/n10/full/nsmb.3110.html>). Spreadsheets can be submitted in excel format. Only one (1) file per figure is permitted; thus, for multi-paneled figures, the source data for each panel should be clearly labeled in the Excel file; alternately the data can be provided as multiple, clearly labeled sheets in an Excel file. When submitting files, the title field should indicate which figure the source data pertains to. We encourage our authors to provide source data at the revision stage, so that they are part of the peer-review process.

Data availability: this journal strongly supports public availability of data. All data used in accepted papers should be available via a public data repository, or alternatively, as Supplementary Information. If data can only be shared on request, please explain why in your Data Availability Statement, and also in the correspondence with your editor. Please note that for some data types, deposition in a public repository is mandatory - more information on our data deposition policies and available repositories can be found below: <https://www.nature.com/nature-research/editorial-policies/reporting-standards#availability-of-data>

While we encourage the use of color in preparing figures, please note that this will incur a

charge to partially defray the cost of printing. Information about color charges can be found at <http://www.nature.com/nsmb/authors/submit/index.html#costs>

[redacted]

Sincerely,
Sara

Sara Osman, Ph.D.
Associate Editor
Nature Structural & Molecular Biology

Reviewers' Comments:

Reviewer #1:

Remarks to the Author:

The manuscript by Yang et al. describes expanding upon the design of a pH-responsive trimeric protein and tetrameric antibody binding protein to produce novel protein cages. The main aim is to produce a protein-based encapsulation system that can disassemble at biologically relevant pH values and release cargo. In this revised version of the manuscript the authors have done a very commendable job to address the reviewers' various concerns. However, there are still open questions about this interesting new system and some additional clarification should be added before publication.

Specific comments:

Cell studies. This reviewer could not find information anywhere on the concentrations of particles used in the cell experiments. Were the concentrations of O432-17-CTX and O432-17-Fc matched in terms of particles, or in terms of [mRuby2-Fc]? A 50% lower concentration of particles is significant, so this should be addressed. Additionally, although

a conjugation protocol is given, the degree of labelling of trimers with AF647 is not described. For future experiments, flow cytometry would be beneficial for identifying percentages of cells that exhibit uptake.

The brief discussion that the authors have added in response to reviewer 2's comment on design success rates is a good addition. However, it is worth elaborating upon this with regard to the fact that cages formed with the tuned pH-responsive designs could not form with cargo. This is an unexpected result, and opens some interesting questions about the system. The explanation on page 20 is rather vague, and doesn't directly address that fact the trimers are the component that will interact with GFP. This strong electrostatic interaction may compete with the trimer-tetramer interactions. This appears not to be the case with O432-17(-); are the pH-tuned trimers sufficiently less stable to allow GFP to disassemble or distort them? In future some melting experiments may be informative. This aspect provides another means to tune the system and could be addressed from engineering of either cage components or cargo.

Page 16, para 2: "little cargo was left after washing since there is no plug present to prevent cargo leakage from the nanoparticle" The driving force for association of GFP with the particles is electrostatics. Since the negative charge is introduced to the trimer, a particle without trimeric component would not be expected to bind GFP. As such, the plugging concept is not the most relevant explanation.

Yields. The assembly yields presented are questionable, and the authors may consider simply removing them. The area under the SEC elution peak only states the % of injected material that elutes within a certain retention volume, not the total yield of assembly. Presumably the samples for SEC were centrifuged or filtered to remove any aggregated material, as such this would not be observed in the SEC trace. This is apparently an issue for most of the variants based on the low signal in SI Fig. 3f. A proper assembly yield would be calculated based on the input amount of protein and output of cages.

Page 14, para 2. The following is misleading: "SEC of the trimer and the resulting assembly, DLS, and NS-EM confirmed three-component nanoparticle assembly of O432-17(+) with and without the presence of the pegRNA" Here, only the NS-EM confirms assembly in the presence of RNA, and the sample looks heterogenous. However, the data in Fig.4b/c is relevant here and if a sample of the cage without RNA was also run, this could provide a very good comparison.

Minor comments:

Introduction: "quite porous" is vague, these cages have eight YxYxY nm triangular pores.

Page 4, para 1 – the authors could state the reason for choosing o42 rather than the i52.3 particles as a starting point, is it the pore size?

Page 4, para 1 – "lower pH's" -> lower pH values

Page 23, para 3 – there is no evidence to support the claim that the cage protects protein cargo.

Page 23, para 3 – Is the internal diameter of 25nm really correct? The O42 particle is smaller than AAV.

SI Fig. 1, the gel lanes should be labelled with the variant numbers.

SI Fig. 3f, the contrast should be increased so that the bands can be identified. It appears that the concentration of protein is just very low. Presumably there was precipitation of protein during these assemblies.

SI Fig. 4, panel c. Missing space for 100uM in legend.

SI, page 14 – “trimeric plug variants containing 10x TCEP” – a molar concentration would be more appropriate, as well as stating the final concentration in the reaction mixture.

Reviewer #2:

Remarks to the Author:

The revisions have addressed my concerns.

Reviewer #3:

Remarks to the Author:

The rebuttal is convincing and includes several new experiments. All raised points have been settled. The manuscript is now publishable in Nature Structural & Molecular Biology.

Author Rebuttal, first revision:

Reviewers' Comments:

Reviewer #1:

Remarks to the Author:

The manuscript by Yang et al. describes expanding upon the design of a pH-responsive trimeric protein and tetrameric antibody binding protein to produce novel protein cages. The main aim is to produce a protein-based encapsulation system that can disassemble at biologically relevant pH values and release cargo. In this revised version of the manuscript the authors have done a very commendable job to address the reviewers' various concerns. However, there are still open questions about this interesting new system and some additional clarification should be added before publication.

Specific comments:

Cell studies. This reviewer could not find information anywhere on the concentrations of particles used in the cell experiments. Were the concentrations of O432-17-CTX and O432-17-Fc matched in terms of particles, or in terms of [mRuby2-Fc]? A 50% lower concentration of

particles is significant, so this should be addressed. Additionally, although a conjugation protocol is given, the degree of labelling of trimers with AF647 is not described. For future experiments, flow cytometry would be beneficial for identifying percentages of cells that exhibit uptake.

We thank the reviewer for the feedback. We include the concentrations of particles used in cell experiments as monomeric concentrations of each component in the methods. We've clarified this statement to also include particle concentration.

“A final monomeric concentration of 10 nM of O432-17-CTX or O432-17-Fc nanoparticles (417 pM final nanoparticle concentration) were incubated with cultured cells in DMEM or DMEM supplemented with 10% fetal bovine serum (FBS)”

We also added a supplementary table describing the degree of labeling of all trimers with AF647 and the relevant methods in the conjugation protocol.

“Degree of labeling was estimated from a fluorophore extinction coefficient of 265,000 M⁻¹cm⁻¹ and absorbance measurements at 280 and 650 nm using a NanoDrop 8000 spectrophotometer.”

Supplementary Table 10: AF647-Nanoparticle conjugation efficiencies and related values.

Sample	AF647 Conjugation Efficiency %	Number of AF647 fluorophores per nanoparticle
O432-17(-)_2HIS-C3	75%	18
O432-17(-)_3HIS_I57V_L75A-C3	21%	5
O432-17(-)_0HIS-C3	63%	15
O432-17(-)_3HIS_I57V-C3	37%	9
O432-17(-)_3HIS_L75A-C3	30%	7
O432-17(+)_2HIS-C3	21%	5
O432-17(+)_3HIS_I57V-C3	62%	15
O432-17(+)_3HIS_L75A-C3	65%	16
O432-17(+)_3HIS_I57V_L75A-C3	61%	15

O432-17(+)_0HIS-C3	72%	17
-----	----

The brief discussion that the authors have added in response to reviewer 2's comment on design success rates is a good addition. However, it is worth elaborating upon this with regard to the fact that cages formed with the tuned pH-responsive designs could not form with cargo. This is an unexpected result, and opens some interesting questions about the system. The explanation on page 20 is rather vague, and doesn't directly address that fact the trimers are the component that will interact with GFP. This strong electrostatic interaction may compete with the trimer-tetramer interactions. This appears not to be the case with O432-17(-); are the pH-tuned trimers sufficiently less stable to allow GFP to disassemble or distort them? In future some melting experiments may be informative. This aspect provides another means to tune the system and could be addressed from engineering of either cage components or cargo.

We thank the reviewer for this feedback. We have added a clearer hypothesis into this section to explain this interesting result:

“These pH-responsive, O432-17(-) variants did not encapsulate pos36GFP, which we attribute to destabilization of the trimers by the point mutations enabling disassembly at more basic pH values and the designed electrostatic interactions between the trimer with the pos36GFP cargo that may compete with the trimer-tetramer interaction. This sensitivity opens up additional opportunities to tune the system by engineering either the cage components or the cargo. For now, given the lack of encapsulation, we were only able to investigate the pH-mediated release of encapsulated cargo for the O432-17(-) variant (O432-17(-)_2HIS; **Fig. 4k**).”

Page 16, para 2: “little cargo was left after washing since there is no plug present to prevent cargo leakage from the nanoparticle” The driving force for association of GFP with the particles is electrostatics. Since the negative charge is introduced to the trimer, a particle without trimeric component would not be expected to bind GFP. As such, the plugging concept is not the most relevant explanation.

We thank the reviewer for this feedback and clarified this statement in the text (new text in italics):

“Incubation of the O42.1 nanoparticles at either low or high pH (4.2 and 8, respectively) did not result in substantial release of pos36GFP cargo into the supernatant, as *the absence of the plug precluded cargo packaging and retention.*”

Yields. The assembly yields presented are questionable, and the authors may consider simply removing them. The area under the SEC elution peak only states the % of injected material that elutes within a certain retention volume, not the total yield of assembly. Presumably the samples for SEC were centrifuged or filtered to remove any aggregated material, as such this would not be observed in the SEC trace. This is apparently an issue for most of the variants based on the low signal in SI Fig. 3f. A proper assembly yield would be calculated based on the input amount of protein and output of cages.

We thank the reviewer for this feedback and have removed the assembly yields as suggested.

Page 14, para 2. The following is misleading: “SEC of the trimer and the resulting assembly, DLS, and NS-EM confirmed three-component nanoparticle assembly of O432-17(+) with and without the presence of the pegRNA” Here, only the NS-EM confirms assembly in the presence of RNA, and the sample looks heterogenous. However, the data in Fig.4b/c is relevant here and if a sample of the cage without RNA was also run, this could provide a very good comparison.

We thank the reviewer for this feedback and have added a new Supplementary Figure 10e showing a gel similar to that in Figure 4b,c, but including the O432-17(+) nanoparticle with and without RNA. We have also pasted this gel below for convenience. The gel shows similar migration of the nanoparticle in the presence and absence of co-migrating RNA cargo. To report these data and clarify, we have revised our earlier sentence to the following (in italics):

“One variant, referred to as O432-17(+), showed comigration of nucleic acid and nanoparticle with and without Benzonase treatment (**Fig. 4b**). *The RNA-packaging three-component assembly migrated similarly to an assembly reaction lacking RNA cargo, and negatively stained electron micrographs and corresponding 2D class averages also closely resembled empty three-component nanoparticles (Supplementary Fig. 7a,b; Supplementary Fig. 10a-f).*”

Minor comments:

Introduction: “quite porous” is vague, these cages have eight YxYxY nm triangular pores.

We thank the reviewer for this feedback and have added more detail in this sentence and in a sentence in the first paragraph of the Results section (new text in italics):

“While such antibody nanoparticles can activate signaling through a variety of cell surface receptors, they *have large pores on certain symmetry axes*, which complicates the packaging and retention of molecular cargoes.”

“We focused on octahedral antibody nanoparticles (O42.1) constructed from a C4-symmetric designed tetramer and C2-symmetric IgG dimers, which are aligned along the C4 and C2 symmetry axes of the octahedral architecture (Divine et al. 2021) (**Fig. 1a**). *The eight C3 axes in this nanoparticle are unoccupied by protein and feature triangular pores 13 nm in length. On this open C3 axis, we aimed to incorporate a designed pH-dependent C3 trimer (Boyken et al. 2019) and tune it to disassemble at pH values corresponding to the native environment of the endosome (Fig. 1b).*”

Page 4, para 1 – the authors could state the reason for choosing o42 rather than the i52.3 particles as a starting point, is it the pore size?

Yes, the large pore size was one of our selection criteria. The sentence we added in the Results section in response to the previous point clarifies this.

Page 4, para 1 – “lower pH’s” -> lower pH values

We thank the reviewer for this feedback and have updated the text as suggested.

Page 23, para 3 – there is no evidence to support the claim that the cage protects protein cargo.

We thank the reviewer for this catch and have updated this sentence to read (new text in italics):

“Our O432 nanoparticle system is capable of packaging both protein and nucleic acid cargoes, *protects nucleic acid cargoes from degradation*, disassembles at biologically relevant pH with precise tunability, incorporates a wide variety of targeting moieties, and is readily internalized by target cells.”

Page 23, para 3 – Is the internal diameter of 25nm really correct? The O42 particle is smaller than AAV.

The internal diameter measured at the narrowest point is 25 nm, compared to AAV (PDB ID 1LP3) which measured 18 nm, whereas the external diameter, ignoring Fab domains, is 34 nm, compared to AAV which measures 28 nm.

SI Fig. 1, the gel lanes should be labelled with the variant numbers.

We thank the reviewer for this feedback and added variant numbers to the gel lanes.

SI Fig. 3f, the contrast should be increased so that the bands can be identified. It appears that the concentration of protein is just very low. Presumably there was precipitation of protein during these assemblies.

We thank the reviewer for their feedback and increased the contrast of the gel.

SI Fig. 4, panel c. Missing space for 100uM in legend.

We thank the reviewer for this catch and have updated the figure legend.

SI, page 14 – “trimeric plug variants containing 10x TCEP” – a molar concentration would be more appropriate, as well as stating the final concentration in the reaction mixture.

We thank the reviewer for this feedback and included a sentence with molar concentrations of each reagent in the final mixture.

“The final reaction mixture contained 75 μM Alexa Fluor, 15 μM trimeric plug, and 150 μM TCEP.”

Reviewer #2:

Remarks to the Author:

The revisions have addressed my concerns.

We thank the reviewer for the positive assessment of our work.

Reviewer #3:

Remarks to the Author:

The rebuttal is convincing and includes several new experiments. All raised points have been settled. The manuscript is now publishable in Nature Structural & Molecular Biology.

We thank the reviewer for the positive assessment of our work.

Decision Letter, second revision:

Message: Our ref: NSMB-A47396B

10th Oct 2023

Dear Dr. Yang,

Thank you for submitting your revised manuscript "Computational design of non-porous pH-responsive antibody nanoparticles" (NSMB-A47396B). It has now been editorially assessed and the editorial team find that it has improved in revision, and therefore we'll be happy in principle to publish it in Nature Structural & Molecular Biology, pending minor revisions to satisfy the referees' final requests and to comply with our editorial and formatting guidelines.

We are now performing detailed checks on your paper and will send you a checklist detailing our editorial and formatting requirements in the next two to three weeks. Please do not upload the final materials and make any revisions until you receive this additional information from us.

To facilitate our work at this stage, it is important that we have a copy of the main text as a word file. If you could please send along a word version of this file as soon as possible, we would greatly appreciate it; please make sure to copy the NSMB account (cc'ed above).

Sincerely,
Sara

Sara Osman, Ph.D.
Associate Editor
Nature Structural & Molecular Biology

Final Decision Letter:

Message: 22nd Mar 2024

Dear Dr. Baker,

We are now happy to accept your revised paper "Computational design of non-porous pH-responsive antibody nanoparticles" for publication as an Article in Nature Structural & Molecular Biology.

After the grant of rights is completed, you will receive a link to your electronic proof via email with a request to make any corrections within 48 hours. If, when you receive your

proof, you cannot meet this deadline, please inform us at rjsproduction@springernature.com immediately.

Your paper will be published online soon after we receive proof corrections and will appear in print in the next available issue. You can find out your date of online publication by contacting the production team shortly after sending your proof corrections.

If you have not already done so, we strongly recommend that you upload the step-by-step protocols used in this manuscript to the Protocol Exchange. Protocol Exchange is an open online resource that allows researchers to share their detailed experimental know-how. All uploaded protocols are made freely available, assigned DOIs for ease of citation and fully searchable through nature.com. Protocols can be linked to any publications in which they are used and will be linked to from your article. You can also establish a dedicated page to collect all your lab Protocols. By uploading your Protocols to Protocol Exchange, you are enabling researchers to more readily reproduce or adapt the methodology you use, as well as increasing the visibility of your protocols and papers. Upload your Protocols at www.nature.com/protocolexchange/. Further information can be found at

www.nature.com/protocolexchange/about.

Please note that *Nature Structural & Molecular Biology* is a Transformative Journal (TJ). Authors may publish their research with us through the traditional subscription access route or make their paper immediately open access through payment of an article-processing charge (APC). Authors will not be required to make a final decision about access to their article until it has been accepted. Find out more about Transformative Journals

Sincerely,
Sara

Sara Osman, Ph.D.
Associate Editor
Nature Structural & Molecular Biology